



# Western disturbances and climate variability: a review of recent developments

Kieran M. R. Hunt[1,2], Jean-Philippe Baudouin[3], Andrew G. Turner[1,2], A. P. Dimri[4,5], Ghulam Jeelani[6], Pooja [4], Rajib Chattopadhyay[7,8], Forest Cannon[9], T. Arulalan[10,7], M. S. Shekhar[11], Sabin T. P.[8], and Eliza Palazzi[12]

[1]Department of Meteorology, University of Reading, Reading, UK
[2]National Centre for Atmospheric Sciences, University of Reading, Reading UK
[3]Department of Geosciences, University of Tübingen, Tübingen, Germany
[4]School of Environmental Sciences, Jawaharlal Nehru University, New Delhi, India
[5]Indian Institute of Geomagnetism, Mumbai, India
[6]Department of Earth Sciences, University of Kashmir, Srinagar, India
[7]India Meteorological Department, Ministry of Earth Sciences, New Delhi, India
[8]Indian Institute of Tropical Meteorology, Pune, India
[9]Scripps Institution of Oceanography, University of California, San Diego, USA
[10]Centre for Atmospheric Sciences, Indian Institute of Technology Delhi, New Delhi, India
[11]Defence Geoinformatics Research Establishment, Chandigarh, India
[12]Department of Physics, University of Turin, Torino, Italy

**Correspondence:** Kieran M. R. Hunt (k.m.r.hunt@reading.ac.uk)

**Abstract.** Western disturbances (WDs) are synoptic-scale weather systems embedded within the subtropical westerly jet. Manifesting as upper-level troughs often associated with a lower-tropospheric low over Western India, they share some dynamical features with extratropical cyclones. WDs are most common during the boreal winter (December to March), during which they bring the majority of precipitation – both rain and snow – to the Western Himalaya, as well as to surrounding areas of north

India, Pakistan and the Tibetan Plateau. WDs are also associated with weather hazards such as heavy snowfall, hailstorms, fog, cloudbursts, avalanches, frost, and coldwaves.

In this paper, we review the recent understanding and development on WDs. Recent studies have collectively made use of novel data, novel analysis techniques, and the increasing availability of high-resolution weather and climate models. This review is separated into six main sections – structure and thermodynamics, precipitation and impacts, teleconnections, modelling

experiments, forecasting at a range of scales, and paleoclimate and climate change – each motivated with a brief discussion of the accomplishments and limitations of previous research.

A number of step changes in understanding are synthesised. Use of new modelling frameworks and tracking algorithms has significantly improved knowledge of WD structure and variability, and a more frequentist approach can now be taken. Improved observation systems have helped quantification of water security over the Western Himalaya. Convection-permitting

models have improved our understanding of how WDs interact with the Himalayas to trigger natural hazards. Improvements in paleoclimate and future climate modelling experiments have helped to explain how WDs and their impacts over the Himalaya



respond to large-scale natural and anthropogenic forcings. We end by summarising unresolved questions and outlining key future WD research topics.





## Contents









**Figure 1.** Map of the Western Himalaya, Karakoram and Hindu Kush mountain ranges. Glacierised regions, large rivers, and selected mountains are marked. National borders are for illustration only, and do not necessarily represent the views of the authors.

## 1 Introduction

The Indian subcontinent experiences four distinct seasons. The monsoon, lasting from June to September, brings the majority of its annual precipitation. This is preceded by the pre-monsoon (March to May), a typically dry, warm season associated with the hottest temperatures of the year; and followed by the post-monsoon (October to November), associated with cooler and less cloudy conditions – except over southeast India, which gets most of its annual rainfall during this season from the so-called northeast monsoon. The fourth season, winter, lasts from December to February. This is a wet season for much of the northern subcontinent, including the Hindu Kush, Karakoram, and Western Himalayas (WH, for locations of these ranges and other important physical features of the region, see Fig. 1) as western disturbances (WDs) – a westward-moving synoptic-scale trough embedded in the subtropical westerly jet – impact the region bringing heavy rainfall to the lower foothills and plains,





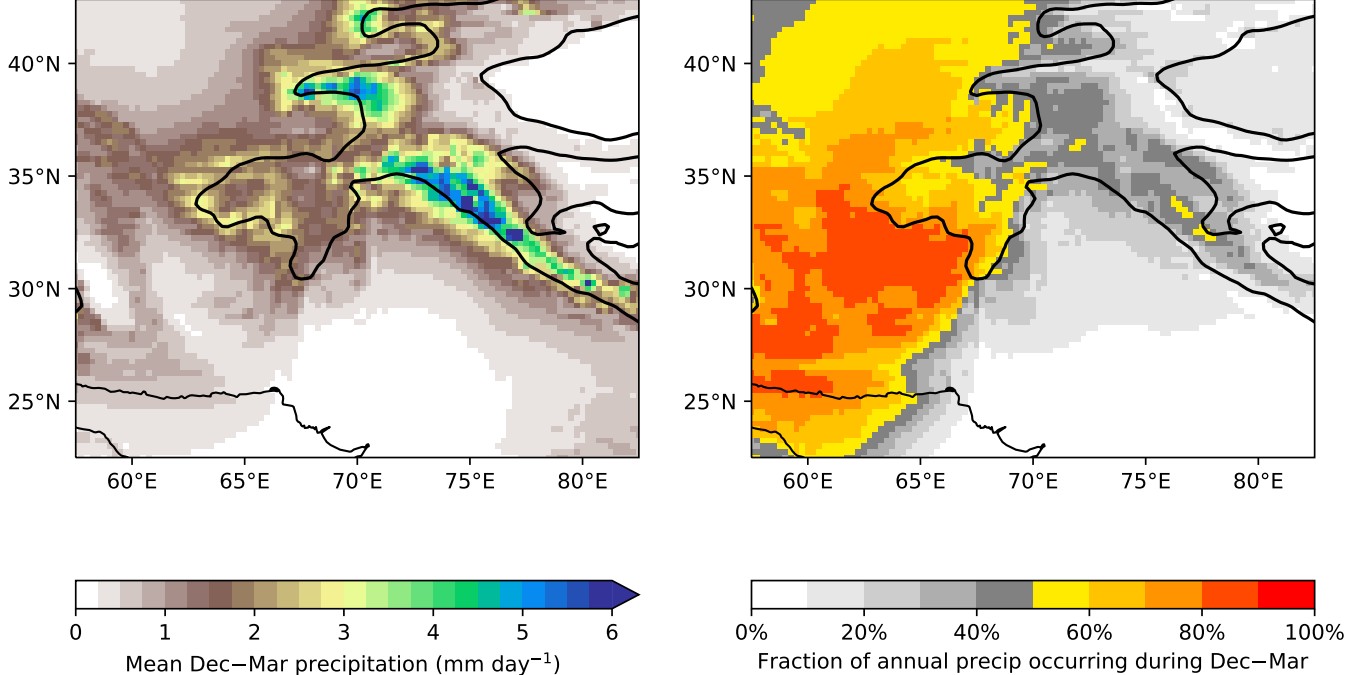

**Figure 2.** Climatological winter (December to March) precipitation over the Western Himalaya and Hindu Kush region. Left: mean winter precipitation. Right: fraction of annual precipitation that falls during the winter months. Black contours denote smoothed orography at heights of 1, 2.5, and 5 km. Precipitation data are from ERA5, covering 1950–2021.

and heavy snowfall to the mountains. The Hindu Kush receives up to 80% of its annual precipitation during the winter months, whereas the WH receives a little less than half – with this fraction decreasing quickly towards the south and east (Fig. 2).

## 1.1 What are western disturbances?

WD cyclogenesis occurs to the west of the Hindu Kush and Himalayas (Mull and Desai, 1947; Pisharoty and Desai, 1956), over regions such as the Mediterranean Sea (Madhura et al., 2015). They then propagate as troughs embedded within the subtropical westerly jet (Singh, 1971) until they reach South Asia. Early studies established that WDs move eastward at speeds of 6–12 m s$^{-1}$, although they disagreed on their lifetime–usually because of inconsistent definitions of genesis and decay – they typically fall in the range of 2 to 12 days (Datta and Gupta, 1967; Rao and Srinivasan, 1969; Chattopadhyay, 1970; Rao and Rao, 1971; Subbaramayya and Raju, 1982). Rao and Srinivasan (1969) detected an average of 6 to 7 WDs per month in winter; in contrast, WDs are much rarer during the summer months (Pisharoty and Desai, 1956), although more recent studies have discussed their importance when interacting with the monsoon trough (e.g. Chevuturi and Dimri, 2016).

WDs typically pass over Iran, Iraq, Afghanistan, Pakistan and India; but occasionally make it as far east as Nepal, Bangladesh and China (Pisharoty and Desai, 1956; Rao and Srinivasan, 1969). Their impacts are usually greatest over north India and the WH, where they bring coldwaves (Mooley, 1957; De et al., 2005), fog (Rao and Srinivasan, 1969; Syed et al., 2012; Dimri





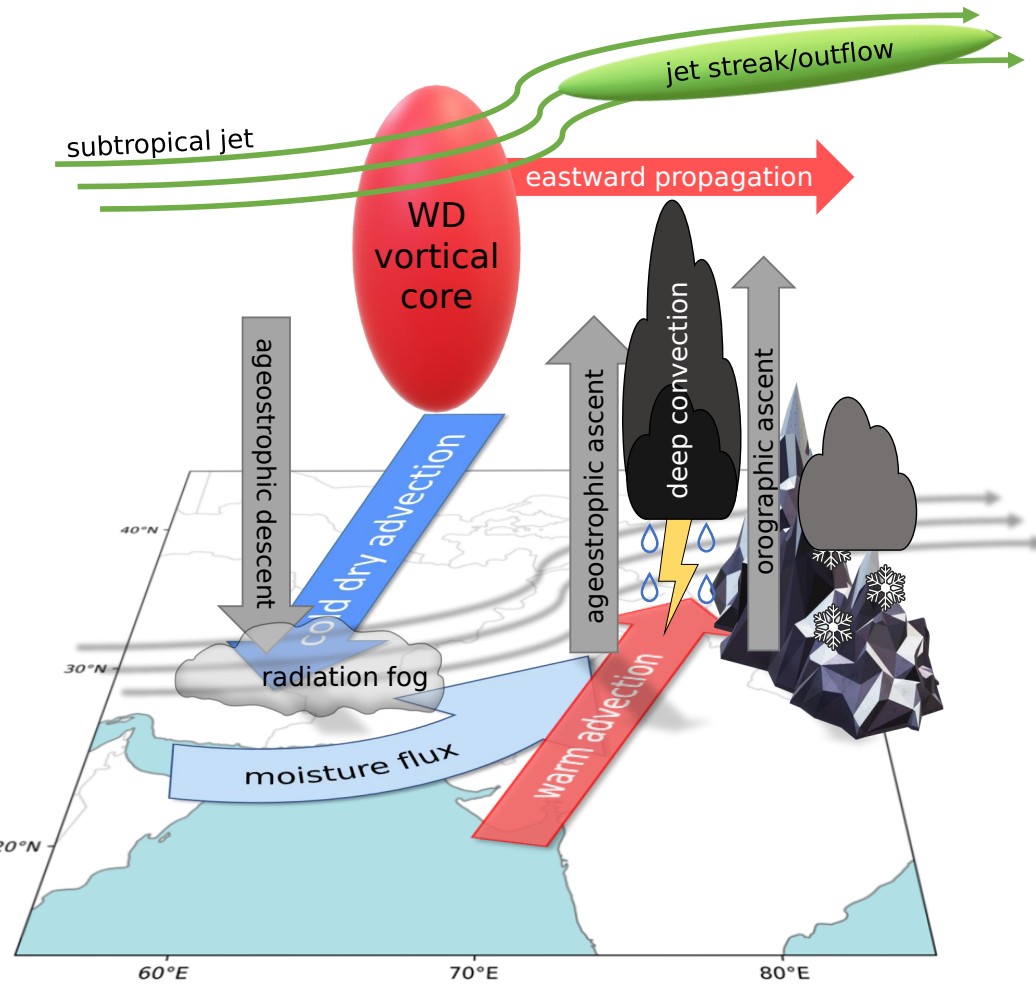

**Figure 3.** Schematic outlining some of the key meteorological features of western disturbances as they approach the Western Himalaya. The WD is characterised by a high vorticity region (red) propagating eastward along the subtropical jet (green). It is associated with a transverse ageostrophic circulation, including broad descent to the west and a broad ascent to the east. Within the latter area, strong orographic ascent occurs along the mountain barrier of the Himalayas. Its outflow forms a jet streak downstream. The cyclonic circulation often also extends to the surface. Cold and dry advection occurs to the west, which with the large-scale descent produces ideal conditions for radiation fog. To the east, warm southerly advection brings moisture from the Arabian Sea. Large-scale ascent, orographic forcing, moisture supply, and latent instability promote deep convection resulting in heavy rain over the lowlands and snow over the foothills (in winter) and mountains.

and Chevuturi, 2014b), avalanches (Ganju and Dimri, 2004), and where baroclinic interaction with the orography can result in heavy precipitation (Ramaswamy, 1956; Singh and Agnihotri, 1977).

Even with scarce *in situ* measurements, diligent use of observations – particularly upper-air soundings and satellite data – means that synoptic-scale WD structure has been well understood for some time. The presence of an associated upper-level





trough was first identified by Singh (1963). Chitlangia (1976) used a composite of six WDs to show that above the surface low, WDs were associated with vorticity and ascent maxima that extended through much of the troposphere, capped by a region of upper-level divergence. Similar results were reported in subsequent case studies by Singh and Kumar (1977, and others), who further identified that frontal regions were sometimes associated with the passage of a WD, particularly in the upper levels. Furthermore, Rao and Rao (1971) observed that WDs were embedded in a region of strong vertical wind shear, from which they deduce their predominantly baroclinic nature.

WD structure was progressively refined over the following decades as satellite data, reanalyses, and higher resolution models became increasingly available. The salient features of WDs, summarised by Dimri et al. (2015) and Dimri and Chevuturi (2016), are: a strong upper-level trough (i.e., a geopotential height minimum) embedded in the subtropical westerly jet, which is usually associated with a weaker surface low; sometimes with frontal features, which can occur at any level, but which tend to weaken as they approach the Himalayas; broad ascent ahead of the system centres – often associated with heavy precipitation – and broad descent behind them; and strong vorticity at their centre, peaking in the upper levels. Key characteristics of the WD, along with its environment, are summarised in Fig. 3.

## 1.2 Why are western disturbances important?

WDs are the dominant synoptic winter weather pattern over the Western Himalaya and surrounding region, and are associated with a wide range of impacts – including cloudiness, rainfall, snowfall, fog, coldwaves, and avalanches. Understanding their evolution is therefore key across all timescales, from weather, to seasonal, to climate. Growing evidence also suggests that WDs are impacting weather outside the winter, leading to extreme preciptiation events, which we discuss in Sec. 3.5.

Winter precipitation brought by WDs is important for both water security (Benn and Owen, 1998; Archer and Fowler, 2004; Thayyen and Gergan, 2010) and agriculture (Yadav et al., 2012). Reservoirs across the region are filled by meltwater from the winter snowfall during the subsequent spring melt season, and supply the spring rabi crops (e.g. barley and wheat), which are important components of the food security of the subcontinent. This meltwater is key to mitigate the seasonal drought that occurs in May and June before the arrival of the summer monsoon. Late WDs occurring during the spring or early monsoon can also impact kharif crops (e.g., rice and maize). As a result, WDs both directly and indirectly impact the economies of India, Pakistan and neighboring countries. On climate timescales, winter precipitation brought by WDs is a vital component of glacier mass balance in a region known for its anomalous glacial advance (the Karakoram anomaly; see Sec. 7.2.2).

## 1.3 Why is an updated review needed?

A comprehensive review of WD literature was carried out by Dimri et al. (2015), which was followed by a short book (Dimri and Chevuturi, 2016). Research since then largely falls into four categories.

Firstly, recent studies have made use of increasingly inexpensive high-resolution models, both for regional climate modelling and numerical weather prediction. WD research especially benefits from these developments, not only because the models are now often convection-resolving – and much of the heavy precipitation associated with WDs is convective – but also because of the improved representation of orography and the land surface with which WD-driven flows have a complex relationship (see



Sec. 2.3.2 and 5.2.1). The large number of high-resolution experiments also serves as a primitive large ensemble – as these models are able to capture processes more faithfully, experiments can more easily establish which physics schemes, forcings, and configurations are most important, collectively driving down the model uncertainty from which earlier studies suffered.

Secondly, the development of automated tracking techniques have allowed recent studies to explore the climatology of WD structure and behaviour in reanalyses and climate model outputs, revealing much about their variability and response to climate

change (see Sec. 2.3.3, 7.2 and 7.3). These developments have helped to link the physical processes of individual storms to the large-scale weather in which they are embedded, and to understand directly the influence of climate change on the statistical behaviour of WDs.

Thirdly, recent studies have made use of the increasing availability of isotope analysis, which, when coupled with trajectory analysis and composite moisture flux analysis, can provide valuable insights into the moisture sources of WD precipitation (see

Sec. 2.4). This research has led to findings that WDs draw most of their moisture from the Arabian Sea, rather the Mediterranean as often previously thought.

Fourthly, significant advances in paleoclimate proxy techniques and associated modelling have allowed studies to better quantify the behaviour of WDs and winter precipitation during ancient climatic epochs (see Sec. 7.1). These studies have revealed periods of increased and suppressed WD activity, primarily through speleothem and sediment analysis, showing

significant variability on centennial timescales.

This review is laid out in a thematic manner that follows previous reviews, covering structure and dynamics (Sec. 2), precipitation and impacts (Sec. 3), large-scale variability (Sec. 4), model evaluation and verification (Sec. 5), extended range and seasonal forecasting (Sec. 6), and finally climate change (Sec. 7). The section on climate change is further subdivided into paleoclimate (Sec. 7.1), observed responses to the instrumental record (Sec. 7.2) and the projected response to future climate

change (Sec. 7.3). Each section starts with a summary of older literature, to orientate the reader and to provide context for the newer research. Finally, in Sec. 8, we outline the important unresolved questions and open research topics that remain for WDs.

We have made a reasonable attempt to include all published literature on WDs, except where manuscripts were not accessible, either through cost (e.g., being in a book), or language. In some parts of this review, we have included additional papers

that cover winter precipitation over the relevant region, as this can be a useful proxy for WD frequency and such papers can add useful evidence to the discussion.

## 2  Structure, dynamics, thermodynamics, and life cycle

### 2.1  Summary of earlier research

As discussed in Sec. 1.1, the main structural features of WDs – that they are upper-level troughs (i.e. potential vorticity anoma-

lies) in the subtropical jet which often extend to the surface, sometimes have associated frontal features, and the large-scale ascent they produce to the east and descent to the west – have been established through multiple case studies collectively using surface observations, upper air soundings and satellite data. They share some structural and dynamical similarities with extra-





tropical cyclones, but differ in scale and intensification mechanisms (Riehl, 1962). Like extratropical cyclones, the available potential energy for WDs comes from the meridional temperature gradient and resulting vertical wind shear (Dimri, 2004).

As deep troughs, WDs are associated with high vorticity in the mid-troposphere which can be further enhanced through orographic interaction (Hara et al., 2004; Dimri and Chevuturi, 2014b). The high levels of vorticity present in WDs, and the lack of other westerly sources in the mid- and upper-troposphere over South Asia makes them ideal candidates for automated tracking algorithms, from which much-needed climatologies can be produced; however, no such work was undertaken prior to 2015 and so there has been a reliance on case studies Hara et al. (2004); Dimri and Chevuturi (2014b). In addition, as

even fairly primitive reanalyses have been shown to simulate WD structure well (Mohanty et al., 1998, 1999), despite scarce observations available for assimilation, they can be combined with track databases to improve understanding of WD structure and variability. This has been one of the key foci of recent WD research.

WD precipitation is characterised by rainfall at lower elevations and snowfall at higher elevations (Barros et al., 2004), which have significant impacts on both lowland agriculture (Yadav et al., 2012) and water storage at higher altitudes (Benn and Owen,

1998; Archer and Fowler, 2004; Bolch et al., 2012). The role of orographic interactions in conjunction with the structure and dynamics of WDs are key to understanding the mechanisms of precipitation formation over the western Himalayas. In addition, large-scale baroclinic conditions and localised orographic forcing also impact the passage of WDs. The feedback between moist processes and orography constrains both the amount and type of precipitation (Smith, 1979; Houze, 2012).

Moisture is a key component of WDs, as the winter atmosphere over the Western Himalaya is typically quite dry. While

classical research argued that WDs brought moisture with them from the Atlantic Ocean or Mediterranean Sea (Mull and Desai, 1947; Pisharoty and Desai, 1956; Rao and Srinivasan, 1969), more recent analyses of moisture flux cast doubt on this idea (Dimri and Mohanty, 1999; Raju et al., 2011), implying a more local source such as the Arabian Sea. As we will see, recent developments in isotope analysis, more systematic composite analyses, and the novel development of moisture backtrajectory analysis have helped to resolve this disagreement on WD moisture sources.

## 2.2 Detection and tracking

Detection of WDs depends on the characteristics of the WD being detected. Some studies have used a bottom-up approach, starting from WD impacts, and investigating the WD characteristics that drive these impacts. Before 2015, Dimri (2013) used bandpass-filtered precipitation and OLR to build a composite analysis that first defined the characteristics and atmospheric circulation of a mean WD that lead to precipitation. More recently, Baudouin et al. (2021) used more advanced statistical tools

(principal component regression) to quantify the link between precipitation and specific atmospheric patterns. They showed, in particular, that 68% of precipitation variability in the upper Indus Basin can be explained by the presence of an upper-tropospheric geopotential low (i.e., a WD), and that this number increased to 88% when taking into consideration the mid- to lower-tropospheric structure of the WD. Using composite analyses and clustering, they further showed how different WD characteristics lead to different impacts in the study area (see Sec. 3.4). While these studies do not directly detect specific WD

events, they help to define the characteristics of WDs that lead to precipitation.





In the last decade, numerous studies have instead used top-down approaches, and more specifically tracking algorithms. Automated tracking algorithms allow the relatively quick production of multidecadal catalogues that can then be leveraged to compute basic statistics such as track frequency and density, as well as more advanced Lagrangian statistics, such as the mean and variability of storm-centred structure. Feature tracking is the only way to get storm-centred composites, which reduce large-scale feature smoothing due to differences in WD location, although smaller scale features – such as fronts – typically disappear from such composites. However, tracking algorithms often simplify the analysis of WDs to just a few parameters: location/tracks and intensity of the centre, and so they do not define the entire spectrum of WD characteristics, although these can be largely reconstructed with further analysis.

Tracking has been done successfully for, e.g. extratropical cyclones (Dacre et al., 2012) and monsoon depressions (Hurley and Boos, 2015), but only recently have authors started to track WDs in reanalysis data. The first systematic attempt came from Cannon et al. (2016), whose methodology loosely followed that used in thunderstorm tracking (e.g. Dixon and Wiener, 1993; Carvalho and Jones, 2001). Using reanalysis data from 1979–2013, they identified regions of negative 500-hPa geopotential height anomaly, using a thresholding approach to allow detection of multiple centres within a large region of negative anomalies. Regions smaller than $5°$ in length were then removed and the remaining regions linked using a spatial correlation (overlap) technique. They identified 600 WDs in total ($\sim$18 per year), less frequent than reported in earlier studies on account of their strict approach.

Hunt et al. (2018b) used a different approach. They computed the T63 spectral truncation of the relative vorticity averaged between 450 and 300 hPa, using ERA-Interim reanalysis data from 1979–2015. Using this field to identify local maxima, regions of positive vorticity were then linked using a $kd$-tree nearest neighbour approach. Tracks that did not pass over South Asia (defined as 20–36.5°N, 60–80°E), were shorter than 48 hours in duration, or that did not propagate eastwards, were removed. They identified 3090 WDs in total ($\sim$86 per year). This is more frequent than reported in earlier studies, as this approach detects many very weak systems.

Later work, described in Nischal et al. (2022) extended this vorticity-based tracking approach to the newer ERA5 and IMDAA reanalyses, returning similar frequencies. Qiu et al. (2022) used a similar method in their study of WDs over southern China: identifying contours of $3\times10^{-5}$ s$^{-1}$ in 450-300 hPa relative vorticity, then linking candidates if these areas overlapped more than 60% in successive 6-hr reanalysis timesteps and the areas share similar physical properties, and rejecting tracks shorter than 24 hours. They identified 2594 WDs passing through their study region (20–40°N, 80–105°E) in December to February 1979–2017, a similar frequency to that reported by Hunt et al. (2018b).

To obtain WD tracks for their analysis of the Karakoram Anomaly, Javed et al. (2022) applied the TRACK algorithm (Hodges, 1995, 1999) to T63 relative vorticity averaged between 400 and 300 hPa in three reanalyses, finding and linking maxima in regions exceeding $1\times10^{-5}$ s$^{-1}$. They then rejected tracks shorter than 24 hours or 1000 km, and whose genesis was not in their genesis study region (20–50°N, 20°W–60°E). See Sec. 7.2.2 for a full discussion of WDs an the Karakoram Anomaly.

Other studies have used simpler, indirect techniques to approximate WD frequency and related statistics (e.g., intensity), either in reanalyses or climate model output. A list summarising these is given below.





- 850–200 hPa thickness, averaged over 60–80°E, 25–40°N (Midhuna et al., 2020).

- EOFs of high-pass filtered daily 500-hPa geopotential height, computed over 50–100°E, 25–40°N. The first two of these EOFs describe a wavetrain strongly resembling WD propagation and explain more than 50% of the variance (Madhura et al., 2015).

- Spectral power of 200-hPa geopotential height in the 5–15 day band, motivated by linking upper-level wavetrains with heavy winter precipitation over Pakistan and north India (Cannon et al., 2015). This is averaged over several different regions.

- Variance of 4–15 day bandpassed 200-hPa geopotential height, also computed over various regions (Krishnan et al., 2018).

- $k$-means clustering on low-level winds (10 m, 925 hPa, 850 hPa) and mean sea level pressure over India. This gives thirty weather distinct weather regimes, four of which are associated with WD-like activity (Neal et al., 2020). A similar approach, applied just to the Karakoram and Western Himalaya, was used in Riley et al. (2021).

Midhuna et al. (2020) showed that their index had a correlation coefficient of only 0.18 with observed annual WDs frequencies, indicating that proxy techniques like these are not necessarily reliable. In the case of Midhuna et al. (2020), this is because they characterise WDs by a cold troposphere; while this may work well for individual events, seasonal mean tropospheric temperature is more strongly a function of jet position.

Top-down and bottom-up approaches are complementary, their combination helps to build new understanding of WD processes.

## 2.3 The lifecycle of WDs

Tracking algorithms provide an objective way to build up large catalogues of WD tracks. These, in turn, can be analysed to provide new information about the climatology (i.e. average behaviour) and variability of WDs. A summary of the typical pathway in which WDs interact with their environment over South Asia is given in Fig. 4, from Baudouin et al. (2021). WDs typically pass along the northern edge of the subtropical jet, and upon arriving over the Western Himalaya, induce moist southwesterlies in their front sector – resulting in cross-barrier flow and precipitation – and cold northerlies in their rear sector – resulting in coldwaves and fog.

### 2.3.1 Cyclogenesis

Although the official IMD definition of WDs states that they 'originate over the Mediterranean Sea, Caspian Sea and Black Sea' [1], even very early studies expanded on this, stating that WD cyclogenesis could occur in maritime regions spanning from the North Atlantic to the Persian Gulf (Pisharoty and Desai, 1956).

---

[1] https://www.imdpune.gov.in/Reports/glossary.pdf



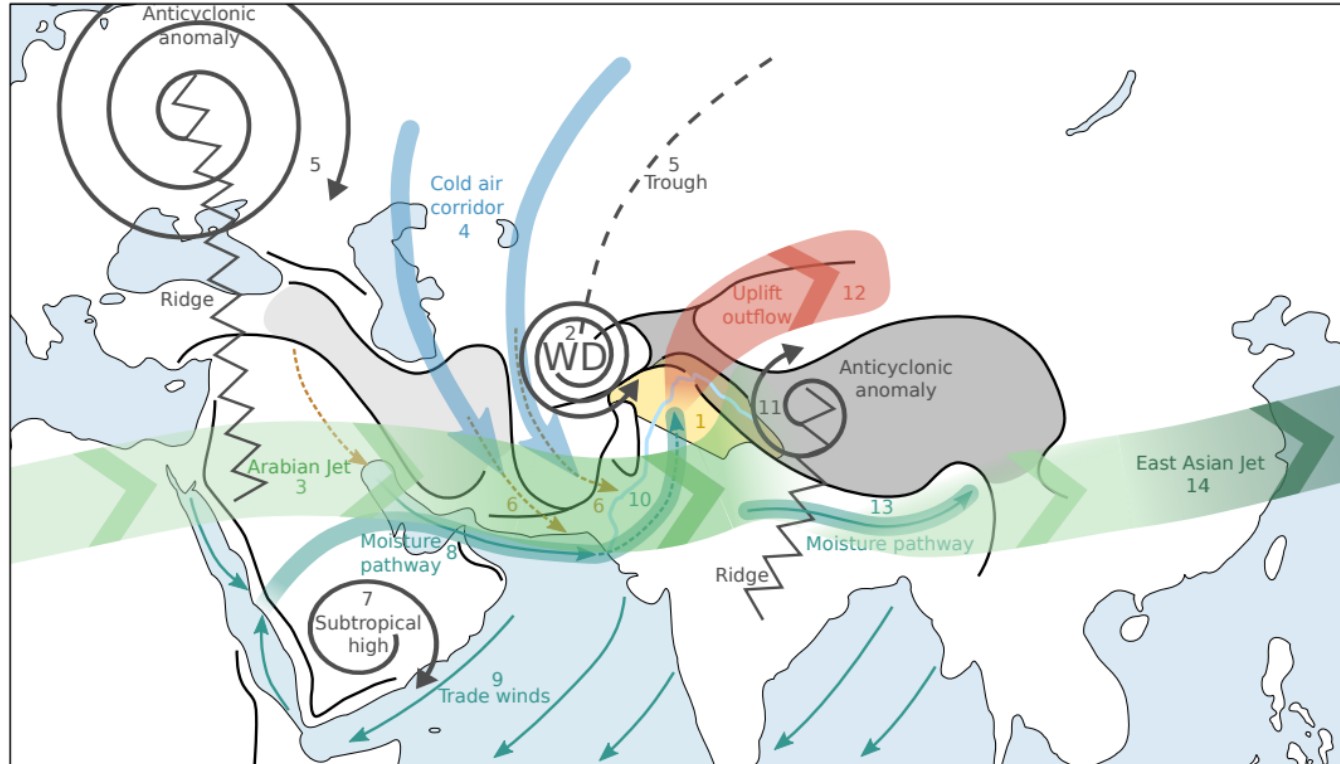

**Figure 4.** From Baudouin et al. (2021). Sketch summarising the atmospheric circulation when a WD interacts with the relief to produce precipitation in the Upper Indus Basin (in yellow). The Indus River in pale blue. The black lines indicate major mountain ranges, and the grey-shaded areas high plateaus (The Tibetan Plateau and the Iranian Plateau). The spirals represent tropospheric geopotential anomalies or centres of actions. The thin green arrows are mean lower troposphere moisture transport. A thicker arrow width indicates higher altitude transport, and a dotted line indicates transient transport due to the WD. The brown dotted lines indicate advection of dry air by the WD near the surface. The blue arrows denote troposphere-wide cold air advection. The thick green arrows represent the upper troposphere jets, with darker colour indicating stronger winds. The red arrow relates to the warm upper troposphere outflow. The numbers are in-text references to Baudouin et al. (2021).

However, such studies are built on collections of case studies, and are thus not necessarily representative of the whole population of WDs, as they tend to focus on high-impact events – usually driven either by intense WDs or those passing over a certain region. Objectively-derived track databases, such as those of Cannon et al. (2015), Hunt et al. (2018b), and Javed et al. (2023), all agree that there is very little geographical preference for WD cyclogenesis except for the following, which are summarised in Fig. 5.

1. All WDs originate from regions to the west of the Western Himalaya, or occasionally spin up *in situ*. This occurs largely because most WDs are advected along the subtropical jet, but also because any systems propagating eastward or southward into the region would be blocked by the Tibetan Plateau.





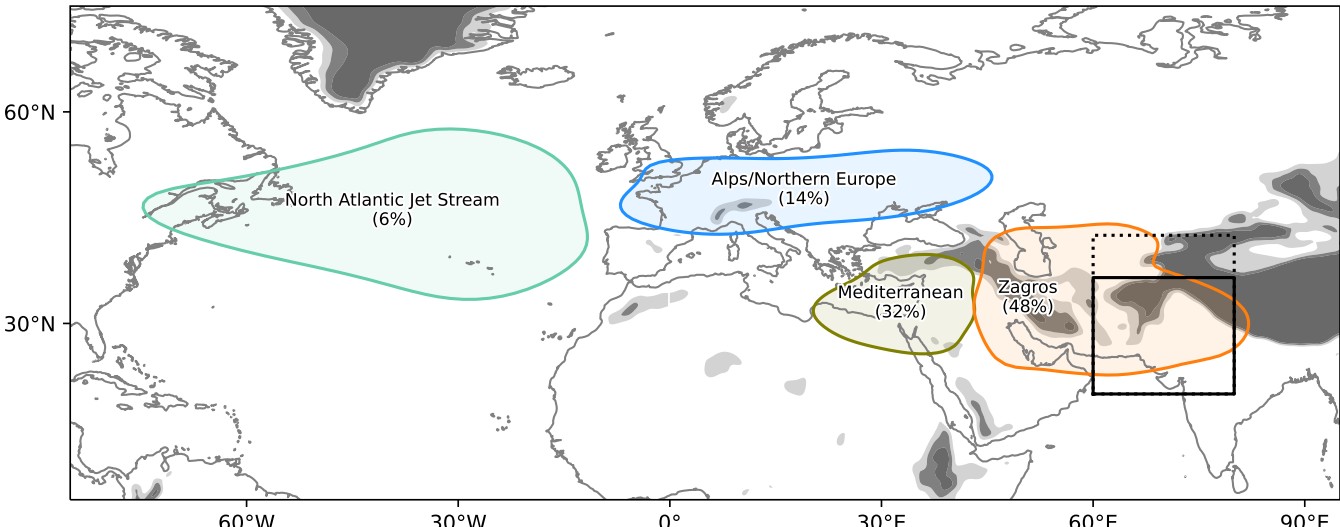

**Figure 5.** Regions of climatological WD cyclogenesis, computed using the WD track dataset described in Hunt (2024). Genesis points from all ∼6000 winter WDs in the dataset are clustered into four groups using a simple *k*-means approach. A Gaussian kernel density estimate is then applied to each cluster, with the 90% contour plotted. Regions are labelled after upstream mountain ranges or water sources, with the figures in parentheses indicating the fraction of the total WD population with geneses in that region. Filled grey contours represent smoothed orography at heights of 1, 1.5, and 2 km. The black box indicates the commonly used WD track capture region (60–80°E, 20–36.5°N). The dotted box shows the extension to 42.5° which has also been used in some recent studies.

2. There is no obvious preference for cyclogenesis over bodies of water, with only around a third of WDs spinning up over the Mediterranean (and a much smaller fraction over the Caspian and Arabian Seas).

3. The vast majority of WDs spin up quite close to the Western Himalaya, with nearly half of all cyclogenesis occurring to the east of the Zagros mountain range in Iran, between one and two WD wavelengths upstream of the Western Himalaya.

4. There is a preference for cyclogenesis in regions of dynamical instability; typically downstream from mountain ranges, but also within the North Atlantic jet stream.

Beyond this, several important questions remain for future research about the the relationship between the WD cyclogenesis regions and downstream characteristics and impacts, as well as whether there are different WD cyclogenesis mechanisms.

### 2.3.2 Intensification and moist thermodynamics

There are three potential pathways through which WDs can intensify. On the largest scale, those in the subtropical jet experience strong vertical wind shear, and hence can grow through dynamical baroclinic instability (Grotjahn, 1993; Lee and Kim, 2003). As they approach the Himalaya, the orography offers a new source of baroclinic instability, subject to the jet-orography configuration (Roads, 1980; Reinhold, 1990; Brayshaw et al., 2009). Finally, as with all rotating precipitating systems, precip-



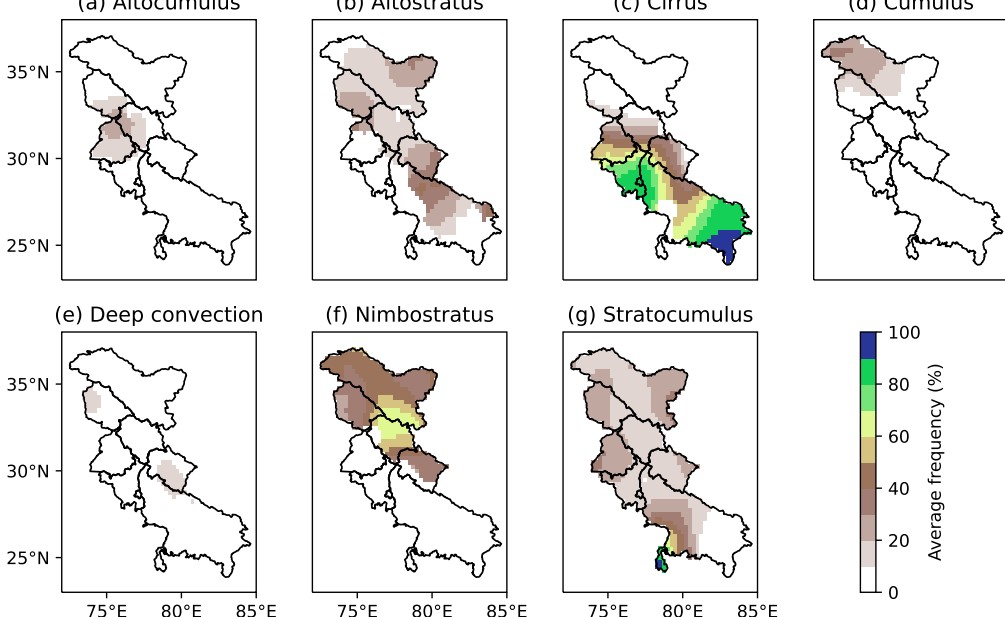

**Figure 6.** Adapted from Kumar et al. (2019a). Mean frequency of seven core cloud types (stratocumulus, cumulus, altocumulus, altostratus, nimbostratus, cirrus, and deep convective) over the northern states of India, between 2007 and 2016. Computed for the winter months (December to March) using CloudSAT data, evaluated from the surface to 18 km.

itation – whether free or forced – can lead to intensification through vortex stretching, driven by diabatic heating and updrafts (e.g. Tory and Frank, 2010). This effect is most pronounced for convective precipitation, which often occurs in WDs. As we will see, in the context of WDs, these mechanisms have been investigated and quantified to varying degrees.

WDs intensify as they reach the Western Himalaya, reaching maximum intensity – both in vorticity and updraft strength – about a day before their centre passes over the orography (Hunt et al., 2018b). Once they have passed the orography (i.e. the Western Himalayas), they quickly dissipate and are thus less commonly found over northeastern India or Bangladesh, with comparatively little WD-associated precipitation falling over the Central and Eastern Himalayas (Dimri et al., 2023).

The processes linking WD dynamics to ascent are neither linear nor readily separable from the role of orography. To first order, WD-driven ascent is the quasigeostrophic response to a PV anomaly in strongly-sheared flow, as demonstrated in both case studies (Kumar et al., 2017) and larger composite samples (Sankar and Babu, 2020, 2021). The latter studies demonstrated some interesting corollaries: updrafts can extend from near the surface to as high as 200 hPa (Sankar and Babu, 2020), and that there is no significant relationship between CAPE and WD precipitation (Sankar and Babu, 2021). However, Para et al. (2020) used a case study of extreme precipitation over Jammu and Kashmir in 2017 to show that the broad quasigeostrophic ascent can be coupled with convection, as latent heat release increases the strength of the upper-level PV anomaly. They also suggested that cold-air intrusions from the midlatitudes destabilise the mid-troposphere, leading to increased baroclinic growth of the WD, later corroborated by Hasan and Pattnaik (2024).



Indeed, it has been through precipitation that most recent studies have approached the topic of WD intensification. Two key ingredients are required for WDs to precipitate: moisture influx and updrafts to raise that moisture aloft. We will discuss potential moisture sources in Sec. 2.4 but review here recent research on the processes that lead to WD precipitation across the
Western Himalaya and their influence on WD dynamics.

Baudouin et al. (2021) approached this problem using a novel, generalised framework. They decomposed the moisture flux associated with heavy precipitation over the upper Indus basin and showed that passing WDs interact baroclinically with – and strengthen – the low-level moist flow from the Arabian Sea moving towards and over the orography. They linked the strength of this interaction – and hence the magnitude of the precipitation – to the intensity and thermal structure of the passing
WD, substantiating the work of Hunt et al. (2018b), who established a statistical link between WD intensity and associated precipitation. Baudouin et al. (2021) further showed that if a WD is too far north or too weak, the baroclinic strengthening of the moisture pathway is reduced, limiting precipitation. A complementary mechanism was proposed by Laskar et al. (2015), who examined two case studies of intense WDs. They argued that moisture influx could also be caused by an extension of the lower-level WD trough into the Arabian Sea, with anomalous southerly moisture transport occurring along its downshear
(i.e. eastern) edge. This may simply be a different interpretation of the atmospheric river framework above (see also end of Sec. 2.4), but both theories explain why the heaviest precipitation falls ahead (i.e. to the east) of WDs, and why precipitation is so sensitive to WD latitude. On one hand, the WD needs to extend far enough south to interact with the mean moisture flux over the Arabian Sea, and, on the other hand, it should be at the right latitudinal position to interact with the relief. Frontal regions, if they are still present in the WD by the time it reaches the Himalaya, may also play a role in uplifting moisture ahead
of the system.

Once moisture arrives over the continent, it must be lifted in order for heavy precipitation to occur. There are two available mechanisms for updrafts in the environment of a WD: ascent can be mechanically forced through interaction between low-level flow and the orography, or it can be quasigeostrophically forced through the dynamics of the WD itself (see Sec. 2.3.4). Indeed, these two mechanisms can interact constructively, strengthening the WD as it approaches the Western Himalaya. Little
work (limited to Dimri, 2004) has been done on explicitly quantifying the role of the Himalayan orography in producing WD precipitation, perhaps because underlying processes can be mostly understood through the interaction of moist flow with complex topography in other regions – e.g. the Andes (Insel et al., 2010) and the Rockies (Silverman et al., 2013). Baudouin et al. (2020b) showed that low-level southerly moisture transport explains about 80% of the variance of winter precipitation over the Himalayan foothills, strongly implying the importance of the forced uplift of cross barrier (i.e. perpendicular to the
orography) flow. They argued that the vertical structure of the moisture flux was important in such cases: moisture transport at 700 hPa had the strongest control on variability, and transport at altitudes greater than 500 hPa typically passed over the orography unimpeded. Battula et al. (2022) corroborated this transport argument in their climatological study, further arguing that strong WDs are the only weather system capable of pushing moisture to high altitudes in the Western Himalaya. This transport can result in substantial diabatic heating and subsequent intensification of the WD (Hasan and Pattnaik, 2024).
As we have seen, WDs can intensify via baroclinic feedback between their primary circulation and induced low-level souther-lies undergoing forced orographic ascent ahead of the system. Baudouin et al. (2021) showed that this interaction could dramat-





ically slow the passage of a WD by enhancing the divergence maximum downstream of the system. This is supported by Chand and Singh (2015), who found, when using satellite data to analyse a group of 10 WDs, that WD propagation speeds varied between 280 and 670 km day$^{-1}$ and had a negative correlation with cloud-top height downstream, implying that WDs associated

with stronger convection tended to propagate more quickly. They found that convection was typically at a maximum 1200 km ahead of the WDs, and associated with cloud-top temperatures between $-50$ and $-60°$C. Sankar et al. (2021) analysed the cloud structure of a WD that passed over north India between 23 and 27 Jan 2017, using geostationary IR and microwave data. They found the deepest convection occurred 200–400 km east of the mid-level trough, typically over orography, and was associated with cloud-top temperatures as low as $-60°$C. In summary, the deep ascent ahead of WDs primarily occurs due to

mass continuity responding to the downstream divergence, supported by quasigeostrophic differential vorticity advection and mechanical uplift of induced lower-level southerlies as they interact with the orography.

Despite this association with deep convection, WDs are usually more easily recognised by their stratiform footprint. This was highlighted recently by Kumar et al. (2019b) who showed using the DARDAR dataset (Ceccaldi et al., 2013, spanning 2007–2016) that during the passage of WDs over North India, the most common cloud types are nimbostratus (33%) and

cirrus (21%), with deep convection being comparatively uncommon (2.3%) even though it is associated with the most severe weather impacts. A companion study (Kumar et al., 2019a) found that about 25% of all clouds over North India occur during the passage of WDs. These results are summarised in Figure 6.

In summary, WDs are associated with strong lower-mid tropospheric southerly moisture flux from the Arabian Sea. As this moisture reaches the orography, it ascends through a non-linear combination of mechanical orographic forcing, quasi-

geostrophic ascent, and convection, primarily responding to the central PV anomaly of the WD. This is borne out in the statistical relationship between stronger WDs (i.e., those associated with stronger upper-tropospheric vorticity) and more intense precipitation (Cannon et al., 2016; Hunt et al., 2018b). The relative importance of each of the known intensification pathways – large-scale baroclinic, orographic-baroclinic, and convective vortex stretching (and there may be others) – to the growth and maintenance of WDs remains an open question.

### 365 2.3.3 Track density

Although the regions impacted by WDs have been known for some time, track databases allow a more rigorous quantification of their frequency and track density, as in Fig. 7, which we use to summarise the results of those track databases outlined above.

The highest track density is around the Western Himalaya, Karakoram, and Hindu Kush. WDs (or their unstable trough precursors) are funnelled into this region by the subtropical jet, whose mean location is very slightly to the south of the mountain

ranges. There, the systems are amplified by local (i.e. orographic) instability and moist processes, resulting in increased track density as existing WDs strengthen and pre-existing troughs in the subtropical jet intensify into WDs (Baudouin et al., 2021).

There is also a branch of tracks that are deflected to the north of the Tibetan Plateau, with the split happening at approximately the location of the Pamir Mountains. Little is known about this split, except, as we will discuss in Sec. 6, that it is very sensitive to the latitude of the subtropical jet and can have significant implications for forecasting WD impacts.



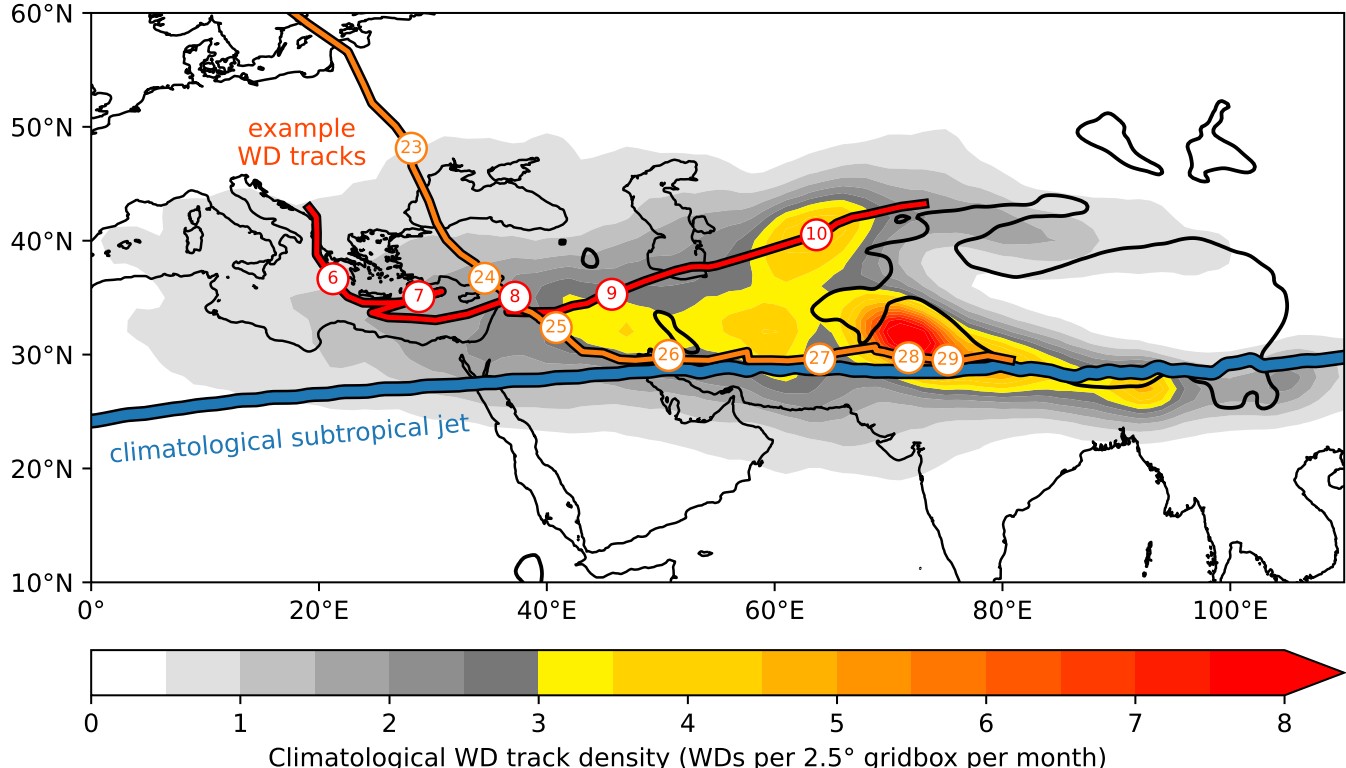

**Figure 7.** Following Hunt (2024). Filled contours show climatological track density of winter (Dec–Mar) WDs, computed from 1950–2022 using a track database deriving from ERA5. The thick blue line shows the mean location of the subtropical jet for the same season. The red and orange lines denote tracks of two real WDs that occurred in January 2020. Circular markers denote the location of that WD at 0000Z on the given day of the month. The thick black line denotes a 2-km contour of smoothed orography.

Tracks from two individual, intense WDs, occurring in January 2020, are also plotted in Fig. 7 and demonstrate the significant variability that can be present in WDs occurring just a few weeks apart. The first WD spun up over the Mediterranean on Jan 5, initially moving slowly before passing rapidly to the north of the Himalayas. The second WD spun up over northern Europe on Jan 22, before migrating southward and then propagated rapidly towards and then over the Western Himalaya, where it resulted in heavy precipitation. This again highlights the very different backgrounds that WDs can have, and the forecasting challenges that result.

This difference demonstrates that not all WDs arise as instabilities on the subtropical jet – but also as extratropical cutoff lows that migrate equatorward (Wernli and Sprenger, 2007; Portmann et al., 2021). Cutoff lows arise when tongues of high PV upper-tropospheric air detach from the extratropical westerlies and they tend to move more slowly than WDs embedded in the subtropical jet, sometimes leading to severe weather (Kalshetti et al., 2022). They comprise about 10% of all WDs (Thomas et al., 2023), but this number is uncertain given difference in tracking methodologies. Little is known about the differences between cutoff low WDs and those arising within the subtropical jet itself.



### 2.3.4 Vertical structure

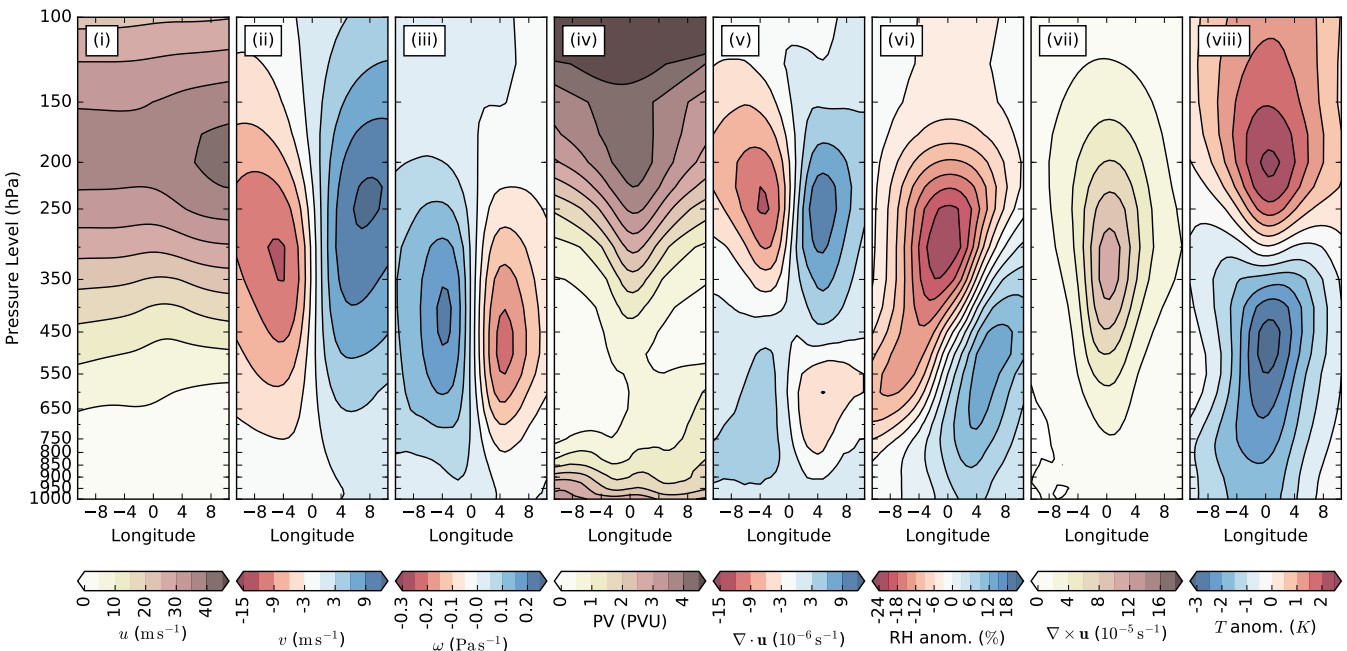

**Figure 8.** From Hunt et al. (2018b). Vertical system-centred composites of WD structure over northern India and Pakistan, taken from west to east through the centre, which is at $0°$ relative longitude. From left to right, the composite fields are: (i) zonal wind speed [m s$^{-1}$], (ii) meridional wind speed [m s$^{-1}$], (iii) vertical wind speed [Pa s$^{-1}$], (iv) potential vorticity [PVU; $10^{-6}$ K m$^2$ kg$^{-1}$ s$^{-1}$], (v) divergence [$10^{-6}$ s$^{-1}$], (vi) relative humidity [%], (vii) relative vorticity [$10^{-5}$ s$^{-1}$], and (viii) temperature anomaly [K]. The anomalies are computed against a 20-day mean. Longitude is defined relative to the centre of the WD composite.

Using either Lagrangian tracking techniques or an Eulerian proxy to develop catalogues of WD events, recent studies have employed compositing techniques to further understanding of WD structure and variability far beyond that provided by earlier
case studies. Compositing can provide useful insights into bulk structure, but with the caveat that finer structural features are smoothed out. Perhaps the most important example of this is the frontal regions of WDs: early case studies indicated that WDs arriving over Pakistan and north India were sometimes associated with frontal weather (Pisharoty and Desai, 1956; Singh and Agnihotri, 1977), but such discontinuities in meteorological fields were not present in the multivariable composite presented in Hunt et al. (2018b). This is likely because such features are often smeared out in composites as they are located
irregularly relative to the centre, and does not disprove their existence. Extratropical cyclone research has included front-centric compositing using theta/theta-e gradients or wind shifts (e.g. Dacre et al., 2012), but as yet no similar analysis has been done to establish if surface fronts are consistent features in WDs.

Regardless, much has been learned from applying compositing and clustering methods to WD catalogues, which we will now discuss. Hunt et al. (2018b) used the 3090 WDs in their database to produce multivariate composite cross sections, such



as the vertical-zonal cross sections in Fig. 8. Many of the features presented there are consistent with earlier case studies, and demonstrate the strongly baroclinic structure of WDs: a northwestward tilt with height, displacement of the temperature and geopotential minima, and a quadrupole in divergence, leading them to remark that WDs resemble 'immature extratropical cyclones', consistent with the description in earlier studies (Mull and Desai, 1947; Dimri and Chevuturi, 2014b). Qiu et al. (2022) reported an almost identical structure for a large composite of WDs that had propagated (or formed) further downstream

and impacted the southeastern edge of the Tibetan Plateau. Sankar and Babu (2020) found in a smaller composite of 10 WDs that warm-air advection increases with height in WDs, another feature that is present in extratropical cyclones. As in extratropical cyclones, the deep PV anomaly associated with WDs – which extends to the tropopause – can result in substantial transport of ozone from the stratosphere to the troposphere (Satheesh Chandran et al., 2022).

Contrary to previous studies (Pisharoty and Desai, 1956; Dimri et al., 2015), whose conceptual models placed the strongest
winds upstream of WDs, Hunt et al. (2018b) found that the subtropical jet accelerated downstream of the WDs. This was later corroborated by Baudouin et al. (2021), who argued that while there may be some cases of jet intensification upstream of (or 'behind') the WDs, the climatological picture was the reverse. However, they also showed that while the strongest anomalous upper-level winds occur to the east of the WD, the strongest absolute winds occur to its south. There is an important implication to this arrangement: intensifying the jet downstream means that WDs can trigger jet streaks, whose associated ageostrophic
circulation can result in regions of strong ascent. As they often behave like Rossby waves, WDs can also impart this ascent through eddy momentum flux from the extratropics (Kalshetti et al., 2022). We will discuss the impacts of such dynamics later.

In summary, WDs are typically associated with a broad vorticity maximum (or geopotential minimum) between 450 and 250 hPa. They are predominantly baroclinic in nature, leading to ascent ahead of the system and descent behind. In the lower levels they induce moist southerlies to the east of their centre and cold northerlies to the west. High relative humidity leads to
a deep cloud structure that results in anomalous heating aloft and anomalous cooling in the mid and lower troposphere.

## 2.4 Moisture sources

As we saw earlier, there has been substantial disagreement among previous studies as to whether WDs draw their moisture from the Mediterranean, the Arabian Sea, Caspian Sea or local evapotranspiration – each requiring very different synoptic to mesoscale dynamics to support their transport mechanism. Earlier studies also did not agree on how much WD precipita-
tion originated from local sources compared to the maritime sources above. In this subsection, we discuss how recent studies have attempted to resolve these uncertainties using new methods, including isotope analysis, moisture trajectories, and improved compositing techniques. Compositing provides a general view on moisture sources and pathways, but can not provide a quantification of the individual sources, for which isotope and back-trajectories are better suited.

All the isotope studies discussed in this section rely on a quantity called deuterium excess (or D-excess for short) which we
now briefly explain. D-excess ($d$) is given by:

$$d = \delta^2\mathrm{H} - 8\delta^{18}\mathrm{O} , \tag{1}$$



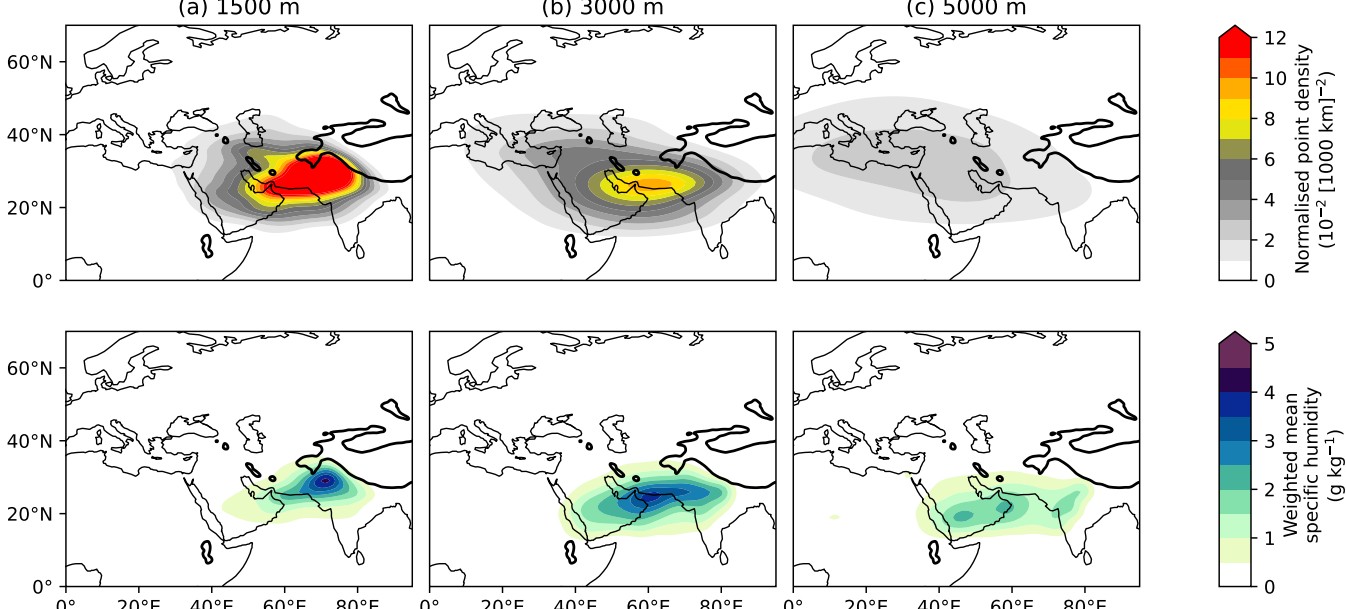

**Figure 9.** Following Hunt et al. (2018c). Potential WD moisture sources identified using trajectory analysis. A $3° \times 3°$ box is drawn around the heaviest precipitation associated with each of the 250 strongest WDs since 1980. From each, 100 uniformly-spaced ($10\times10$) particles are released at heights of (a) 1500 m (b) 3000 m and (c) 5000 m above the surface. Each particle is integrated backwards in time through 72 hours using wind fields from ERA-Interim. The top row shows the spatial probability distribution of such particles at a lead time of 72 hours. The bottom row shows the weighted mean specific humidity for particles at all lead times as a function of location, computed by multiplying the normalised point density in the row above by the parcel specific humidity in order to show the relative contributions of different moisture sources at each elevation.

where $\delta^2 H$ is the ratio of deuterium to hydrogen and similarly $delta^{18}O$ is the ratio of $\delta^{18}O$ to $\delta^{16}O$, both measured relative to a standard reference water. D-excess is higher in atmospheric water vapour that has evaporated from surfaces in colder or less humid climates, and thus precipitation arising from moisture originating from different basins has different D-excess values
(Pfahl and Sodemann, 2014), for example, the Mediterranean has D-excess values of around 22‰, much higher than the global average of 10‰(Gat and Carmi, 1970; Natali et al., 2022) and higher than the Arabian Sea (Jeelani et al., 2017; Jeelani and Deshpande, 2017). However, this is complicated by fractionation – wherein rain preferentially forms from low D-excess water – further increasing the D-excess in moisture in air parcels that have been transported a long distance, orographically lifted, or even locally recycled (Kong et al., 2013). Ideally, therefore, the results of isotope analysis over the western Himalaya should
be disambiguated with a complementary moisture trajectory or moisture flux analysis.

Thus, while recent studies all agree that winter precipitation over the western Himalaya and surrounding region have high D-excess, they disagree on the interpretation of that result. Some have attributed this to moisture coming from the Mediterranean, Black, or Caspian Seas, where naturally high D-excess is further increased through fractionation during the long journey to South Asia (Jeelani and Deshpande, 2017) and in some cases, this position has been further supported by basic trajectory



analysis (Saranya et al., 2018; Lone et al., 2019; Singh et al., 2019a; Lone et al., 2020, 2023). However, these studies are largely flawed, relying on only short sample periods or applying trajectory analysis either only to case studies or for whole seasons.

A few studies have attributed the high D-excess to local recycling, in other words, evaporation from the subcontinent itself (Jeelani et al., 2018; Oza et al., 2022; Joshi et al., 2023). Local moisture recycling as the dominant source is unlikely given the
volume required to feed the snowpacks and glaciers and support agriculture, and the frequency with which heavy precipitation events occur during the winter months. The most likely majority moisture source is the Arabian Sea, as it is the closest body of water to the Western Himalaya and passing WDs are often associated with southerly moisture flux. It is also the only body of water that can supply enough moisture for extreme precipitation events that are occasionally associated with WDs (Rao et al., 2016). This is supported by a few studies combining isotope analysis with back-trajectories (Jeelani et al., 2017; Dar et al.,
2021), both of which further argue that moisture from the Mediterranean must also play a significant role, even if it is not the majority moisture source.

One possible resolution to these disagreements is that winter precipitation moisture sources vary significantly as a function of location – for example, the orientation of some valleys may block southerly moisture flux from the Arabian Sea but allow northwesterly moisture flux from the Caspian Sea – and due to intraseasonal variability of large-scale conditions (Sec. 2.6) –
for example, WDs situated further north are less effective at extracting moisture from the Arabian Sea (Baudouin et al., 2021), a feature controlled by variations in latitudinal position of the subtropical jet.

By focusing solely on back-trajectory analysis, studies can be freed of this constraint, as trajectories can be initialised at any location and any altitude, rather than being confined to a particular observation point or catchment. Hunt et al. (2018c) used reanalysis data to compute Lagrangian back trajectories to determine the moisture sources of heavy winter precipitation events
over northern India and Pakistan. They also concluded that the Arabian Sea was the majority source, with smaller contributions from the Caspian and Mediterranean Seas. An updated version of their analysis is shown in Fig. 9, where particles are released at selected altitudes above heavy WD precipitation events and then integrated backwards in time. Particles in or slightly above the boundary layer (1500 m) largely originate from the Arabian Sea or surrounding region, and are responsible for the majority of the moisture. Particles released from the mid-troposphere (5000 m) largely originate from the Mediterranean but are only
responsible for a small fraction of the total moisture. The elevation-based dependence for source attribution may be the reason for disagreement in earlier studies. Boschi and Lucarini (2019) also used moisture back trajectories, computed using a modified version of FLEXPART (Stohl and James, 2004, 2005). Applied to 20 extreme winter precipitation events over northern Pakistan and Uttarakhand, they found that the main moisture sources were the Red, Mediterranean, and Caspian Seas, the Persian Gulf, and the Gulf of Aden.

Beyond isotope and trajectory methods, recent work by Baudouin et al. (2021) highlighted the utility of composite moisture flux analyses in investigating precipitation moisture sources. They identified a mean moisture pathway between the Red Sea and the North Arabian Sea, and that WDs were able to steer it (transiently) towards the western Himalaya and surrounding region. Results obtained using this method are very similar to those obtained from large-sample back-trajectory studies (e.g., Fig. 9). These pathways are analogous to the atmospheric rivers that are responsible for winter precipitation and flooding to the



west, in Iran (Dezfuli, 2020; Dezfuli et al., 2021; Esfandiari and Lashkari, 2021). Atmospheric rivers have also been explicitly linked to the majority of winter precipitation variability and extremes over the western and central Himalayas (Rao et al., 2016; Thapa et al., 2018; Lyngwa et al., 2023), where composite analysis shows circulation that strongly resembles that of a WD. The altitude of these moisture pathways also appears to be important, with the strongest moisture transport occur between 850 and 700 hPa, higher than usual in the tropics, with transport at 700 hPa explaining the largest fraction of precipitation variability

(Baudouin et al., 2020b).

Both isotope and trajectory methods are useful, but each have shortcomings that mean it is better to draw on results from both methods where possible. Trajectory methods give more precise results for moisture sources, and along-trajectory statistics like parcel humidity can be computed. However, large uncertainties can arise from the representation of orographic and boundary layer processes, both of which are crucial ingredients for WD precipitation. Indeed, the evaporative processes that increase

parcel humidity are subgrid processes that are not necessarily well simulated in reanalyses. Further to this, trajectory calculations can be computationally expensive as large ensembles are required to reduce uncertainty. Isotope methods can therefore provide more accurate estimates of moisture partitioning, since they do not depends on small-scale processes being adequately represented by a reanalysis model. However, long time series are required to ensure a representative contribution from all moisture sources. Results from studies that depend on a single year of data, e.g. Lone et al. (2020) and Dar et al. (2021), must

therefore be taken cautiously. Further work is needed with isotopic methods to better distinguish between Mediterranean and local recycling sources, both of which are associated with high D-excess (>20‰). Composite moisture flux analysis is more robust to orographic and subgrid processes, but by definition does not describe the entire distribution of possible sources, as back-trajectories can.

## 2.5 Seasonality

WDs are often thought of as occurring only during the winter months, but a long list of case studies (e.g. Yadav et al., 2017a; Gupta et al., 2023; Thapliyal and Singh, 2023a) and recorded interactions with the summer monsoon (e.g. Pisharoty and Desai, 1956; Chevuturi and Dimri, 2016; Hunt et al., 2021) show that they can occur at any time of year. In this section, we will quantify that seasonality.

Following Hunt (2024), the seasonal cycle of WDs is plotted in Fig. 10 as a function of their intensity. While WDs do

occur year round, they are nearly ten times more frequent during the winter months (Dec–Mar) than during the height of the summer monsoon (Jul–Aug). This cycle is controlled almost wholly by the seasonal migration of the subtropical jet (Datta and Gupta, 1967; Hunt et al., 2018b), which typically moves northwards away from the western Himalayas from April to October in response to increased radiative and latent heating of the boreal tropics (Sadler, 1975; Schiemann et al., 2009). Owing to the stability of the jet over the winter months, there is little variation in monthly average WD frequency between December and

March (16-18 per month) and this falls only slightly (13 per month) in the extended winter (November and April).

Intense WDs – which we have here taken to be those whose peak vorticity exceeds the 80th percentile of the whole population, in order to match frequency estimates of earlier authors (e.g. Rao and Srinivasan, 1969) – are much more limited to the winter months, appearing only very rarely between May and October. Such intense systems need not only a favourable jet



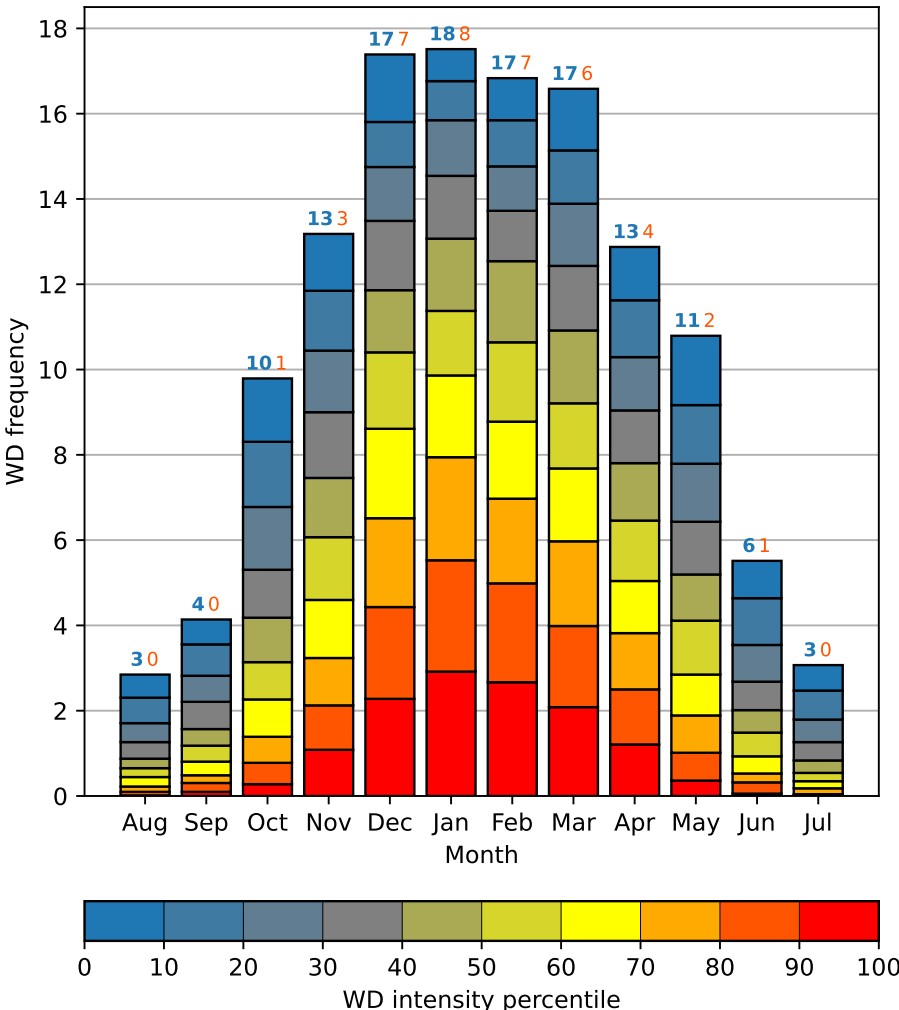

**Figure 10.** The seasonal cycle of WD frequency, computed between 1950 and 2022, following Hunt (2024). Mean WD frequency is given for each month, computed as the number of WDs entering the region (50–80°E, 20–40°N). These are further stratified by overall intensity percentile, where intensity is defined as the maximum value of 350 hPa $\zeta$ that a WD reaches during its lifetime. At the top of each bar, the total WD frequency (rounded to the nearest integer) is given in bold blue and the frequency of WDs exceeding the 80th intensity percentile is given in red. Computed using a WD track database derived from ERA5.

location, but also high upper-level baroclinic instability, which is uncommon outside of the winter months (Schiemann et al., 2009; Hunt et al., 2018a).

Hunt (2024) also showed that different regions affected by WDs can have different seasonal cycles. During December to March, they predominantly impact the Western Himalaya and Hindu Kush. As the jet starts to migrate northward in the spring, these regions are still impacted by WDs, but the northern mountains – the Karakoram and Pamirs – receive considerably more.





Perhaps the most important unanswered question is whether WDs occurring during, or near to, the monsoon have funda-
mentally different characteristics than typical winter WDs. It is certainly the case that even relatively weak WDs can wreak
havoc during the monsoon given the much more plentiful moisture supply, but it is not clear how, if at all, this rectifies onto
other characteristics, such as intensification rate and vertical structure. Interactions between WDs and the summer monsoon
are discussed in Sec. 3.5.

### 2.6 Variability

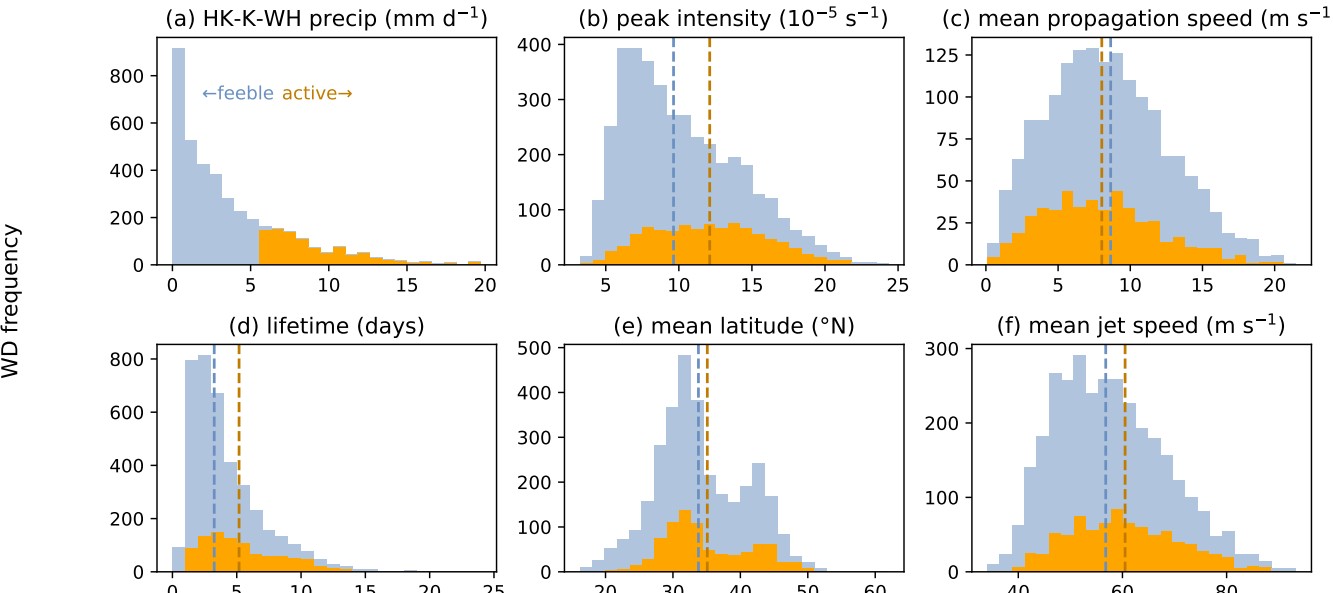

**Figure 11.** Stacked histograms of selected winter (Dec–Mar) WD characteristics, drawn from the Hunt (2024) database. (a) mean daily
precipitation averaged over a box including the Hindu Kush, Karakoram, and Western Himalaya [70–80°E, 30–40°N]; for each WD passing
between 60 and 80E, the maximum of these daily means is taken. (b) the peak intensity – measured as the highest value of 350-hPa relative
vorticity recorded at the WD centre as it passes between 60 and 80°E. (c) the mean zonal propagation speed of each WD between 60 and
80°E. (d) the total lifetime of the WD. (e) the mean latitude of the WD centre as it passes between 60 and 80°E. (f) the mean zonal wind
speed of the subtropical jet – computed along the jet axis between 50 and 80°E – while the WD is between 60 and 80°E. 'Active' WDs
(orange) are defined as being in the upper quartile of the precipitation values given in (a). On each subplot, the mean of the active and feeble
WD distributions are given by vertical dashed orange and blue lines respectively. All data from ERA5.

Having discussed the mean state of WDs, we turn now to their variability. As noted in the Introduction, many estimates of
WD characteristics come with wide error bars and thus merit more thorough investigation. Figure 11 summarises the variability
of six key WD characteristics, based on the database described in Hunt (2024). There is no formal definition of 'active' and
'feeble' WDs, despite common use – especially of the former – in literature. We therefore follow consensus usage, that active
WDs are responsible for heavy winter precipitation, and define them thus: consider mean daily preciptiation averaged over box





[70–80°E, 30–40°N] that covers the Karakoram, Hindu Kush, and Western Himalaya, then active WDs are those that cause that precipitation to exceed the 75th percentile of the distribution associated with all winter WDs. The area-averaged precipitation values have a lower bound of about 5 mm d$^{-1}$ for active WDs, broadly consistent with observations in Singh et al. (2019b). Although feeble WDs are only associated with light precipitation, they are sufficiently frequent that they comprise a large fraction of the total seasonal precipitation (Riley et al., 2021).

Peak WD intensity (Fig. 11(b)) varies considerably, with 350-hPa relative vorticity ranging between 5–20×10$^{-5}$ s$^{-1}$. Very few active WDs have intensities at the lower end of the distribution, and as such have a higher average peak intensity (12×10$^{-5}$ s$^{-1}$) than feeble WDs (10×10$^{-5}$ s$^{-1}$). As with other types of cyclone (e.g., hurricanes Emanuel, 2000), the intensity distribution is strongly right-skewed.

Mean WD propagation speed (Fig. 11(c)) also varies considerably, with systems ranging from being almost stationary up to
a top speed of about 20 m s$^{-1}$. The majority of WDs have speeds between 6–12 m s$^{-1}$, in agreement with the values given in early literature (Datta and Gupta, 1967; Rao and Srinivasan, 1969; Subbaramayya and Raju, 1982). The mean propagation speed of 8 m s$^{-1}$ (∼7° longitude per day) varies little between active and feeble WDs, with the latter moving slightly quicker, in agreement with Chand and Singh (2015) and Baudouin et al. (2021).

While the full distribution of WD lifetimes (Fig. 11(d)) ranges from 2–20 days, almost all systems live for between 3
and 10 days, in general agreement with earlier literature on the topic (Datta and Gupta, 1967; Rao and Srinivasan, 1969; Chattopadhyay, 1970; Subbaramayya and Raju, 1982). Active WDs are significantly longer-lived on average (5 days) than feeble WDs (3 days), giving them a longer time to intensify and draw more moisture into the region.

WD latitudes (Fig. 11(e)) have a bimodal distribution, as we saw in Sec. 2.3.3. They are generally concentrated around a mean value of 35°N but can vary from as far south as 20°N to as far north as 50°N. These latitudes constitute a wide range
of conditions, not only thermodynamically, but also in terms of orography and moisture availability, leading to varied WD behaviour and impacts (Baudouin et al., 2020b, 2021).

The mean jet speed (Fig. 11(f)) associated with active winter WDs is about 60 m s$^{-1}$, and about 4 m s$^{-1}$ slower for feeble WDs. The value varies from about 40 m s$^{-1}$ to rare instances of over 80 m s$^{-1}$. Notably, this is much faster than the WD propagation speeds, implying that WDs travel according to the group velocity, not the phase velocity, of waves within the
subtropical jet.

Very little research has covered the intraseasonal variability of WD frequency, as most focuses on their interannual variability (see Secs. 4 and 7.2). Therefore, quantifying and understanding variability on this timescale remains an important research topic. Two recent studies have published WD databases with monthly counts, from which we can estimate intraseasonal variability. Dimri et al. (2023), drawing their database from IMD weather reports, has a mean WD frequency of 18.3 per season
(Dec–Feb) over India, with an average monthly range of 2.7 – in other words, for a given season the average difference between the months with the highest and lowest WD frequencies. For the same metrics, the database from Hunt (2024) gives 52.9 WDs per season with an average monthly range of 4.7. This falls to 10.1 per season with an average monthly range of 3.3 if we use the same definition of 'active' WDs as above. Crucially, this suggests that the ranges of monthly WD frequency within a given season are not particularly sensitive to the choice of tracking or detection metric, even if the mean frequency is.





In summary, WDs are highly variable across all of their key characteristics (precipitation, intensity, speed, lifetime, latitude). This multidimensionality is seldom reflected in composite or case study analyses of WD behaviour and demonstrates a clear need for a formal classification scheme (such as the categories of North Atlantic hurricanes). Some attempts have been made, either clustering on dynamics and precipitation (Hunt et al., 2018b), or setting thresholds for upper-level wind speed (Javed et al., 2023), but neither has been adopted in wider literature or forecasting. The question thus remains on what the best way to

categorise WDs is – impacts (precipitation, location), or dynamics (vorticity, wind speed)?

## 3    Precipitation and other impacts

### 3.1    Summary of earlier research

Precipitation has long been identified as the main impact of WDs (Veeraraghavan and Nath, 1989; Yadav et al., 2012; Dimri, 2013). We have already discussed how recent advances help to better understand the relationship between WD characteristics

and precipitation (see Secs. 2.3.2 and 2.6). In this section, we more broadly explore the characterisation of WD impacts and how recent research has more robustly linked WDs to their associated hazards.

WDs have long been known as the link between large-scale forcing and winter precipitation over the Himalaya, with subseasonal and seasonal precipitation anomalies linked to changes in subtropical jet latitude (Schiemann et al., 2009) and upper-level kinetic energy flux convergence (Yang et al., 2002). However, not all WDs are associated with precipitation of severe intensity

(otherwise known as 'active' WDs). Dimri (2006) estimated that only a third of WDs are active (2-3 out of 6-7 per month during the winter); Mohanty et al. (1998) estimated an even smaller number (4-6 per season).

Part of the challenge lies in the accurate measurement of precipitation over the Western Himalaya. Precipitation here has high spatial variability because of the complex orography (Andermann et al., 2011), but measuring it is difficult because the gauge network is sparse due to the largely inaccessible terrain (Rajeevan et al., 2005; Kidd et al., 2017) and because gauges

have large uncertainty (and can even be buried) when measuring snow (Strangeways, 2010; Rasmussen et al., 2012; Cullen and Conway, 2015). Precipitation estimates from satellites also have high uncertainty not only because passive microwave retrievals perform poorly over snow cover (Ferraro et al., 1998; Tian et al., 2009), but also because the interaction between WDs and orography can produce very heavy local stratiform precipitation (Veeraraghavan and Nath, 1989), and so precipitation rate is not necessarily well correlated with cloud-top temperature. Reanalyses may prove to be useful, as long as they have sufficiently

high resolution to accurately capture the orographic convergence that intensifies WD precipitation (Dimri and Niyogi, 2012). WD precipitation is also dependent on underlying land use characteristics (Semwal and Dimri, 2012; Thomas et al., 2014).

We will also consider advances in the understanding of the role of WDs in water security, showing how recent studies have better quantified how WDs impact the Himalayan water cycle. They are an essential part of this water cycle, even outside winter, since spring snowmelt contributes up to 40% of Indus discharge and up to 20% of Ganges discharge (Ramasastri,

1999). Snowmelt also forms a small but important contribution to the discharge of the Brahmaputra river (Hill et al., 2020). This spring runoff is particularly important for agriculture in years with a delayed summer monsoon onset (Bamzai and Shukla,



1999; Liu and Yanai, 2002), but an abundance can lead to pre-monsoon flooding and landslides (Agrawal, 1999; Thayyen et al., 2013). In the winter months, snowfall from WDs also helps to replenish glaciers (Benn and Owen, 1998; Bolch et al., 2012).

When additional moisture is injected into the WD environment, through extension of the summer monsoon trough (Chevuturi
and Dimri, 2016), or through the presence of tropical depressions or tropical cyclones (Kalsi and Halder, 1992), serious flooding can occur. This was the cause of the 2013 Uttarakhand floods, the precursor meteorology of which has since been extensively studied (e.g. Singh and Chand, 2015; Joseph et al., 2015; Ranalkar et al., 2016; Vellore et al., 2016; Houze et al., 2017).

There are several different pathways through which WDs can interact with the monsoon or tropical cyclones. Kalsi and Halder (1992), expanding on Rao (1976), identified four ways in which WDs can interact with tropical air masses: (i) inten-
sification or initiation of tropical depressions; (ii) enhancing precipitation within existing tropical depressions; (iii) causing tropical depressions to recurve northward; and (iv) triggering breaks in the monsoon. Using satellite data, Kalsi and Halder (1992) examined seven case studies of WD-tropical depression interaction and found that these interactions could occur before, during, or after the monsoon, often leading to distortions in the tropical depression cloud field and heavy precipitation. They also found that immature depressions could be intensified by WDs, but could also be weakened by the vertical wind
shear associated with WDs. In one case, they observed that outflow from a tropical cyclone affected a WD. This research highlighted the need for further investigation of WD-tropical cyclone and WD-monsoon interactions, which would be important for understanding the full range of impacts these storms can have on the region.

Finally, we will look at how recent research has linked other natural hazards to WD activity. Historically, such hazards – including landslides and avalanches, lightning, and fog – have only been anecdotally associated with WDs (Rangachary
and Bandyopadhyay, 1987; Ganju and Dimri, 2004; Srinivasan et al., 2005; Das et al., 2006; Thayyen et al., 2013); as have coldwaves (Bedekar et al., 1974; Samra et al., 2003; De et al., 2005). However, recent improvements to cataloguing both WDs and hazards means that a more frequentist approach can now be taken.

## 3.2 Precipitation

### 3.2.1 Evaluation of precipitation datasets

The quality and variety of precipitation datasets has improved remarkably over the last decade, due to improvements in satellite retrieval, higher-resolution reanalyses with improved parameterisations, and increasing availability of gauge data.

Baudouin et al. (2020a) completed a comprehensive intercomparison of 20 gridded precipitation datasets (satellite, reanalysis, and gridded gauge), including the recently published ERA5 and MERRA-2 reanalyses, over the Indus River basin. Using a cross-validation technique to determine whether sources of variability were internal (e.g. bias within a single dataset or com-
mon methodology) or external (e.g. due to WDs or the diurnal cycle), they showed that reanalyses capture daily and seasonal variability well, which has useful implications for regions where gauges are sparse or do not undergo rigorous quality control. They found that gauge products tend to underestimate winter precipitation, which may be due to limitations in detecting snowfall (Strangeways, 2010; Dahri et al., 2018); even so, they concluded that of all datasets, APHRODITE (Yatagai et al., 2012, 2017) best captured the daily and intraseasonal variability. Nischal et al. (2022) undertook similar work for winter pre-





cipitation over the Western Himalaya, focusing particularly on the newly-released regional reanalysis, IMDAA (Rani et al., 2021). They showed that, compared with fourteen other datasets, the IMDAA reanalysis captured spatial variability well over the region but exhibited a significant wet bias over the Western Himalayan orography.

Some studies have also compared gridded precipitation datasets against individual station gauges to try and make the verification more direct, although this often relies on the unsafe assumption that the gauge data are themselves error-free. Hussain et al. (2017) found elevation-dependent positive biases at higher altitudes in both TRMM and APHRODITE when compared with 27 gauges over the Karakoram and Hindu Kush, though they noted this could be due to gauge undercatch. Kanda et al. (2020) compared seven gridded datasets from different sources against 19 station gauges over the northwest Himalaya and found a consistent dry bias, seemingly at odds with Hussain et al. (2017). However, six of the seven datasets (excepting ERA-Interim) strongly agreed with each other, suggesting that the gauge data were perhaps improperly processed before analysis. Nageswararao et al. (2018b) found a very good agreement between 26 gauges over northwest India and both APHRODITE and the IMD 0.25°×0.25° gridded gauge precipitation dataset, particularly on interannual timescales. However, this argument is somewhat circular, since these gauge data are used to produce both APHRODITE and the IMD dataset. Similarly, Elahi et al. (2023) reported good similarity between gauges in the southern Hindu Kush and gridded-gauge (APHRODITE) and satellite-based (MSWEP) precipitation datasets.

### 3.2.2 Contribution to winter precipitation

Neal et al. (2020) derived thirty weather regimes over India using $k$-means clustering on low-level winds and surface pressure. Four of these regimes were linked to WDs activity, all of which resulted in heavy precipitation over northwest India. Three of the four WD regimes also resulted in precipitation over northeast India, but it is not known whether this arises from downstream remnants of WDs or some other phenomenon.

Midhuna et al. (2020) collated the dates of 582 WDs from IMD weather reports published between 1987 and 2016. They found that days on which a WD was present over the region accounted for over 80% of the December–March precipitation over the Western Himalaya and over 90% over some parts of central and southern Pakistan; however, they recorded a correlation coefficient of only 0.4 (i.e. explaining about 20% of the variance) between monthly precipitation and number of WD days over the Western Himalaya. These contribution values were somewhat higher than given in Hunt et al. (2019a), who used a more restrictive attribution method – only counting rainfall occurring within 800 km of a tracked WD – to show that WDs were responsible for about 55% of winter precipitation over the Western Himalaya and surrounding region, and over 70% in central Pakistan, Rajasthan and Uttar Pradesh. A similar number, 65%, was reported for the Karakoram by Javed et al. (2022). Determining the true attribution may require a more complex approach, as the results of Midhuna et al. (2020) are by definition an upper bound, and the fixed-radius method of Hunt et al. (2019a) neglects important factors like the variable scale of WDs and remotely-triggered cloudbursts (Singh and Thapliyal, 2022). Even so, Hunt et al. (2018c) found that in about 90% of extreme winter precipitation events over the Western Himalaya and northern Pakistan, there was a WD within 1000 km (this falls to 20% in summer).





These results suggest that the true value of the precipitation contributed by WDs is likely to be somewhere between the values stated by Hunt et al. (2019a) and Midhuna et al. (2020), but certainly a majority. This can be confirmed more directly

using stable water isotopic analysis of precipitation over the Western Himalaya: Jeelani and Deshpande (2017) showed that WD precipitation contains a higher deuterium excess (D-excess; 20‰), a strong indicator of vapour source (Dansgaard, 1964), and therefore that (a) it has a different moisture source to precipitation occurring during the ISM and (b) it is the dominant source of precipitation over Kashmir Valley. Maurya et al. (2018) conducted similar analysis on high-altitude snowpack and glaciers over the Western Himalaya and came to the same conclusion.

## 3.3 Water security


Water security and hazards are what make WDs relevant both in everyday life and in future climate scenarios. In bringing the majority of winter precipitation and the majority of heavy and extreme precipitation events to the Western Himalaya and northwest India, WDs are a vital component of water security in these regions. Better knowledge of WDs is thus directly beneficial for decisions around water and disaster management in the Western Himalayan region and downstream (i.e. in the Indus,

and Ganges basins), such as reservoir management, irrigation timing and intensity, efficient water distribution, and disaster mitigation strategies. Advances in these areas also require better observations, forecast skill (particularly on subseasonal and seasonal timescales) and decision-support tools.

### 3.3.1 Glaciers

As the primary source of precipitation in the Western Himalaya, WD activity is strongly linked to glacier growth and ablation

there. Maurya et al. (2018) analysed $\delta^{18}O$ and $\delta^2H$ excess in glacial meltwater in Punjab and found that a significant fraction of the water originating from these glaciers was attributable to WDs. In an 11-year snow budget of the Chhota Shigri glacier in the Pir Panjal Range, Mandal et al. (2022), found that the majority of winter precipitation came from WDs. Similarly, a comprehensive study by Javed et al. (2022) found that WDs supply about 65% of all snow falling on glaciers in the Karakoram. There is also evidence that WDs play an important role in sustaining central Himalayan glaciers (Sundriyal et al., 2018). See

Sec. 7.2.2 for a discussion on how recent trends in WD behaviour have affected glacier mass balance.

### 3.3.2 Water supply

In the highlands, WD-triggered precipitation replenishes the glaciers and snowpack (Gurung et al., 2017) which in turn recharge the rivers – about two thirds of the mean annual discharge of the Indus river comes from spring and summer snowmelt (Bookhagen and Burbank, 2010; Biemans et al., 2019) – and replenish both natural and artificial reservoirs during the spring and early

summer (Das and Meher, 2019). The relative contribution of glaciers and snow to the spring meltwater is highly sensitive to location: analysing $\delta^{18}O$ and $\delta^2H$ in the Sutri Dhaka Glacier Basin in the Western Himalaya, Singh et al. (2019a) found that downstream river discharge there comprised 80% icemelt and 20% snowmelt. Recent isotopic analysis has estimated that





70% of all water discharged by the Indus river comes from precipitation associated with WDs (Jeelani et al., 2021). Similar contributions have been found for other basins in the northwest Himalaya (Lone et al., 2022a; Nabi et al., 2023).

Given that much of the precipitation associated with WDs is solid and serves as seasonal storage, winters with a dearth of WD activity can be particularly devastating. Basu et al. (2017) attributed crop-hampering low rains in November and December 2015 to an anomalously low frequency of WDs, which they associated with strong teleconnections with ENSO and the North Atlantic Oscillation (see Sec. 4 for further discussion). In fact, a prolonged weakening of WD activity can cause so-called 'flash droughts'. Defining flash droughts as events when soil moisture falls from the above the 40th to below the 20th

percentile (defined for each pentad) in a period of less than twenty days, Mahto and Mishra (2020) showed that the Himalaya experienced more flash droughts outside of the ISM than during it, and attributed this to fluctuations in WD activity.

### 3.3.3   Crops and flora

In the lowlands, WDs provide the rainfall needed to support the rabi crop, upon which the region is highly dependent (Usman et al., 2018). In bringing cold weather and snowfall to higher elevations, WDs can also have profound effects on local plant

life at both weather and climate time scales. Omesh et al. (2015) explored the flowering dates of 46 tree species in the Terai region (foothills of the central Himalaya) in 2014 and 2015, and linked the later flowering dates in 2015 to increased WD activity bringing colder weather to the region. Rashid et al. (2020) highlighted a more extreme example, where intense WDs in the autumns of 2018 and 2019 brought snowfall that decimated Kashmiri apple orchards. Using newspaper reports, Bhat et al. (2024) showed that hailstorms, predominantly associated with WDs occurring between April and June, cause significant

damage to agricultural land and fruit orchards in Kashmir, reducing productivity in these sectors by as much as 70%. In all of these cases, these events occurred at the edges of the winter season, and highlight the susceptibility of Himalayan flora to any changes in the seasonality of WDs. There is already some evidence that such a change is occurring (Valdiya, 2020) and, as we will see in Sec. 7.1, this could have disastrous consequences. Indeed, Ravishankar et al. (2015) found that even as far south as Maharashtra, mango crop could be impacted by WD activity.

As we have already seen, WDs are not always destructive to flora, particularly at lower elevations where a constant stream of weaker storms provides the necessary rainfall to irrigate the rabi crop. This positive relationship also sustains the forests of the Western Himalaya: Sebastian et al. (2019) found a significant correlation between winter rainfall and forest growth there, which was especially prominent on interannual time scales, highlighting the importance of the contribution of WDs to total annual rainfall in the region. Similarly, Malik et al. (2022) showed that monthly winter precipitation (and hence WD activity)

is an important control in the growth of certain fir species in the Hindu Kush. Similar results were reported for the western Himalaya by Chinthala et al. (2022) and for cedars in the Hindu Kush by Bhattacharyya et al. (2023), the latter arguing that such flora will benefit from increasing WD precipitation (Sec. 7.3.2). The impact of WD activity – predominantly temperature – has been implicated in the flowering time of many species of tree in the lowlands of the Central Himalaya (Bajpai et al., 2015).

Bajpai and Kar (2018) quantified the local abundance of *Pinus* pollen deposits in ice blocks taken from three glaciers in the Western Himalaya. These glaciers sit well above the tree line, and so the pollen has to be advected there by southerly or





southwesterly winds. Combined with the fact that *Pinus* growth is proportional to winter rainfall, this study neatly highlights how pollen deposits can be used as a proxy for winter precipitation across the Western Himalaya and northwest India. This is a tool which we will see leveraged in many studies in our discussion of paleoclimate in Sec. 7.1.

## 730  3.4  Natural hazards and other impacts

Aside from heavy precipitation and droughts, WDs have been associated with a variety of other natural hazards, including fog, coldwaves, lightning, hailstorms, avalanches and landslides. Such hazards can have wide-reaching consequences, including increasing the risk of road traffic accidents (Hammad et al., 2019). In this section, we discuss recent developments linking WDs to each of these hazards.

### 735  3.4.1  Fog, dust and pollution

WDs have long been anecdotally associated with fog over northern India (Kundu, 1957), especially over Delhi and the surrounding Indo-Gangetic Plains (IGP) where the high hygroscopicity[2] of aerosols is particularly useful in fog production, reducing the required supersaturation from around 0.3% to 0.18% (Wang and Chen, 2019). Ahead of the WD, moist southerlies pass over cool ground, supporting advection fog; behind the WD, subsidence leads to the clear and calm conditions that
support radiation fog (summarised by Sawaisarje et al., 2014; Ghude et al., 2017).

Improved observation techniques over the previous decade have allowed these relationships to be more carefully quantified. Payra and Mohan (2014) tallied foggy days over Delhi between 2002 and 2011. They found that fog was present on about 80% of January and December days, but only 57% of February days and 53% of November days, attributing this variability to WD activity. These values are about twice as high as over the IGP as a whole (Srivastava et al., 2016). Using a radar in Varanasi,
Sunil and Padmakumari (2020) analysed fog events over the Indo-Gangetic Plain (IGP) during December 2014 and January 2015. They attributed the majority of variability observed during this period to the passage of WDs, noting the important role they played in advecting additional moisture into the region and showing that as a result, the advection fog ahead of WDs was typically thicker than the radiation fog behind them, on account of the increased moisture supply. Similarly, Patil et al. (2020) analysed eight cases of radiation fog using data from an eddy covariance site in Hisar (Haryana) over 2017 and 2018.
They showed that boundary layer turbulence was strongly suppressed in the rear sector of WDs as a result of calm near-surface winds, which combined with high humidity and surface cooling (due to clear nocturnal skies), provided perfect conditions for radiation fog.

Longer-term studies have shown a recent increase in foggy days over north India, consistent with an apparent increase in WD frequency (see Sec. 7.2). Using meteorological registers from 33 stations spread across the IGP and Himalayan foothills,
Srivastava et al. (2016) concluded that foggy days have more than doubled over this region since 1970. Hingmire et al. (2019) also found a significant increase in foggy days from 1980 to 2013 using data from four major cities over the IGP (Delhi, Lucknow, Hissar, Amritsar). While this increasing trend in fog events may be explained by changes in WD activity, increasing

---

[2]the relative tendency of a solid substance to absorb moisture from its surrounding environment





levels of pollution over the region and the increased moisture flux associated with WDs in a warming world may also play a role (Verma et al., 2022, see also Sec. 7.3).

Smith et al. (2022) conducted a comprehensive study, where they identified over 10,000 fog events from observations over India between 2000 and 2020, classifying them according to typology. They found that nearly half of radiation fog events that occurred near Delhi in December and January – the foggiest months – were associated with passing WDs. In contrast, only about one third of passing WDs were linked with the onset of radiation fog. This could be because a significant fraction of WDs are confined to the upper troposphere and have only a weak presence at the surface (Sec. 2.3.4).

By linking their database of fog events with the Hunt et al. (2018b) WD track catalogue, Smith et al. (2022) were able to directly quantify the effect of WD location on radiative fog onset over Delhi. They showed that for almost half of these events, a WD was detected within 1000 km; of those, fog occurred primarily to the southeast of WDs (51%), but can also occur to the northeast (14%) or west (35%). Rather surprisingly, this implies that radiative fog occurs predominantly ahead of the WD. They further argued that recent trends in WD frequency cannot adequately explain the recent increasing trend in 770 fog activity, which they instead attribute to urban expansion and increased aerosol loading. Hingmire et al. (2022) similarly attributed projected near-term increases in widespread fog over north India to local changes in aerosol and pollution loading. However, Gunturu and Kumar (2021) attributed the recent increases in wintertime fog over the Indo-Gangetic Plains to reduced WD activity, arguing that fewer WDs have led to reduced cloud cover and increased radiative cooling. Reduced WD frequency, and hence weaker near-surface winds and precipitation, has also been linked to increased pollution over north India, both in 775 models (Paulot et al., 2022) and observations (Xie et al., 2023; Patnaik et al., 2024). The influence of WDs on air quality also extends to the Central Himalaya – Sundriyal et al. (2018) showed that dust present in the Dokriani Glacier had come from arid regions like the Thar desert (see Fig. 1), and had likely been advected there by WDs. As such, dust deposition there displayed strong interannual variability. There is observational evidence that the northerlies ahead of WDs can worsen air quality in the western Himalayan lowlands, as dust and pollution is advected from north and central India (Kuniyal, 2015). We will discuss 780 in Sec. 7.2 how attributing recent trends to WDs can be challenging, since results are particularly sensitive to the choice of WD dataset.

    These studies have all focused on how passing WDs affect aerosol concentrations or fog events, but it is still not known whether there is a relationship in the other direction – i.e., whether regions of high aerosol concentration have a significant affect on WD precipitation. A recent case study by Kemmannu and Manjunatha (2023), for example, suggested that there may 785 be a positive relationship between aerosol optical depth and the precipitation from dissipating WDs over north and northeast India.

### 3.4.2 Coldwaves

WDs can cause coldwaves – prolonged periods of below-average temperature – both through cold air advection by northerlies in the rear sector and through reduced insolation caused by their deep and widespread cloud cover, as well as radiative conditions 790 that favour surface frost (Mooley, 1957). Recent composite studies of WD structure have shown that they are associated with a lower-tropospheric cold anomaly of 2-3K (Hunt et al., 2018b; Singh et al., 2019b; Midhuna et al., 2020). In case studies, WDs





have been associated with considerably more severe coldwaves (Gupta et al., 2019). A composite analysis of coldwaves over north India by Ratnam et al. (2016) showed that a large subset were associated with upper-level wave activity that strongly resembled a composite WD. Similarly, Sandeep and Prasad (2020) analysed a composite of 21 coldwaves between 2000 and 2016, finding they were often associated with a WD-like trough in the upper troposphere. They also provided some statistics: the average duration of their coldwaves was 10 days, the longest – which they confirmed was initiated by the passage of a WD – caused a temperature drop of 4-6 K for 26 days. Athira et al. (2024) separated coldwaves into normal and intense (i.e. temperature falling below two standard deviations for at least four days). Subsequent composite analysis linked normal coldwaves to WDs, but the intense coldwaves were found to be more commonly associated with omega blocking over Siberia.

### 3.4.3 Lightning and hailstorms

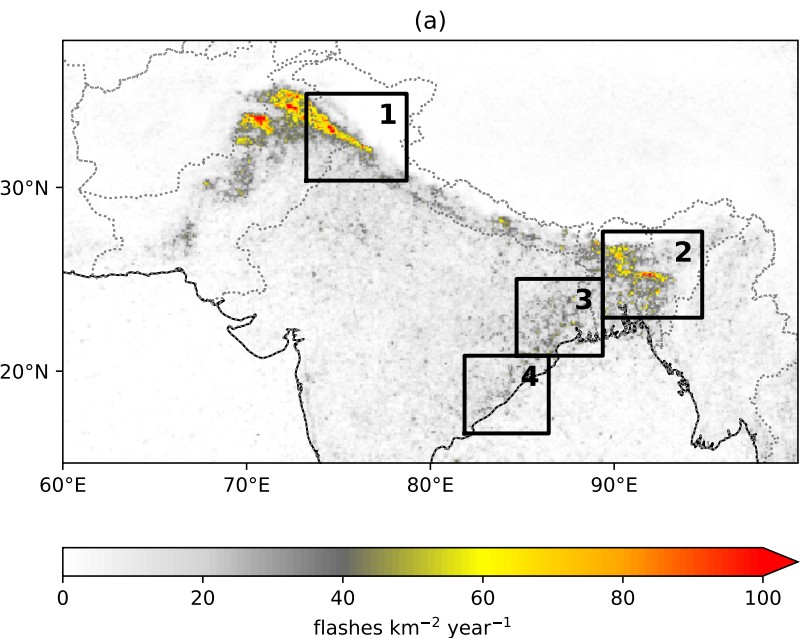

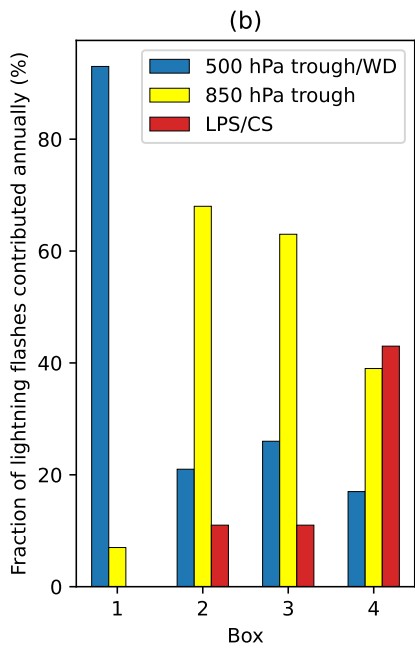

**Figure 12.** Following Unnikrishnan et al. (2021). (a) annual lightning frequency [flashes km$^{-2}$ year$^{-1}$] in the period 1998–2013, from the TRMM LIS 0.1 Degree Very High Resolution Gridded Climatology (Albrecht et al., 2016). (b) fraction of lightning strikes in each of the boxes labelled in (a) associated with WDs [blue], low-level troughs [yellow], and low-pressure systems or cyclonic storms [red]. LPS = low-pressure system; CS = cyclonic storm.

Lightning causes a significant fraction of natural hazard-related mortality in India, and is most common along the Himalayan foothills (Singh and Singh, 2015; Selvi and Rajapandian, 2016; Yadava et al., 2020). Across Pakistan and North India, unlike the rest of the subcontinent, the most severe thunderstorms usually occur in the pre-monsoon (March to May), where they lead to many fatalities (Bhardwaj et al., 2017). These storms can also spawn tornadoes (Bhan et al., 2016). Pre-monsoon storms can also occur near the Himalayan foothills in northeast India and Bangladesh, where they are associated with nor'westers, known





locally as *kal-baisakhi* (Roy and Chatterji, 1929; Das et al., 2014). The link between nor'westers and WDs remains an open question.

Despite their impact, difficulty in obtaining accurate observations of lightning strikes has resulted in few mechanistic or statistical studies. Unnikrishnan et al. (2021) showed that WDs are responsible for 93% of lightning strikes in the western Himalaya (mostly over Jammu and Kashmir, see Fig. 12), more than caused by any other type of synoptic-scale weather over any other part of the country. In a similar study, Murugavel et al. (2021) showed that lightning is very frequent along the Western Himalayan foothills during the winter and pre-monsoon months, which they attributed to passing WDs. Similarly, a study of hailstorms in Kashmir, compiled from archived newspapers (Bhat et al., 2024), found that most storms occurred between April and June. Victor et al. (2023) argued that the seasonal peak in thunderstorm activity over Kashmir (May to June) was caused by moisture transport from WDs reducing CIN, rather than increasing CAPE which can be supplanted by orographic lifting.

Dutta et al. (2016) used a C-band polarimetric radar in Delhi to examine four winter hailstorms. Despite a small sample size, they found that those associated with low CAPE and high vertical wind shear (e.g. those associated with WDs) produced smaller hail than those associated with high CAPE and low wind shear (e.g. those associated with cloudbursts). Using five years of thunderstorm reports over India, Sharma et al. (2023) showed that WDs occurring before the onset of the summer monsoon (April and May) were often associated with substantial lightning outbursts around the Western Himalaya and north India. They argued that this was due to southerly moisture flux associated with WDs passing over the quasi-permanent heat low present over northwest India during these months, leading to large-scale instability. This has been supported by later case studies (Shukla et al., 2022; Gupta et al., 2023), which also highlight the presence of WD-triggered duststorms. A radar-based study by Jaiswal et al. (2023) also showed, for a case study in May 2020, that convection-driven gravity waves associated with WDs may play a role in intensifying lightning and hailstorms.

### 3.4.4 Landslides and avalanches

In bringing heavy precipitation to large regions of steep orography, WDs present an inherent landslide risk – although this has not been quantified until very recently. Using a global database of precipitation-triggered landslides (Kirschbaum et al., 2010, 2015), Hunt et al. (2021) investigated the synoptic-scale conditions present during 327 landslides that occurred in the Upper Indus Basin between 2007 and 2015. During the extended winter (October to April), such landslides occurred at a rate of about one every twenty days, with 61% being associated with the passage of a WD. As in Baudouin et al. (2021), they showed that the presence of a WD at the right latitude (30°N) enhances southwesterly moisture flux from the Arabian Sea, resulting in precipitation covering a large area over the Upper Indus Basin and that the likelihood of a landslide is greatly reduced if the WD is too far north during its passage through the region. Further, they showed that winter landslides in the Upper Indus Basin that could not be linked to WDs were associated with local, smaller-scale, orographically-driven cloudbursts. The interaction between a WD and a tropical low brought record-breaking rainfall and widespread landslides to Uttarakhand in October 2021 (Thapliyal and Singh, 2023a).





Similarly, in bringing heavy snowfall to the Himalayas in both the winter and pre-monsoon seasons (and occasionally during
the summer monsoon itself), WDs are a potential driver of increased avalanche risk (Hara et al., 2004), especially given most
avalanches occur in the western Himalaya between January and May (Ballesteros-Cánovas et al., 2018). So far, there is only
an anecdotal link between WDs and avalanches, with a case study by Nischal et al. (2022) citing heavy WD snow as the cause
of an avalanche in Uttarakhand in April 2021. Like WDs, fatal avalanches can occur in any season (McClung, 2016). WD-
related mountain hazards are thus becoming an increasingly important issue as tourism to the Himalaya grows rapidly (Ziegler
et al., 2023), such as in the 2014 Everest avalanche (killing at least 13 people), and the 2014 Annapurna snowstorm (killing
43). Conversely, favourable weather conditions caused by the absence of WDs leads to more successful winter summits of the
8000-m mountains (Szymczak et al., 2021), including the first ever wintertime summit of K2 in 2021 (Matthews et al., 2022).

In summary, recent research has confirmed statistical links between WDs and a range of natural hazards, particularly extreme
precipitation events, landslides, fog, coldwaves, and lightning. What remains unclear, however, is which WD characteristics –
e.g., location or intensity – are most responsible for driving each of these hazards.

## 3.5   Interaction with the Indian summer monsoon

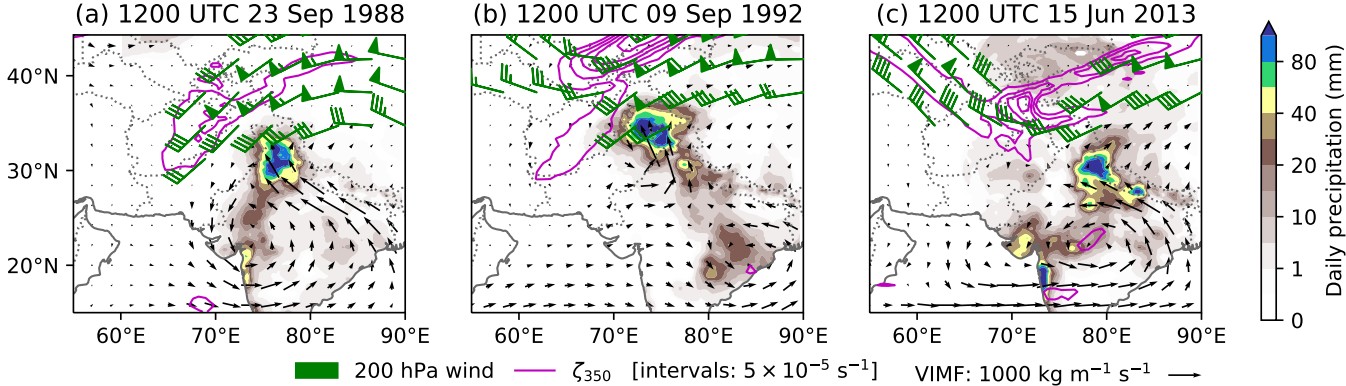

**Figure 13.** Synoptic charts of three cases of severe flooding associated with passing western disturbances: (a) the 1988 Punjab floods, (b)
the 1992 India–Pakistan floods, and (c) the 2013 Uttarakhand floods. Charts are shown for the day on which the heaviest daily precipitation
occurred. Green barbs show 250 hPa winds (where they are in the top quartile for the domain shown), black arrows show vertically integrated
moisture flux, and magenta contours show positive 350 hPa relative vorticity in intervals of $5 \times 10^{-5} \, \mathrm{s}^{-1}$, all computed using ERA5 reanalysis
(Hersbach et al., 2020). Coloured contours show the daily 03 UTC – 03 UTC precipitation, from the APHRODITE gridded gauge product
(Yatagai et al., 2012).

Extreme rainfall events associated with WDs can be catastrophic. These negative WD impacts most commonly occur near the
beginning and end of the Indian summer monsoon (ISM), when southward excursions of the subtropical jet result in embedded
WDs interacting with moist, monsoonal air masses. Strong southerly moisture flux and large regions of ascending air associated
with these WDs create favourable conditions for intense convective and orographic rainfall (Dimri and Niyogi, 2012), which, if
sustained, can result in heavy flooding such as that occurring in Punjab in 1988, north India and Pakistan in 1992, Uttarakhand





in 2013 and Kashmir in 2014. Figure 13 shows synoptic charts for the first three of these cases, highlighting the role played by WDs (magenta contours), the subtropical jet, and their interaction with the monsoon in causing the precipitation that led to these floods.

WDs are most commonly associated with the winter months, when the subtropical westerly jet is furthest south (Schiemann et al., 2009). However, they can occasionally occur during the ISM (June–September), either due to southward excursions of the jet, or as elongated, 'digging' troughs or cutoff lows (Kalshetti et al., 2022). Perhaps the most notorious example occurred in June 2013, when a WD interacted with a monsoon low-pressure system, resulting in devastating floods over Uttarakhand (Kotal et al., 2014). Other cases have been noted, with Uttarakhand seemingly the most common region impacted (Thapliyal and

Singh, 2023b). Majid et al. (2023) showed that floods in the Indus Basin in September 2014 were linked the accumulated effect of several WDs interacting with the summer monsoon. WDs can also interact with tropical storms outside of the monsoon, such as a case in October 2021 which led to record-breaking rainfall in parts of Uttarakhand (Saxena et al., 2023; Thapliyal and Singh, 2023a).

Chevuturi and Dimri (2016) used a high-resolution (3 km), convection-permitting experiment carried out in WRF to inves-

tigate the meteorological causes of the 2013 Uttarakhand floods in more detail. They found that as the WD and monsoon low pressure system converged, a sharp frontal region developed between the cold, dry air associated with the WD, and the warm moist air associated with the low pressure system leading to a pulsatory extension of the monsoon (Pisharoty and Desai, 1956). These conditions led to convective instability that was supported by deep vertical wind shear associated with the subtropical jet and low-level monsoon easterlies, and further enhanced by local orography (Western Himalaya).

The devastating flood in Kashmir valley in September 2014 was caused by the interaction of a WD with an active ISM trough, leading to record-breaking rainfall occurring over parts of Central and North India (Yadav et al., 2017a). This was discussed further in an editorial by Valdiya (2020), which noted that while the southeastward reach of winter WDs seems to have diminished in recent decades, the end of their active season seems to have moved forward from February to April or May. This shift in seasonality was later confirmed by Hunt (2024), as we will discuss in Sec. 7.2. In fact, WDs can occur at any time

of the year (hence their occasional interaction with the summer monsoon), but are usually most active between November and February (Fig. 10).

Hunt et al. (2021) sought to generalise such interactions, identifying 59 cases of interacting WDs and tropical depressions (TDs) over India investigating each using synoptic charts, moisture-tracking trajectory analysis, and vorticity budget analysis. They identified four broad types of interaction (summarised in Fig. 14): (i) a vortex merger, where the upper level PV maximum

of the WD and the lower-level PV maximum of the depression superpose to create a single vortex column; (ii) a jet-streak excitation, where the WD causes a downstream jet streak (see earlier discussion) and a depression enters the associated region of ageostrophic ascent to the left of the entrance and intensifies; (iii) a TD-to-WD moisture exchange, where the depression advects moisture towards the WD, intensifying the precipitation associated with the latter and (iv) a WD-to-TD interaction, the reverse of (iii). They suggested that the 2013 Uttarakhand event was a vortex merger preceded by a TD-to-WD moisture

exchange. One shortcoming of this study was the manual identification of interaction types, with the authors highlighting the need for an automatic method for consistent use in operational forecasting or climate model output. Sadaf et al. (2021) showed





that these sorts of interactions are a vital source of monsoonal rainfall over Pakistan, estimating that 55% of summer rainfall is due to interactions between upper-level troughs (i.e., WDs) and monsoon low-pressure systems.

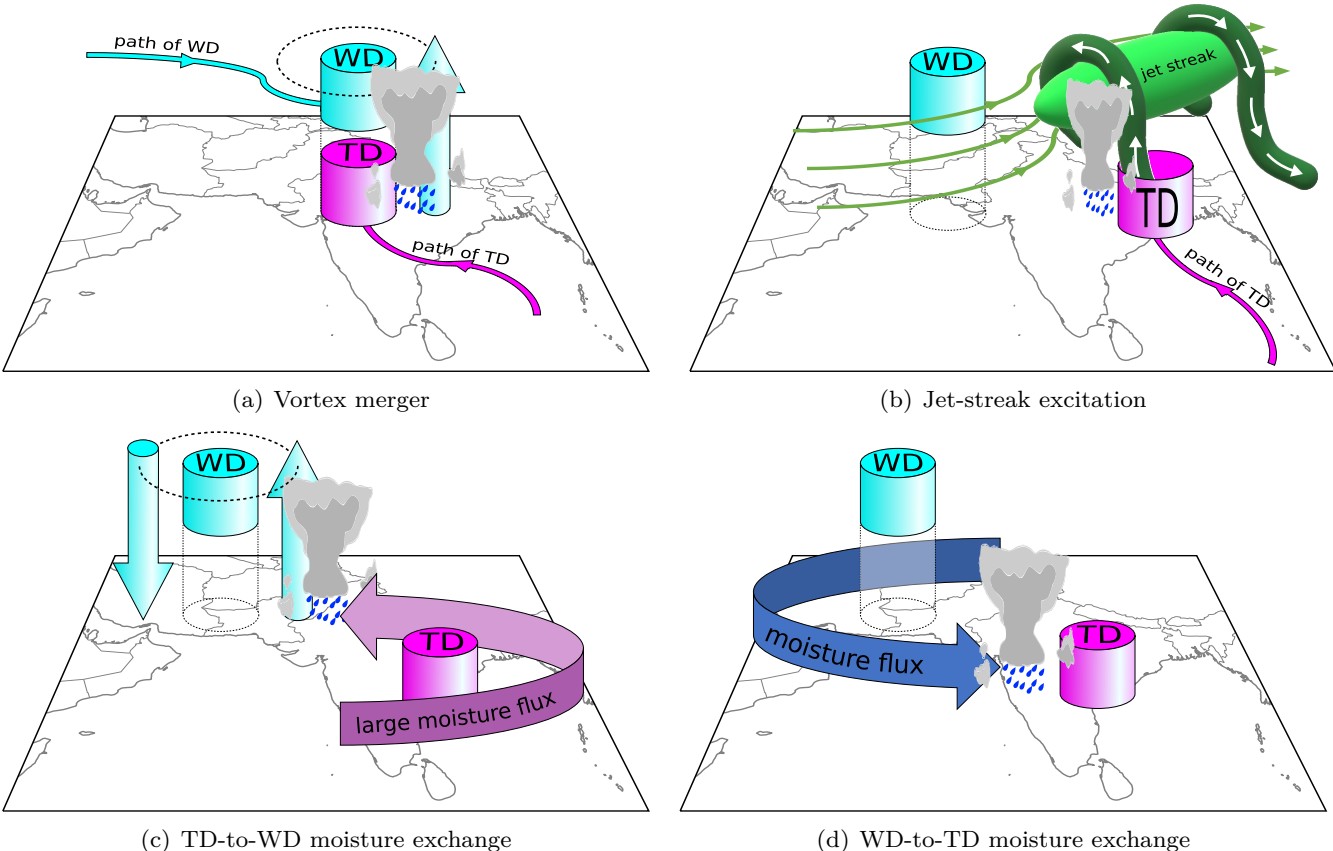

(a) Vortex merger

(b) Jet-streak excitation

(c) TD-to-WD moisture exchange

(d) WD-to-TD moisture exchange

**Figure 14.** From Hunt et al. (2021). Schematics for each of the major types of interaction between WDs and TDs. (a) vortex mergers occur when the vorticity anomaly associated with a WD superposes that of a TD; (b) jet-streak excitations occur when tropical depressions are stimulated through quasi-geostrophic forcing from a streak entrance region associated with a western disturbance; (c) TD-to-WD moisture exchange occurs when net moisture flux passes from a tropical depression to a western disturbance and (d) vice versa, a WD-to-TD moisture exchange occurs when net moisture flux passes from a WD to a TD.

## 4 Large-scale forcing and teleconnections

### 4.1 Summary of earlier research

Interest in the teleconnections driving variability in winter precipitation over the Himalaya has only emerged relatively recently and as such has been covered by few studies. Yet, even very simple analysis – correlating monthly WD frequency with SSTs



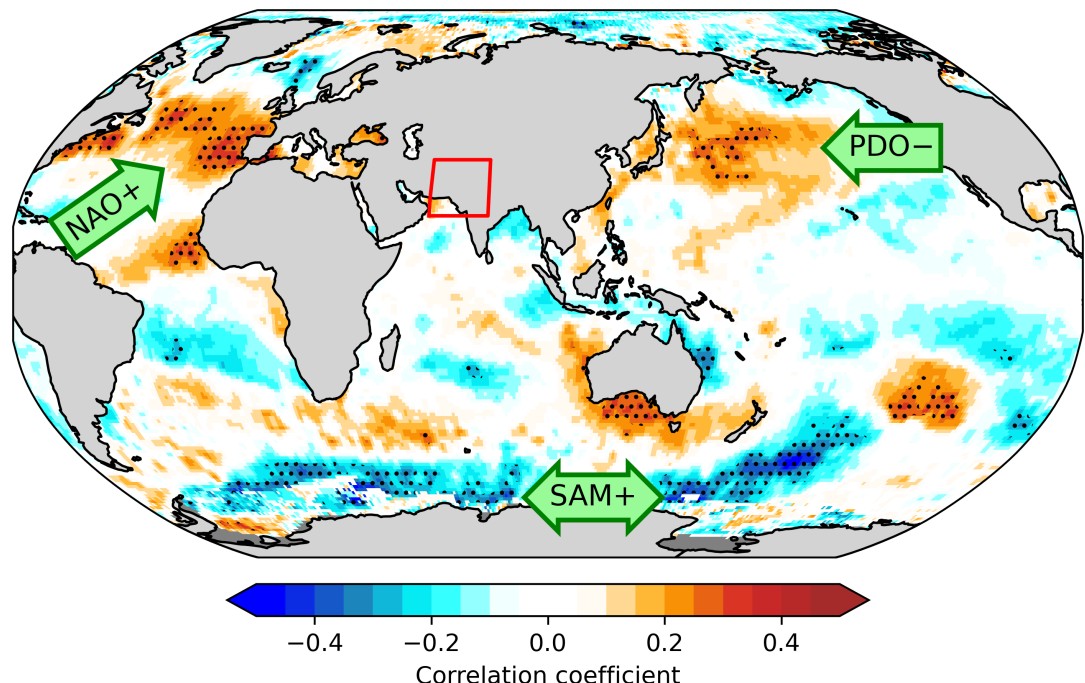

**Figure 15.** Correlation between seasonal mean SST and WD frequency computed for winters (November to March) between 1950 and 2021. Stippling indicates regions where the correlation is significantly different from zero at the 95% confidence interval. Green labels indicate where large regions of significant correlations correspond to known modes of interannual and interdecadal oscillation. The red box marks the region in which WDs are counted, using the ERA5-derived track dataset described in Nischal et al. (2022). SST data are from HadISST (Rayner et al., 2003). Both datasets were detrended before correlation.

(Fig. 15) – reveals a broad inventory of plausible teleconnections[3]. Establishing teleconnections with slowly-varying modes of variability offers potential improvements to seasonal forecasts of WD frequency and winter precipitation.

Studies have generally agreed that the sign of the correlation between ENSO and seasonal winter precipitation over the Western Himalaya is positive (Roy, 2006; Syed et al., 2006; Yadav et al., 2009; Bhutiyani et al., 2010), but typically report weak or insignificant correlations. Competing hypotheses emerged, e.g., a weakening of the Siberian High (Syed et al., 2006), a shift in the jet due to an anomalous Walker circulation (Yadav et al., 2009), and increased moisture supply to the Upper Indus Basin (Mariotti, 2007; Syed et al., 2010) but studies agree that the end result is increased WD activity. Roy (2006) also reported

a positive relationship with the Pacific Decadal Oscillation (PDO; an interdecadal mode of climate variability, whose positive mode is associated with cooler SSTs in the western midlatitude North Pacific and warmer SSTs in the eastern North Pacific).

     Studies have generally agreed on a positive correlation between the North Atlantic Oscillation (NAO; an interannual mode of climate variability whose positive mode is associated with low sea-level pressure over Iceland and high sea-level pressure

---

[3]The SAM and NAO are, strictly speaking, atmospheric phenomena. However, they also rectify onto SSTs forming very similar patterns to those shown here (Kushnir et al., 2006; Sen Gupta and England, 2007).



over the Azores) and winter precipitation over the Western Himalaya that has likely weakened over recent decades (Syed et al.,
2006; Yadav et al., 2009; Filippi et al., 2014), with the proposed teleconnection mechanism in each case being an intensification
of the subtropical jet leading to increased WD frequency.

As we will see in the following subsections, recent studies have therefore sought to further quantify these teleconnections,
but, more importantly, explain the causal processes that support them.

## 4.2 The North Atlantic and Arctic Oscillations

Since Yadav et al. (2009) and Filippi et al. (2014), a number of studies have focused on quantifying or explaining the relation-
ship between the NAO and winter precipitation over India, typically using changes in WDs and the subtropical jet to explain
the mechanism. As we will see, these studies generally agree on the sign and significance of the relationship, with exceptions
mostly arising due to small sample sizes.

Like earlier studies, Nageswararao et al. (2016) found a positive correlation between the NAO and winter precipitation over
northwest India, peaking in the 1960s and subsequently weakening at the expense of a strengthening relationship with both
ENSO and the Arctic Oscillation (AO; a mode of climate variability related to the NAO but with no fixed periodicity, whose
positive phase is associated with low surface pressure over the Arctic and high surface pressure in the midlatitudes). Using
gauge data over Jammu and Kashmir from 1980 onwards, Zaz et al. (2019) quantified a strong positive correlation (0.68)
between the NAO and seasonal mean winter precipitation which they attributed to northward displacement of the subtropical
jet during negative NAO phases. They also found a recent strengthening of the relationship with the AO. (Sanap, 2023) also
reported a positive correlation between the AO and winter precipitation over North India, which they attributed to changes in
intensity of the subtropical jet.

Increased winter precipitation during positive NAO and AO phases is most likely associated with increased WD frequency.
Attada et al. (2019) approached this problem through winter temperature variability over the northern Arabian peninsula. They
showed that warmer winters there are associated with negative phases of the NAO and AO, which decrease the upper-level
meridional temperature gradient, weakening the subtropical jet and decreasing WD frequency. Midhuna and Dimri (2019)
noted increased WD frequency and a colder, wetter winter over the Western Himalaya during years with a positive AO. Like
Attada et al. (2019), they linked this to a stronger subtropical jet associated with a stronger meridional pressure gradient. In a
similar study, Midhuna and Pattanaik (2023a) found a significant positive correlation between NAO phase and the number of
extreme precipitation days in a given winter over north India. A composite analysis by Hunt and Zaz (2021) found that winters
with a strongly positive NAO resulted in a stronger subtropical jet, which in turn forced more frequent and more intense WDs,
driving increased winter precipitation over the Western Himalaya and Indus Basin.

Through its relationship with WDs, the NAO affects many components of the western Himalayan climate system. For
example, Nageswararao et al. (2018a) quantified the impact of climate variability – including the NAO, AO, and ENSO – on
rabi crop production in northwest India from 1966 to 2011. They found that the results were sensitive to crop species, but that
wheat production (and thus winters with greater precipitation) had a positive correlation with both the NAO and AO. Greene
and Robertson (2017) found evidence of a link between the NAO and decadal variability of winter and spring precipitation in





the Upper Indus Basin in CMIP5 models. Hingmire et al. (2019) found that non-foggy days over the Indo-Gangetic Plain, which they associated with the presence of WDs, preferentially occurred during periods of positive NAO, although – as discussed in

Sec. 3.4.1 – this interpretation is probably incorrect as WDs are often associated with the onset of fog (e.g., Sawaisarje et al., 2014; Patil et al., 2020; Smith et al., 2022). A relationship between the NAO and winter weather over the western Himalaya has also been established in several paleoclimate studies. Chinthala et al. (2022) found a weak but significantly positive correlation between the NAO and winter precipitation over the Western Himalaya in their isotopic analysis of fir tree cellulose. Forke et al. (2019) analysed carbonate sediments in the Arabian Sea deposited over the last 5000 years and found that increased winter

runoff was associated with periods of positive ENSO and positive NAO.

In contrast, some studies have also asserted a negative correlation between the NAO and winter precipitation over the Western Himalaya. Devi et al. (2020) constructed a simple seasonal forecast model to predict winter precipitation over the Western Himalaya and found a strong negative correlation (-0.57) with the NAO in the preceding autumn (September to November). Narasimha Rao et al. (2021a) found a weak negative correlation between the NAO and winter precipitation over the northwest

Himalaya between 1996 and 2016, but noted that the relationship was only statistically significant in November. Basu et al. (2017) attributed a weak 2015 winter monsoon to reduced WD activity as a result of positive ENSO and positive NAO, stating that the positive NAO set up an unfavourable standing wave in the subtropical jet with a blocking anticyclonic anomaly over the subcontinent. Using plankton-based proxies, Munz et al. (2017) showed that unusually wet winter seasons since 1750 coincided with strong negative phases of the NAO. Similarly, Giosan et al. (2018) used plankton-based proxies to show that

strong winter monsoons during the mid-Holocene (4500 to 3000 years ago) were contemporaneous with a more negative NAO and southward migration of the subtropical jet, bringing increased WD activity to South Asia.

In summary, the relationship between the NAO and WD frequency (and thus also winter precipitation) has been extensively investigated. Almost all recent studies agree that positive phases of the NAO lead to a strengthening of the subtropical jet, increased WD frequency, and hence wetter winters over the western Himalaya. Studies that disagree with this position are

uncommon and are either limited to short study periods or paleoclimate studies.

## 4.3   The El Niño Southern Oscillation

The El Niño Southern Oscillation (ENSO) is a climate pattern defined by anomalous SSTs in the central and eastern tropical Pacific Ocean, varying irregularly with a period ranging from 2 to 7 years. ENSO has a strong effect on the Indian summer monsoon – which is significantly weaker during the positive phase of ENSO (El Niño) and significantly stronger during the

negative phase (La Niña) – via modulation of the Walker circulation (Ju and Slingo, 1995; Torrence and Webster, 1999; Turner et al., 2005). However, the relationship between ENSO and the winter weather of India is somewhat less clear although the rectification of ENSO onto changes in upper-tropospheric temperature gradients over South Asia (Xavier et al., 2007) is a potential pathway.

Several recent studies have identified a positive correlation between ENSO phase and seasonal winter precipitation over the

western Himalaya, Karakoram, and surrounding areas (Pal et al., 2015; Kamil et al., 2019; Abbas et al., 2023; Bharati et al., 2024) and that this can be measured indirectly, e.g., through rabi crop yield (Nageswararao et al., 2018a). Two of these studies





explored causal mechanisms in more detail: Kamil et al. (2019) showed a significant positive correlation between seasonal WD frequency and ENSO phase, whereas Bharati et al. (2024) showed that although ENSO has a slight effect on WD frequency and intensity, the required increase in moisture flux is actually explained by a southward shift of the subtropical jet during

El Niño, bringing WDs nearer to the Arabian Sea moisture supply. These results contrast with Basu et al. (2017), who also disagreed with other authors on the sign of the relationship with the NAO. They argued in a case study that the winter drought of November and December 2015 over north India, caused by an anomalously low WD frequency, was partially the result of a positive ENSO warming the Arabian Sea and pushing the subtropical jet further north. Composite analyses by the other studies mentioned here suggest that this interpretation is probably incorrect.

Several recent studies have explored the multidimensional aspect to the relationship between ENSO and the winter weather of the western Himalaya. Dimri (2017), building on Yadav et al. (2013), hypothesised that different flavours of ENSO may have different effects on the winter precipitation. He separated El Niño events into those warming predominantly in the west Pacific (warm pool events) and central Pacific (cold tongue events). The use of a small sample size meant that no clear signal emerged in the composite precipitation fields, but he argued that changes to the Walker Circulation between the two flavours

affect the position and strength of the subtropical jet, resulting in increased precipitation over the Western Himalaya during a Central Pacific El Niño, but an increase in the foothills during a West Pacific El Niño. The stratosphere may also play a role in modulating the effects of ENSO on WDs. Remya et al. (2017) investigated the role of sudden stratospheric warming and the QBO on WDs between 1980 and 2010. They found that when SSW events coincided with the easterly phase of the QBO, the subtropical jet increased in strength, resulting in enhanced WD activity. Crucially, they also found that this effect was

much stronger during La Niña than during El Niño, suggesting that a complete picture of the role of ENSO on WDs must also take into account the QBO. Similarly, Shrestha et al. (2019) hypothesised that a 2–5 year periodicity they identified in winter precipitation over Nepal may be related to ENSO and the QBO.

## 4.4 The Indian Ocean Dipole

The Indian Ocean Dipole (IOD) describes a pattern of anomalous SSTs in the equatorial Indian Ocean, whose magnitude

typically peaks between September and November (Saji et al., 1999; Vinayachandran et al., 2009). A positive IOD event is defined by anomalously warm SSTs in the west of the basin (off the Somali coast) and anomalously cold SSTs in the east of the basin (near the Maritime Continent).

The post-monsoon peak in IOD magnitude suggests that only WDs occurring early in the season (Nov–Dec) would likely be affected, with the hypothesis being that a positive IOD event might lead to more moisture availability via a warmer western

Indian Ocean. However, very little work has been done on connecting the IOD to the IWM or WDs. It is perhaps disregarded as being not only downstream of the region impacted with WDs, but also relatively distant, and whereas ENSO noticeably modulates large-scale dynamics on a global scale, the IOD does not. Pal et al. (2015) included the IOD in a hidden Markov model they devised to understand processes driving winter rainfall in NW India, using observations from 14 stations. Their model identified four regimes: the two associated with precipitation were linked to a zonal wave train across Eurasia, which




the authors argued were WDs. Subsequent correlation analysis showed that these modes were more common and more intense in years with a positive IOD or positive ENSO.

In summary, recent studies have established and attempted to explain a wide array of teleconnections between WD frequency and large-scale remote forcings. Studies generally agree that positive phases of the NAO and AO lead to increased WD frequency, and the single study that examines the relationship between the IOD and WD frequency concludes that the correlation is positive here, too. Studies agree on a positive correlation between ENSO and winter precipitation, although different WD mechanisms have been proposed. Furthermore, this relationship may be modulated by ENSO flavour, the QBO, or other secondary influences. No work has yet established a link between WDs and any of the SAM, MJO, or PDO, and although recent work by Aggarwal et al. (2024) suggested the recent negative PDO may have decreased pre-monsoon (March–May) rains over north India through decreased WD frequency, they did not disentangle this signal from climate change.

## 5 Model evaluation and verification

### 5.1 Summary of earlier research

The first explicit efforts to model WDs, in which they were initialised in simple NWP models, showed that even these simple models were able to capture their large-scale structure and propagation (Ramanathan and Saha, 1972; Chitlangia, 1976). Since then, two broad lines of inquiry have emerged, quantifying the respective impacts of improved resolution and physics parameterisations.

Thus far, studies have found that increasing horizontal resolution consistently leads to improved representation of WD behaviour and impacts. At climate model resolutions, this improvement comes from the better-resolved orography – as demonstrated for 90 km to 30 km by Dimri (2004), and then down to 10 km by Dimri and Niyogi (2012). At this scale, improvements in resolution have a greater impact on modelled WD precipitation than advancements in convective or boundary layer parameterisation schemes (Dimri, 2006). As resolution increases beyond 10 km, models permit convection to occur explicitly (even if it is not well resolved until kilometre-scale grids), which results in better representation of the interactions between convection and orography (Garvert et al., 2007; Roberts et al., 2009) and a considerable improvement in the simulation of high-intensity WD precipitation (Norris et al., 2015) when compared to models where convection is fully parameterised. Marginal gains have also been observed up to 2.2 km (Norris et al., 2017) and there is evidence that increasing beyond 1.5 km would further allow realistic simulation of orographic updrafts (Kendon et al., 2012), in particular by better simulating mountain crests and wind funneling, which may be an important feedback in WD-orography interaction (Baudouin et al., 2021).

Studies largely agree that the choice of parameterisation scheme has little impact on WD simulation, except perhaps at very coarse resolutions (Azadi et al., 2002). This is true both for microphysics schemes (Dimri and Chevuturi, 2014a) and land surface schemes (Thomas et al., 2014), although the latter suggested that the choice of land surface scheme may be more important at higher resolutions as the feedbacks between low-level humidity and cloud structure should be better represented. Simulations of WD structure and precipitation are, however, sensitive to initial conditions (Semwal and Dimri, 2012) and choice of data assimilation (Rakesh et al., 2009b, a; Raju et al., 2015).



Three important questions remain unanswered by this earlier research. How good are climate models at representing WDs and their statistics? What can be learned about the structure of WDs and their interaction with the environment from high-resolution convection-permitting models? How important is the choice of convection scheme in simulating WDs? In the next subsection, we will discuss how recent research has tackled these questions.

## 5.2 New developments

As with previous research, recent developments fall into two categories: testing model sensitivity to the variation of empirical parameters, initialisation datasets, or data assimilation; or evaluating model capability, either in NWP or climate frameworks.

### 5.2.1 Numerical weather prediction models

Investigating model sensitivity, with the object of improving the representation of the dynamics or impacts of WDs, has remained a popular area of research, particularly in WRF. Patil and Kumar (2016) modelled four case studies in WRF to test the sensitivity of simulated rainfall to different microphysics schemes using a 9-km resolution nested domain with parameterised convection. All of their experiments showed a southwesterly lower-tropospheric wind bias over the Western Himalaya, leading to significant biases in the location of heavy precipitation. They found only a low sensitivity to the choice of microphysics scheme. Sarkar et al. (2019) examined 30 WD case studies and also found that the choice of convection and microphysics scheme had little impact on precipitation biases, with a consistent wet bias over orography. In a parallel study, Sarkar et al. (2020) found that adjusting parameters associated with the boundary layer parameterisation resulted in more realistic humidity and theta profiles in a composite of 9 WDs modelled in WRF. Choice of initialisation datasets appears to be much more important than convection or microphysics schemes, as shown by Thomas et al. (2018), who simulated three WD case studies in WRF at 15 km resolution using two different land-use datasets. They found that experiments initialised with AWiFS (developed by ISRO) simulated precipitation significantly better that those initialised with the (somewhat older) USGS dataset. Simulations of summer monsoon precipitation, including over north India, are also highly sensitive to the choice of land use datasets (Menon et al., 2022).

Recent NWP model evaluation studies have focused on assessing how well extreme precipitation associated with WDs is simulated in various WRF configurations. The key difference with earlier studies of similar focus is increased resolution, which is sometimes sufficiently fine to permit explicit convection. (Sarkar et al., 2019) ran 30 case studies at 9 km with both explicit and parameterised convection, but found that the parameterised simulations produced more realistic rainfall and concluded that 9 km was too coarse to turn off convection schemes. Regardless, as we have already seen, studies have shown that WRF can produce fairly realistic heavy WD precipitation even at these grey-zone resolutions. This is because they are still capable of capturing much of the necessary local thermodynamics – (Patil and Kumar, 2017) demonstrated realistic CAPE and OLR behaviour in two WRF case studies – as well as the synoptic-scale dynamics – Mannan et al. (2017) demonstrated realistic precipitation even for the unusual situation of WDs passing over Bangladesh, where they draw on moisture flux from the Bay of Bengal. Higher resolution convection-permitting runs improve further on these results, not only due to improved representation of convection, but also due to improved representation of the finer details of the orography, to which the local





dynamics can be highly sensitive. Narasimha Rao et al. (2021b) ran two case studies at 3 km resolution to investigate how well WRF, with assimilation, could forecast WD-triggered extreme snowfall in the Western Himalaya. They found that their 72-hour forecasts simulated the extreme precipitation well, but that the forecasting system benefitted significantly more from 4D-VAR assimilation than 3D-VAR, highlighting how the former is more sensitive to changes in local stability and circulation.

Higher resolutions also permit the evaluation of fog simulations, as Kutty et al. (2021) did for an intense WD-triggered fog event over Delhi that occurred during 15–16 February 2000. Testing a range of resolutions in simulating Himalayan weather for a full year, Karki et al. (2017) showed that convection-permitting resolutions resulted in far better representation of WDs and associated precipitation through better resolved diurnal and orographic processes.

Moving away from WRF, Laskar et al. (2015) comprehensively examined two cases of intense WDs that occurred during
March 2015. Using output from the IMD operational model, the GFS, and local Doppler weather radars, they found that extreme precipitation associated with the WDs, linked to anomalous southerly moisture flux from the Arabian Sea, was undersimulated by the models due to their poor representation of deep convection. Dutta et al. (2022) showed that this weak wind bias in forecast WDs could be overcome by assimilating winds from Doppler radars in north India.

### 5.2.2 Climate models

Another important aspect of model evaluation is to assess how well WD statistics – including frequency, intensity, teleconnections, and associated seasonal precipitation – are captured in longer model runs. This is important, for example, before using climate models to investigate the response of WDs to climate change (which we cover in Sec. 7.3).

Palazzi et al. (2015b) conducted a comprehensive review of the performance of CMIP5 models in simulating precipitation over the Karakoram, Hindu Kush, and Himalaya. They showed the multi-model mean had a wet bias in each of these moun-
tainous regions in both summer and winter seasons, and that most individual models had a wet bias in winter precipitation, consistent with earlier results from Lee et al. (2010) and Su et al. (2013) for GCMs in mountainous or high-elevation regions. Many models also struggled to capture the seasonality correctly. Crucially, they argued, much like Sperber et al. (2013) and Sperber and Annamalai (2014) did for monsoons, that there is no 'best' model or group of models in simulating precipitation over these mountainous regions. These results were extended for CMIP6 models by Meher and Das (2024), which argues that
almost all CMIP models have strengths and weaknesses in representing the range of mechanisms required to drive precipitation over the Western Himalaya. Greene and Robertson (2017) further investigated the interannual and decadal variability of winter and spring precipitation over the Upper Indus Basin in CMIP5 models, but found, like Palazzi et al. (2015b), that only a small fraction of the 31 models examined reproduced the seasonal cycle accurately. They showed that in the eight that did, interannual variability was correlated with ENSO behaviour, and – to a lesser extent on decadal scales – the PDO and NAO.

Hunt et al. (2019a) applied the tracking algorithm from Hunt et al. (2018b) to the historical experiments (simulating the 20th century) of 26 CMIP5 models. They found that the structure and variability of WDs was generally well captured, but that there was a strong dependence on model resolution: higher resolution models had a narrower jet and correspondingly more intense WDs. They also identified a systematic climatological wet bias for winter precipitation across all models. The authors linked this to positive biases in WD frequency and intensity due to upstream biases in baroclinic instability of the subtropical jet. A





similar wet bias was found in a subset of three CMIP6 models by Baudouin et al. (2020b), although they also agreed that the underlying WD dynamics that brought moisture flux into the region were well captured. Ehsan et al. (2020) also found a wet bias in winter precipitation over South Asia in their investigation of 79 members (distributed between six models) of the North American Multimodel Ensemble. They found that the predictability of winter precipitation was weak but showed that expected teleconnection patterns were well captured, suggesting that higher resolution models may perform better.

Indeed, higher resolution climate models do perform better: Iqbal et al. (2017) found that models of the CORDEX-SA experiment simulated winter precipitation across the Hindu-Kush and Karakoram well. Similarly, Ghimire et al. (2018) evaluated 11 CORDEX-SA experiments and found that precipitation was generally well simulated over the Himalaya, though a slight wet bias persisted over the Western Himalaya, and there were elevation-dependent biases elsewhere in the Himalaya.

In summary, recent modelling studies fall largely into two categories: assessing model sensitivity to changes in resolution or
parameterisation schemes, or more simply quantifying the skill of NWP or climate models in simulating WDs and their impacts. Recent sensitivity studies have shown there is little, if anything, to be gained by switching or adjusting parameterisations, but have demonstrated that the choice of the land surface dataset is important. Studies agree that increasing resolution is the most obvious way to improve WD simulation in models. This is reflected in both NWP and climate model evaluation studies, with the coarser resolution of the latter resulting in important biases in moisture transport and precipitation over the Western Himalaya.
The broad structure of heavy WD precipitation can now be accurately simulated by models with resolutions at the coarser end of the grey zone ($\sim$10 km); however, the complex interactions between convection and orography that modulate finer features of this precipitation need higher resolution models, ideally with explicit convection, to be resolved. In the next section, we will see how operational weather forecast models perform at predicting WDs over a range of timescales.

## 6 Forecasting

### 6.1 Summary of earlier research

As we saw in Sec. 3, WDs are responsible for high-impact natural hazards including lightning, fog, coldwaves, and landslides. They are also a key component of water security across the northern subcontinent. This necessitates useful forecasts on a range of scales, from the short-term (a few days) to the seasonal (several months).

Early forecast models, despite their simple physics and coarse resolutions, were able to predict WD tracks a day in advance
(Ramanathan and Saha, 1972) by leveraging the large spatial scale of the subtropical jet and associated WD circulation. Increasing model complexity and finer resolution brought improved forecasts: Dash and Chakrapani (1989) showed that a T21 spectral model with five vertical levels could forecast WD precipitation reasonably well up to 48 hours and Gupta et al. (1999) showed that the NCEP model run operationally by NCMRWF (T80, 18 levels) could forecast WD precipitation well up to 72 hours. These forecasts suffered from dry biases near and over the Himalayas, driven in part by their inability to resolve the
orography. This was also reported in the 1° IMD operational model by Hatwar et al. (2005). It is interesting to note that these dry biases in early NWP forecasts stand in contrast to the prominent wet biases found in CMIP5 models over the same region and season (Sec. 5) – although this could be due to the former simulating heavy precipitation and the latter simulating seasonal




means. Das et al. (2003) showed that these biases could be significantly reduced by dynamical downscaling – in their case using NCMRWF operational forecasts to drive a 10 km nested model. The same method also improved short-term forecasts of high impact cloudbursts associated with WDs (Das et al., 2006). Developments in data assimilation have also improved WD forecasts, such as by using satellite data to improve estimates of initial conditions (Rakesh et al., 2009b; Dasgupta et al., 2004).

There are thus three prevailing challenges for forecasting. Firstly, although coarser models can forecast WD movement and circulation well, reliable forecasts of their impacts require a much higher resolution – likely convection permitting, as we saw in Sec. 5 – to capture the interactions between convection and the Himalayan orography. Secondly, operational tracking of WDs has only recently become possible, with the recent development of objective WD-tracking algorithms (Sec. 2.2). Before this, forecast verification was largely confined to case studies, and evaluation of seasonal forecasts, which are essentially used to predict statistics of WDs (e.g., seasonal frequency and the regions likely to be impacted), was particularly challenging. Thirdly, verification of impacts – particularly forecasts of heavy precipitation – is challenging due to the relative sparsity of observations over the Western Himalaya, though this has steadily improved over the satellite era.

In this section, we give an outline of the current state of operational forecasts of WD activity at different time scales: short-range (up to about five days ahead), medium-range (up to about 15 days ahead), extended-range and sub-seasonal (up to about 30 days ahead).

## 6.2 Short range

There has not yet been an assessment of the skill of short-range operational NWP forecasts in predicting WD tracks or impacts, due in part to the lack of WD tracking algorithms until recently, and the tendency for forecast verification studies over the subcontinent to focus on the summer monsoon. As recent modelling studies have shown that even relatively coarse NWP models simulate the synoptic-scale dynamics associated with the passage of WDs well (Sec. 5), there is little doubt that operational NWPs would produce skilful forecasts of WD tracks several days in advance. However, the lack of systematic analyses remains a major gap in our understanding.

Therefore, in this section, we illustrate the potential value of such analyses with a short case study. As an example, Fig.16 shows warning bulletin maps released in the four days prior to, and on the day of, four recent WDs. Two of these cases resulted in fatalities: the WD on 07 Jan 2022 led to 22 deaths in Punjab (Pakistan), due to heavy storms and a coldwave; the WD on 18 Oct 2021 led to 57 deaths in Uttarakhand due to heavy flooding. The skill of the October 2021 forecast warnings are also discussed in more detail in Thapliyal and Singh (2023a).

Some preliminary observations:

1. IMD's day-0 warnings, essentially serving as nowcasts, tend to align well with the GPM-IMERG, especially as these warnings pertain to heavy rain or thunderstorms.

2. Risk forecasts are generally accurate up to lead times of 3-4 days, even in regions with complex orography.

3. However, the general tendency is for warnings to escalate in severity as the event draws closer. While this approach minimises false alarms, rapid escalation of warnings challenges timely and effective response. This happened in the 07



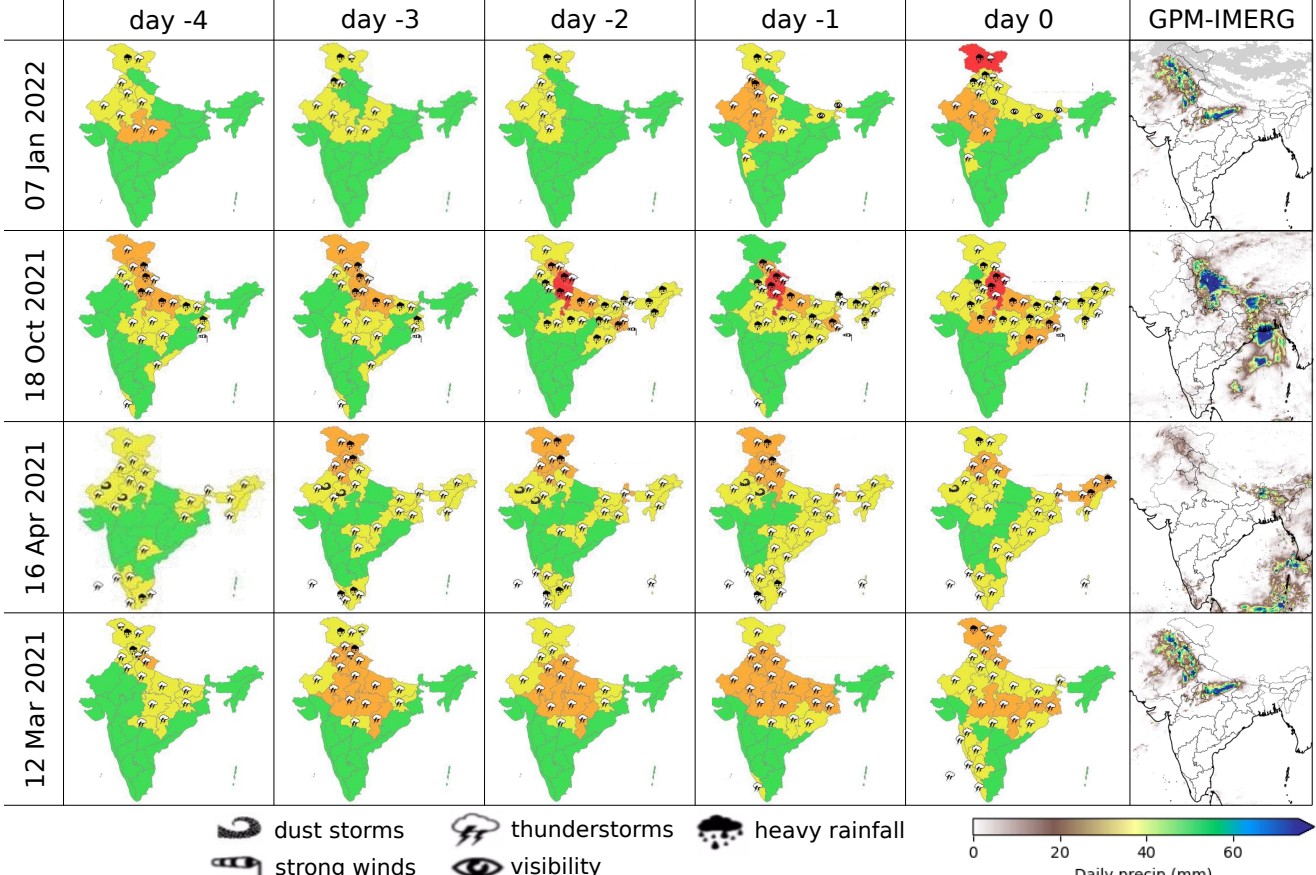

**Figure 16.** Warning bulletins issued by the IMD for four recent dates associated with severe WD weather. For each date, warnings issued up to four days in advance are given (first four columns), as well as the warnings issued on the day itself (second column from right) and observed precipitation from GPM-IMERG (last column). The IMD describes the warning colours thus: green – no warning (no action); yellow – watch (be updated); orange – alert (be prepared); red – warning (take action). Areas masked in grey in the GPM-IMERG panels indicate where data is currently unavailable. All bulletins were taken from the official webpage of the IMD (https://internal.imd.gov.in/pages/press_release_mausam.php).

Jan 2022 case in Ladakh, where a yellow warning the day before the event was upgraded to a red warning on the day itself.

4. There exists an inconsistency between IMD's high-resolution nowcasts (produced using radar and satellite products; not shown) and their day 0 warning bulletin maps. Such disparities can lead to confusion among stakeholders.

A full analysis of warning bulletins and nowcasts in the context of WDs is left as an important topic for future work.




## 6.3 Medium range

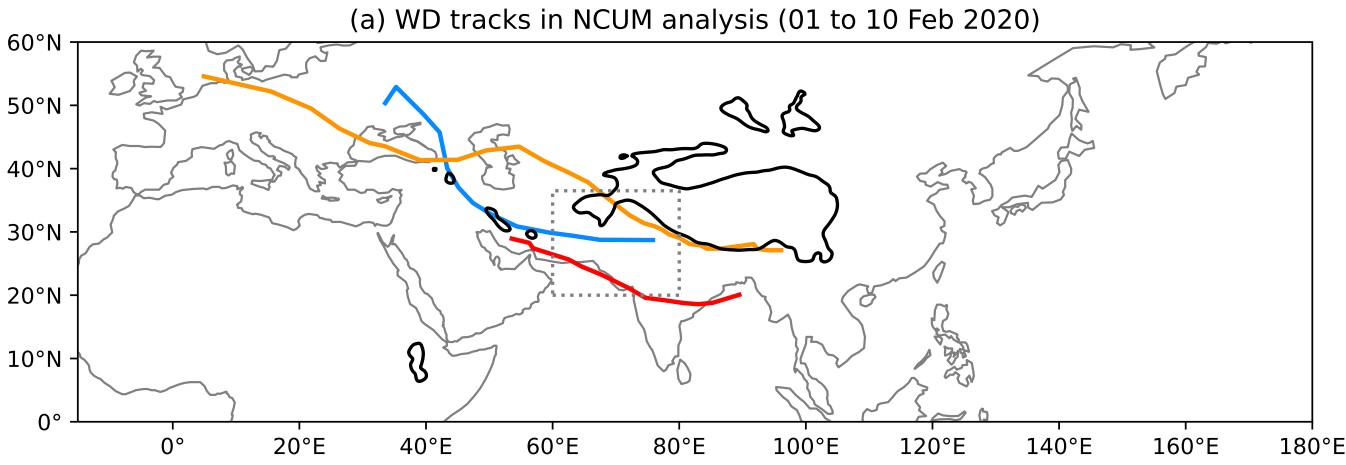

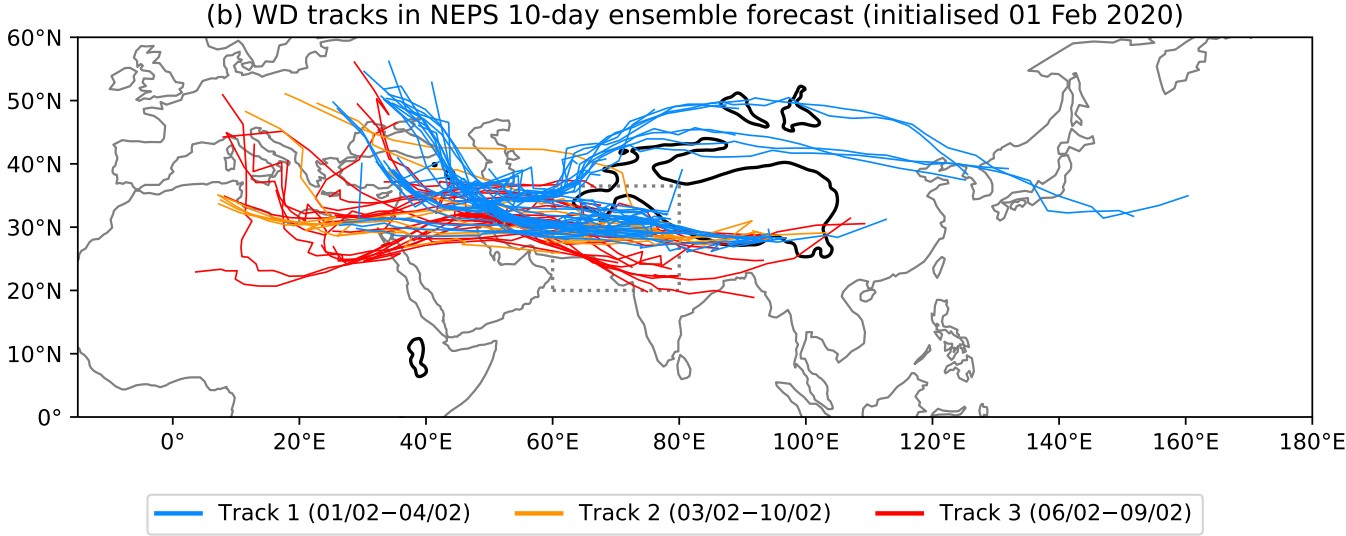

**Figure 17.** Adapted from Arulalan et al. (2020). (a) tracks of three WDs identified in the NCMRWF analysis between 01 and 10 February 2020. The dotted grey rectangle indicates the capture region, through which tracks must pass to be detected. (b) tracks of the same three WDs identified in the 23 members (including one control member) of a 10-day forecast of the NCMRWF ensemble prediction system, initialised on 1 Feb 2020. The thick black line is a smoother 2 km orography contour.

As with short-range forecasting, little work has been done on directly assessing medium-range forecasts in the context of either WD activity or impacts. Arulalan et al. (2020) recently applied the WD-tracking algorithm from Hunt et al. (2018b) to an operational 10-day forecast from the NCMRWF ensemble prediction system (NEPS), as well as the NCMRWF analysis over the same ten day period (Fig. 17). This is a manual verification of a single ensemble forecast and so we cannot draw





any robust conclusions about the performance of NEPS in forecasting WD tracks. While the tracks are forecast reasonably well in this case, it does highlight some potential challenges for medium-range WD forecasts: the development of spurious tracks (i.e., false positives) and the sensitivity of forecast WD track locations to the relative position of the jet and the Tibetan Plateau. The former problem is not unique to WDs and we shall not dwell on it further here. The latter is demonstrated by
track 1 in Fig. 17 and its analogues in Fig. 17(b). Tracks that approach the subcontinent at a latitude south of about 36°N hit the Karakoram or Western Himalaya, where the angle of incidence deflects them south into Pakistan and India where they become WDs. At latitudes above this – about 36 or 37°N – tracks are incident on the Pamirs (see Fig. 1), where they may be deflected south, as before, or north, taking them into Kyrgyzstan and China. This 'Pamir bifurcation' may be sensitive to shifts in the subtropical jet of order one hundred kilometres, as well as to the characteristics of the WD itself, presenting a significant
forecasting challenge as very similar initial conditions could lead to forecasts predicting either an active WD or no WD at all. This is an important area for future research. An additional issue is the reliability of reanalysis tracks as 'truth'. The sensitivity of these results to the tracking algorithm, which could be explored, e.g., using the 10-member ERA5 reanalyses ensemble, is an important consideration for future research in this area.

## 6.4 Extended range and subseasonal

Recent work at IMD, summarised here for the first time, has applied a modified version of the Hunt et al. (2018b) algorithm to track WDs in archived forecasts produced by their extended-range forecast system (Pattanaik et al., 2019), with a view to producing such tracks operationally in the future. The modification was required because the forecast output has daily sampling frequency and so, among other things, the sensitivity had to be reduced to mitigate incorrect linkages.

Verification has been performed for several of the WD cases listed in Hunt et al. (2020) and here we briefly discuss the
results from one selected case (15 March 2015). For this case, IMD operational runs are available with start dates of 11 March, 04 March, 26 February, and 19 February (Fig. 18). Tracks from sixteen ensemble members are shown – four members each at two resolutions (T126 and T382) for two models (GFS and CFS) – for each of these start dates across the four panels of Fig. 18. Only those tracks that are present in the red box between 13 March and 17 March are displayed. For verification, we also include the same WD tracked in ERA5, under the same conditions (i.e. using the same modified algorithm, with daily
frequency data).

For the 11 March initialisation, all four models predict the track well. However, as we go back through previous start dates, the models diverge, sometimes not producing any tracks at all; although both the 19 February and 26 February initialisations have significantly higher WD activity than in the respective model climatologies.

Midhuna and Pattanaik (2023b) showed that extended-range forecasts of winter precipitation over north India in the NCEP
CFSv2 could accurately forecast regional precipitation anomalies as far as three weeks in advance, implying that WD tracks and intensities are generally forecast well. These preliminary analyses suggest that more work is needed to produce useful operational forecasts of WDs in subseasonal to seasonal timescales, and that a more systematic analysis is needed.





**Figure 18.** Operational forecasts of WD track location from the 16-member IMD extended-range ensemble forecast, for a WD impacting the western Himalaya on 13 March 2015. Each panel represents a different initialisation date, with track from the ensemble members given by thin blue lines, and the ERA5 track given by a thick blue line. Filled contours represent the probability of an ensemble member having a WD track pass within 400 km between March 11 and March 13. The closest initialisation is in the bottom right panel (11 March 2015).

# 7 Climate change

By far the most important question on WDs is how they respond to climate change, and there has been little consensus on this in both earlier and more recent studies. There are, broadly speaking, three lines of inquiry: examination of the paleoclimate record, examination of the historical (i.e. instrumental) record, and use of climate models.

The following subsections discuss the extent to which recent studies have been able to leverage longer datasets, the development of tracking algorithms, higher-resolution climate models, dynamical downscaling, and increasing use of advanced





paleoclimate techniques to address these uncertainties in the response of WD activity and winter precipitation to changes in
past, present, and future climate.

## 7.1 Paleoclimate proxy measurements and simulations

Paleoclimate research has become increasingly popular over the last few decades, especially as more advanced proxy techniques have been developed and refined. For precipitation, these include speleothems, marine and lake sediments, tree rings, and pollen analysis. Precipitation itself is often used in studies as a proxy for WD activity over the Western Himalaya – and it is
a fairly reliable one, since the majority of the winter precipitation there is brought by WDs. There are two key reasons to study WDs in paleoclimate: firstly, to understand the impact of external forcing on WDs (e.g., during deglaciation events or Holocene trends); secondly, to characterise the decadal- to centennial-scale variability of WDs and their impact on the evolution of human societies (for which we have two case studies: the Indus civilisation, and the common era).

However, the different characteristics of WDs can only partially be reconstructed using paleoclimate records. Since most of
this information comes from precipitation proxies, the main challenge is to disentangle WD-driven precipitation from other types of precipitation, particularly the summer monsoon. In addition, analyses often make do with proxies from winter precipitation dominated areas nearby (e.g., Iran, central Asia), and extrapolate the result to the study area (e.g. Petrie and Weeks, 2018).

In this section, we aim to cover the literature approximately in chronological order of the period studied, rather than attempt-
ing to group them thematically by proxy type or region, to make the variability over the late Pleistocene (126-11.7 ka[4], the Holocene (11.7 ka onwards), and finally the Common Era (2 ka onwards) stand out more clearly to the reader. Studies based directly on recent observational records (i.e., since about 1900) are discussed in the next section (Sec. 7.2).

### 7.1.1 Late Pleistocene (80–11.7 ka)

The Late Pleistocene was dominated by cold global temperatures and very low sea levels, which culminated during the Last
Glacial Maximum (LGM; approximately 26–20 ka). Of note, the subsequent deglaciation saw a brief reversal to cold conditions in the Northern Hemisphere (Younger Dryas period, 12.9–11.7 ka)

Mineral analysis of moraines in the Karakoram (Ganju et al., 2018) suggested significant glacial advances at around 60 ka and 30 ka, the extent of the latter likely holding until the end of the LGM (18.2±1.8 ka). These advances are associated with periods of increased winter precipitation and colder temperatures, which the authors link to increased WD frequency. In their
comprehensive review of the paleoclimate of Ladakh, Phartiyal and Nag (2022) argued that the climate of the NW Himalaya alternated between summer monsoon-dominated precipitation and WD-dominated precipitation. They argued that the monsoon dominated throughout much of the region in the Late Pleistocene (∼80–35 ka), and that the region was then westerly-dominated from 32–17 ka, consistent with the analysis in Ganju et al. (2018). Nag et al. (2023) analysed lake sediments in Ladakh and placed the WD-dominated period slightly later – during the post-glaciation from 19.6 to 11.1 ka. Within this, they identified a
particularly wet period from 17.4 to 16.5 ka. Similarly, using fossilised pollen, Kar and Quamar (2020) identified particularly

---

[4]ka, or *kiloannum*, measures the number of years, in thousands, before 1950 CE)





wet winters, which they attributed to enhanced WD activity, at the end of the Pleistocene (12.2-11.8 ka), also agreeing with other studies that there was significant variability in both summer and winter precipitation over the Western Himalaya since the beginning of the Late Pleistocene.

### 7.1.2  Early Holocene (12–8 ka)

The Holocene began around 11.7 ka, at the end of the Younger Dryas. The early Holocene marked signficant developments in human civilisation, including in South Asia, as a warmer and more stable climate allowed for the development of agriculture. The period as a whole witnessed a general warming trend and the continued retreat of ice sheets that had expanded during the LGM and Younger Dryas. Studies looking into Western Himalayan climate in this period are often extensions of Late Pleistocene studies.

Both Phartiyal and Nag (2022) and Nag et al. (2023) agreed that the summer monsoon dominated the climate of Ladakh throughout almost the whole early Holocene, with WDs starting to dominate again around the end of this period (8-7.5 ka). This view was not shared in the review by Phartiyal et al. (2022), who argued that the early Holocene was in fact WD dominated, except for arid periods between 10.8–10 ka and 8.8–8.6 ka. Kar and Quamar (2020) also argued for increased WDs in the early Holocene, although their technique could not readily distinguish between summer and winter precipitation.

### 7.1.3  Mid Holocene (8–4 ka)

The mid-Holocene (ca. 6 ka) was marked by an increased seasonality in insolation, resulting in colder winters and warmer summers in the Northern Hemisphere as well as significantly reduced Arctic sea ice (Wanner et al., 2008; de Vernal et al., 2013), leading to a weaker equator-to-pole temperature gradient and hence a weaker winter subtropical jet (Park et al., 2018; Xu et al., 2020). Of particular interest in this period is the so-called 4.2-ka event, an abrupt change in the climate that led to

widespread aridity, marking the end of the mid-Holocene. It is speculated that this aridity led to widespread societal collapse across Eurasia and Africa, including contributing to the demise of the Indus Valley Civilisation (Staubwasser et al., 2003; Robbins Schug et al., 2013; Ran and Chen, 2019). This partially explains the large number of studies focused on this period.

Sediment analyses suggest that for most of the mid-Holocene, precipitation over the western Himalaya and surrounding regions was dominated by WDs, and that this period was, on average, drier than the present day (Trivedi and Chauhan, 2008;

Demske et al., 2009; Leipe et al., 2014; Hamzeh et al., 2016; Phartiyal and Nag, 2022; Phartiyal et al., 2022). This has been supported by modelling studies leveraging output from the Paleoclimate Modelling Intercomparison Project (PMIP) experiments of CMIP5. Hunt and Turner (2019) tracked WDs in the PMIP model output and showed that the reduced winter equator-to-pole temperature gradient that results from using mid-Holocene solar forcing indeed produces a significantly weaker subtropical jet, and, hence, fewer and less intense WDs. They argued that this was the primary cause of the reduced winter

rainfall seen in the mid-Holocene experiments (15% lower than for pre-industrial conditions). Using both speleothem records from north India and PMIP model output, Kumar et al. (2019d) also found weaker mid-Holocene winter precipitation over the western Himalaya compared to the pre-industrial period.





There are contrasting views on the impact of the 4.2-ka event on winter precipitation over north India and Pakistan. Although there was consensus among earlier studies agreed that it resulted in a period of increased aridity across the Western and Central

Himalaya and surrounding regions, the exact timing of the event has yet to be resolved (Singh et al., 1974; Phadtare, 2000; Staubwasser et al., 2003; Dixit et al., 2014; Leipe et al., 2014; Kar and Quamar, 2020).

Using oxygen isotope records from sediment in the northern Arabian Sea, Giesche et al. (2018) suggested that there was a period of relatively strong winter precipitation between 4.5 and 4.3 ka, followed by a significant and abrupt drying at 4.2 ka, lasting through 4.1 ka. This was corroborated by Carolin et al. (2019), who found evidence of regional drying upstream

(modern-day Iran) in the speleothem record between 4.26 and 3.97 ka. A later speleothem study (Giesche et al., 2023), drawing on a western Himalayan cave, came to a similar conclusion, that there was a period of recurring winter and summer droughts over a 230-year period from 4.2–3.97 ka, with the summer monsoon recovering by 3.7 ka. In their multi-proxy assessment of climate in areas upstream of the Indus valley, Scroxton et al. (2022) showed that the 4.2 ka event was likely not to have been regionally coherent, although they did find a region-wide drying event at 3.97 ka. They argued that the 4.2ka event was

"transmitted" from the Mediterranean or Middle East and was thus linked to a period of reduced WD frequency.

The spatiotemporal heterogeneity of the 4.2 ka event identified in Scroxton et al. (2022) may explain why some studies have actually found evidence of increased winter precipitation during this period. Giosan et al. (2018) used plankton fossils as proxies for winter precipitation over India during the later mid-Holocene (4.5-3.0 ka). They found a significant strengthening of winter precipitation and WD activity during this period, contemporaneous with a weaker summer monsoon (Fig. 19). Drawing

on other studies, they linked the stronger winter precipitation to southward migration of the jet and a more negative NAO, although the latter is at odds with Forke et al. (2019), who linked increases in runoff in the Arabian Sea since 5 ka to periods of positive NAO and ENSO and found that winter precipitation promptly declined again at the end of this period (3.3-3.0 ka). A similar later decline was reported by Joshi et al. (2017), who used speleothem records from across the central Himalaya to quantify the climate between 4.0 and 1.6 ka. They identified a major winter drought at 3.4 ka, after which WDs recovered

through about 2.7 ka. Lone et al. (2022b) also reported signatures of increased WD activity from 4.2-3.4 ka, followed by a long period of decreased activity from 3.4-0.6 ka, derived from sediment analysis from across lakes across the northwestern Himalaya. Giesche et al. (2023) reported a period of dry winters between 3.6 and 3.4 ka in addition to their analysis of the 4.2-ka event.

In summary, the mid-Holocene was a period of dry winters for the Western Himalaya. During the 4.2-ka event, which

probably affected the region for several centuries, some areas experienced much drier winters and low WD frequency, whereas some areas experienced the opposite. This climatic shift may have contributed to the decline of the Indus Valley Civilisation, but there is increasing evidence in both pollen and archaeological records to suggest that they may have simply changed their agriculture techniques over several generations to reflect the new conditions (Bates, 2020; Spate et al., 2022).

### 7.1.4 Late Holocene (4 ka – present)

The late Holocene, which covers from about 4 ka to the present day, has had a generally stable climate with conditions largely similar to the present day. The Common Era (2 ka – present) has been extensively studied due to the ready availability of





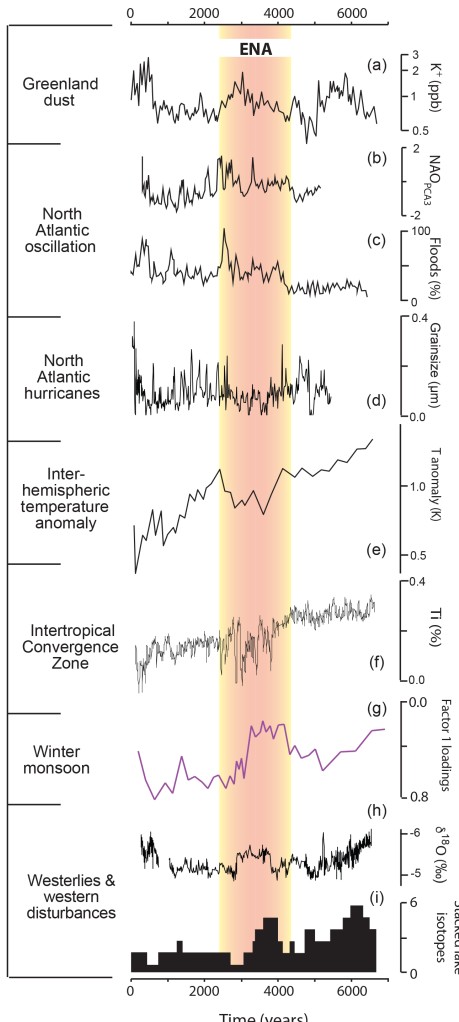

**Figure 19.** From Giosan et al. (2018). Northern Hemisphere hydroclimatic conditions since the mid-Holocene. The period corresponding to the early neoglacial anomalies (ENA) interval is shaded in red hues. From high to low (panels labeled a through i): (a) Greenland dust from non-sea-salt $K^+$ showing the strength of the Siberian Anticyclone (O'Brien et al., 1995); (b) NAO proxy reconstruction (Olsen et al., 2012) and (c) negative NAO-indicative floods in the southern Alps (Wirth et al., 2013); (d) grainsize-based hurricane reconstruction in the North Atlantic (van Hengstum et al., 2016); (e) interhemispheric temperature anomaly (Marcott et al., 2013); (f) ITCZ reconstruction at the Cariaco Basin (Haug et al., 2001); (g) winter monsoon ancient DNA-based reconstruction for the northeastern Arabian Sea (Giosan et al., 2018); (h) speleothem $\delta^{18}O$-based precipitation reconstruction for northern Levant (Cheng et al., 2015); and (i) stacked lake isotope records as a proxy precipitation-evaporation regimes over the Middle East and Iran (Roberts et al., 2011).

high-resolution tree-ring records and historical documents. Periods of note in this era include the Roman Warm Period (c. 250



BCE to 400 CE), the Medieval Warm Period (c. 900 CE to 1300 CE) and the Little Ice Age (c. 1300 CE to 1850 CE). The last 100 years, dominated by instrumental records and active weather observations, are left for the next section.

Previous assessments of paleoclimate records have agreed that winter precipitation over the Western Himalaya was particularly weak between 4000 and 3500 years ago, following the 4.2 ka event, but disagreed on whether it was weak or strong over the following two millennia (Rühland et al., 2006; Mishra et al., 2015; Rawat et al., 2015). However, there is agreement that the intensities of seasonal winter precipitation and the summer monsoon have generally been anticorrelated over this period (Kotlia et al., 2012; Sanwal et al., 2013; Kotlia et al., 2015). There is also consensus among older studies that winter precipitation over

the Himalaya was significantly enhanced during the Little Ice Age (ca. 500 to 200 years ago; Bamzai and Shukla, 1999; Wu and Qian, 2003; Ye et al., 2005).

An important recent development is the discovery that winter precipitation across the Himalaya is strongly anticorrelated with summer monsoon strength on centennial timescales, as demonstrated by two differential studies comparing speleothems and lake deposits from central India and the western Himalaya (Kotlia et al., 2017; Mishra et al., 2018). As we will see, this

leads to better understanding of changes in circulation, but also means that proxies sensitive to the summer monsoon can be used to learn about winter precipitation and vice versa.

Aichner et al. (2015) developed a novel method for analysing leaf wax carbon in an organic sediment core extracted from the Eastern Pamirs, arguing that these are a good proxy for winter temperature and precipitation. They identified cool, wet winters (i.e. increased WD frequency) at 3.5–2.5 ka, 1.9–1.5 ka (100–500 CE), and 0.6–0.1 ka (1400-1900 CE). These were

interspersed with warm, dry winters (i.e. reduced WD frequency) at 2.5–1.98 ka (500 BCE – 20 CE) and 1.5–0.6 ka (500–1400 CE), i.e. the Roman and Medieval Warm Periods respectively. This was supported by Lone et al. (2022b), who analysed lake sediments from across the northwestern Himalaya. They found the late Holocene was in general rather dry, and there was a period of extremely low WD frequency during the Medieval warm period (1.7–0.6 ka; 300–1400 CE).

There is solid consensus among earlier studies that the subtropical jet strengthened and migrated anomalously far south

during the LIA, resulting in increased winter and spring precipitation across north India, and the western and central Himalayas. (Lückge et al., 2001; Brown and Johnson, 2005; Prasad et al., 2007; Newton et al., 2011; Kotlia et al., 2012; Konecky et al., 2013; Böll et al., 2014).

In her review of paleoclimate studies focusing on Himalayan glaciers in the LIA, Rowan (2017) argued that peak moraine building occurred between 1300 and 1600 CE, slightly earlier than the coldest period of Northern Hemisphere air temperatures.

The cooler air temperatures led to both a southward shift in the summer monsoon (i.e. away from the Himalayas) as well as increased winter precipitation from WDs from a period approximately spanning 1400–1800 CE. This early glacier growth was corroborated by Mehta et al. (2021), who estimated peak growth occurred in the NW Himalaya around 1400–1600 CE. Banerji et al. (2019) proposed a potential mechanism to explain the summer-winter anticorrelation: using lead dating of sediments in mudflats along the southern Gujarat coast, they confirmed that the region experienced an unusually cool and wet climate during

the LIA. Combining their results with an earlier study (Prasad et al., 2007), they argued that the climatic changes during this period were due to a southward migration of the ITCZ, which prevented the summer monsoon from reaching much of north India, and also allowed for the subtropical jet to pass unusually far south. Thapa et al. (2022) confirmed, using model results





from the CESM Last Millennium Ensemble (Otto-Bliesner et al., 2016), that changes in solar (or orbital) forcing have been responsible for most of the variance in subtropical jet latitude over the last millennium.

Most other recent studies have focused on constraining the period over which the LIA most strongly affected the Himalayas, revealing more complexity and variability than was previously thought. Yadav et al. (2017c) used a network of tree-ring chronologies in Jammu and Kashmir to reconstruct spring precipitation over the last 500 years. They identified decadal-length droughts centered at about 1490, 1620 and 1860, but perhaps most interestingly noted that the last thirty years were among the wettest in the record, eclipsing the LIA. It is likely that such droughts did not cover the entire region, however, as a
similar 400-year tree-ring chronology (Ahmad et al., 2020), focused on the Hindu Kush, highlighted a completely different set of drought periods (1593—1598, 1602-–1608, 1631-–1645, 1647-–1660, 1756–1765, 1785-–1800, 1870-–1878, 1917-–1923, and 1981–1995), which the authors hypothesised were linked to ENSO and the AMO. Indeed, it seems that there was probably little regional coherence in general during the LIA, as Shekhar et al. (2017) showed, by combining tree-ring data with glacial mass balance models, accumulation and ablation occurred at quite different periods for glaciers in Uttarakhand, Himachal
Pradesh, and Jammu and Kashmir. They hypothesised that the variability was driven by ENSO, the NAO and the AMO. Building on this in Shekhar et al. (2018), they argued that WD variability was a probable driver of seasonal winter droughts over the western Himalaya. Both a recent speleothem study focused on the central Himalaya (Kotlia et al., 2017) and a recent tree-ring study focused on the western Himalaya (Chinthala et al., 2023) agreed that the increased WD frequency associated with the LIA ended around 1820 CE.

Several studies, such as the glacial sediment analysis by Bali et al. (2015) and the tree-ring chronology by Yadav et al. (2017b) have argued that moist winters have continued since the end of the LIA, but these may well be regionally confined as studies covering broader areas suggest that the gradual post-LIA warming is likely to have resulted in weakening winter precipitation subject to strong interannual variability (Munz et al., 2017). Using plankton fossils as a proxy for temperature, Munz et al. (2017) quantified western Himalayan winter precipitation climatology with biennial resolution from 1750 CE
onwards. They found three intense winter seasons in c.1800, c.1890, and c.1930, and linked them to strong negative NAO events – in contrast to the positive correlation between NAO and winter precipitation intensity that has been found in more recent data (Sec. 4.2). They noted that teleconnections with ENSO and the PDO have weakened over the last century, which they postulated as due to global warming.

We conclude this section on paleoclimate with a broad summary, given in Tab. 1. Two items are of particular interest. Firstly,
the disagreements – notably the timing and regions affected by the 4.2 ka event and the LIA – these conflicts may indeed be caused by significant heterogeneity in regional responses to large-scale forcings, but as we will see later, such a pattern is not well supported in observations or climate models. Roy et al. (2022) reported similar inter-study discrepancies in their review of western Himalayan climate over the last 2000 years, and these may also be due the difficulty in disentangling impacts from the summer monsoon, which, as we have seen, is often anticorrelated with winter precipitation over north India on centennial
timescales. Secondly, the pattern that has emerged over the Pleistocene and Holocene – i.e., a warmer planet leads to drier winters with reduced WDs and vice versa – will be a very useful piece of evidence when it comes to understanding how WDs respond to global warming, as we will see in Section 7.3.




For a review of older paleoclimate studies, the reader is referred to Herzschuh (2006), Dimri et al. (2016) and Kar and Quamar (2020).

| Age | Dates | Winter precipitation | Consensus |
|---|---|---|---|
| Late Pleistocene | 60–18 ka (until end of LGM) | strong | good (few studies) |
| | 18–12 ka (until end of YD) | quite strong/variable | poor |
| Early Holocene (Greenlandian) | 12–8 ka | very weak | good (few studies) |
| Mid Holocene (Northgrippian) | 8–4 ka | weak | very good |
| | 4.2-ka event | very weak | good (disagreement on timing and region) |
| Late Holocene (Meghalayan) | 4–3.5 ka | weak | very good |
| | 3.5–2.5 ka | strong | good (few studies) |
| | 2.5–1.9 ka (Roman Warm Period) | weak | good (few studies) |
| | 1.5–0.7 ka (Medieval Warm Period) | weak | good |
| | 0.7–0.2 ka (Little Ice Age) | strong | very good (disagreement on region) |

**Table 1.** Summary of paleoclimate study results from Sec. 7.1, indicating the estimated strength of winter precipitation (and hence WD frequency) over the Western Himalaya and the level of consensus among recent studies for selected geological periods.

## 7.2 Instrumental and reanalysis records

Before 2015, only two studies had attempted to count WDs in the historical record (i.e. since the beginning of the twentieth century). Both did so by associating station-based precipitation with upper-level troughs, and neither found a significant trend in frequency for their respective choice of season and region (Das et al., 2002; Shekhar et al., 2010). Shekhar et al. (2010) noted that this could not therefore explain the observed decline in snowfall days over the same period. A similar recent decline in snowfall over the Western Himalaya was also reported by Gusain et al. (2014) and Singh et al. (2015), with both noting a significant elevation-dependence (the observed decline was much stronger at lower elevations). This elevation-dependent snowfall trend is consistent with other mountainous regions across the globe (Pepin et al., 2022). However, accurate measurements of snowfall are challenging to obtain and so such results must be considered cautiously (Winiger et al., 2005; Palazzi et al., 2013).

There are several methods by which studies have quantified WD variability or trends over the last century or so, and they can broadly categorised as follows.

1. Direct counting using national, state, or station-based reports. This typically involves identifying events where heavy precipitation has co-occurred with an upper-level trough. These are given in blue in Figure 20.

2. Feature-based tracking in reanalysis data, as discussed in Sec. 2.2. These are given in red in Figure 20.





3. Using filtered variance or clustering of an appropriate meteorological field in reanalysis. These are given in green in Figure 20.

4. Inferring changes in WD frequency from trends in other winter weather variables, typically precipitation, in the Western Himalaya or surrounding region. This is discussed in Sec. 7.2.2.

### 7.2.1 Counting WDs

A handful of studies – shown in blue in Figure 20 – have leveraged national or local weather reports to aggregate WD counts. These reports typically attach WD status to storms passing over north India in the winter months that are associated with both heavy precipitation and an upper-level trough. Earlier studies have suggested a decline in WD frequency.

Building on Midhuna et al. (2020), Dimri et al. (2023) tabulated India-wide WD frequency from 1986 to 2020 using annual IMD reports. They reported a significant and dramatic decline of about 1% per year. Kumar et al. (2015) showed that between 1977 and 2007, the number of WDs impacting Himachal Pradesh has decreased significantly, resulting in a significant decline in total winter precipitation (13%) and heavy winter precipitation event frequency (25%) across the state. Focusing on the Western Himalaya, Shekhar et al. (2010) found no significant trend in WD frequency over the period 1984–2007, either in winter (December to February) or in a more extended definition of the winter season (November to April). A similar study, conducted for Jammu and Kashmir between 1980 and 2019 by Ahmed et al. (2022) reported no significant trend in WDs overall, but noted a significant seasonality to the trends – significant increases in January and February WD frequency contrasting significant decreases in November and March – which they linked to similar trends in regional precipitation. Das et al. (2002) tallied the frequency of WDs impacting the Western Himalaya in pre-ISM months of March, April, and May from 1971 to 2000, finding a significant decline only in May. Both Das et al. (2002) and Ahmed et al. (2022) are thus at odds with Valdiya (2020), who argued for a lengthening of the WD season over northern India – i.e., increased frequency in the spring months. Further east, Ahmad and Sadiq (2012) provided a comprehensive overview of WD frequency and variability, grouped by season and latitude, derived from reports issued by the Pakistan Meteorological Department between 1973 and 2007. They found a significant increase in the annual frequency over Pakistan but noted it was subject to strong interdecadal variability: a significant negative trend in frequency until 1985 was followed by a significant positive trend. They also found that the sign of the trend was sensitive to latitude: strongly positive in the north of the country (i.e. north of 31°N) but negative in the south (south of 28°N).

The WD track databases compiled by Cannon et al. (2016), Hunt et al. (2018b), and Javed et al. (2022) based on feature-tracking algorithms (see Sec. 2.2) can also be used to investigate WD trends and variability – noting the usual caveats that apply to using reanalyses to quantify climatological trends. None of these databases, which are shown in red in Fig.20, contain a statistically significant trend. (Javed et al., 2022) investigated WD trends over four adjacent regions – the Karakoram and the western, central and eastern Himalaya (we use data for the western Himalaya in Fig. 20) – and recorded steady WD frequency in each. However, they did find a 10% increase in precipitation intensity over the Karakoram since 1980, which they attributed to an increase in WD intensity driven by increased baroclinic instability. In a follow-up study, Javed and Kumar (2024) attributed



**Figure 20.** Adapted from Hunt (2024). Trends of WD activity since 1950 from the studies discussed in Sec. 7.2 that have available data. For each study, data are presented in the original units, with the stated season and impact region given. Those in blue derive their time series from observational records, such as IMD bulletins; those in red use tracking techniques applied to reanalysis data; those in green use variance-based methods applied to reanalysis data. For each time series, a black $+/-$ in the upper right corner indicates the trend is significantly different from zero at the 95% confidence level; a grey $+/-$ is used for the 50% confidence level.



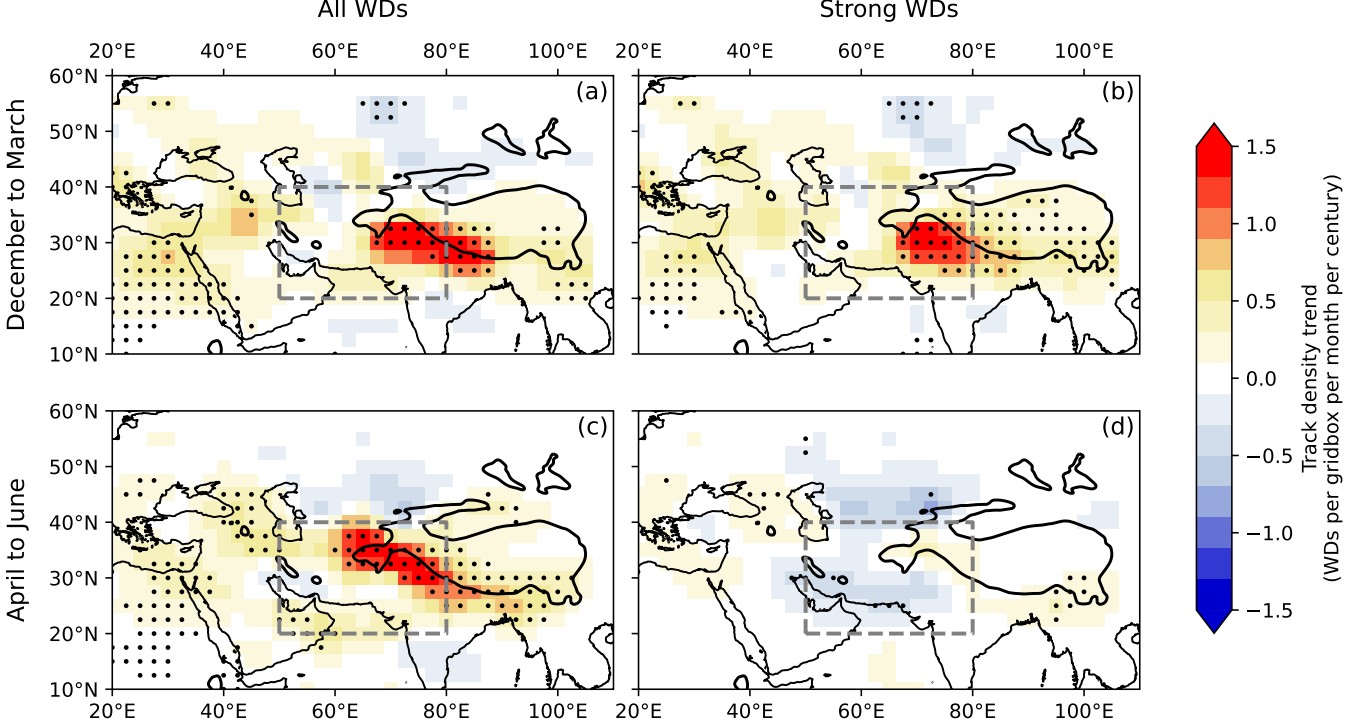

**Figure 21.** From Hunt (2024). Trends in WD frequency from 1950 to 2022. The number of unique tracks passing through each $2.5° \times 2.5°$ gridbox per month is counted, aggregated by season, and the trend is computed. Two seasons are shown: (a) and (b) December to March, and (c) and (d) April to June. To determine whether trends are sensitive to WD detection thresholds, trends for "strong WDs", i.e., those whose peak 350 hPa $\zeta$ is in the top half of the population, are given in (b) and (d). Stippling indicates where trends have $p < 0.05$. The dashed grey box (50–80°E, 20–40°N) indicates the region typically used for aggregating WD statistics.

increased WD intensity and precipitation over the Karakoram to increased moisture availability in recent decades. They also attempted to define a genesis potential index, but it did not correlate well with their observed WD genesis, perhaps because – following tropical cyclone literature – they took vertical wind shear as detrimental, rather than favourable, to the genesis of

baroclinic WDs. Nischal et al. (2022) extended the work of Hunt et al. (2018b) to the newer ERA5 and IMDAA reanalyses (at both T42 and T63 spectral truncations). Similarly, they did not find a significant trend in any of their four datasets: the lowest $p$-value, 0.15, applied to a weak positive trend in the T42-truncated ERA5 track frequency – this is the one shown in Fig. 20. Motivated by earlier disagreements, Hunt (2024) used the T42 ERA5 database to explore WD frequency trends as a function of location and season (Fig. 21). He found that winter WDs have increased significantly over the Western and Central

Himalaya and Hindu Kush in the last 70 years, likely due to increasing strength of the subtropical westerly jet (Pena-Ortiz et al., 2013). He also found that due to delayed northward retreat if the subtropical jet, WDs have also become significantly and substantially more frequent in the pre-monsoon and early monsoon. This has lead to a large increase in the frequency of potentially dangerous monsoon-WD interactions (Sec. 3.5) and corroborating the earlier work of Valdiya (2020). Hunt (2024)





found no signfcant trend in the intensity of winter WDs, as measured by their upper-level vorticity, and the response of WD
intensity to climate change remains an open question.

Where authors want to quantify WD frequency (or related statistics, e.g., of their intensity or precipitation) without using a
tracking algorithm, they typically turn to using some measure of upper-level geopotential variability. These are shown in green
in Fig. 20 and discussed below. For detailed information on the methods used in each of these studies, refer to Sec. 2.2.

Cannon et al. (2015) applied their geopotential variance method to December to March data over two boxes, one to the west
of the Karakoram (58–62°E, 32–36°N) and one over the Central Himalaya (75–78°E, 28–30°N). They found a positive trend
in activity over the Karakoram between 1980 and 2010, but a significant decline in activity over the Western Himalaya across
the same period, which agrees with many of the report-based (blue) studies. The results for the Karakoram box were replicated
by Krishnan et al. (2018), who applied a very slightly different technique to both ERA40 (1958–2002; not shown) and ERA-
Interim (1979–2015; shown in Fig. 20) reanalyses, reporting significantly positive trends in their proxy for WD frequency in
both reanalyses. Madhura et al. (2015) found that the frequencies of PCs associated with their WD-like EOFs, computed using
data from December to April 1948–2011, both exhibited significantly positive trends. The four WD weather regimes from
Neal et al. (2020) can be summed over each season to give a measure of WD days (Fig. 20), which is in weak decline. Dar
(2023) used a simple method to identify WD days, with negative 500-hPa geopotential anomaly over Kashmir. They reported
a significant decline in this quantity since 1980. This is contrast to other studies focusing on Kashmir, and probably indicates
the metric used was too simplistic.

In summary, there is disagreement among recent studies on the sign and significance of the trend in WD frequency over the
past seventy years. The trend is sensitive to the study region chosen, with studies generally reporting positive trends in fre-
quency over the Hindu Kush, Karakoram and north Pakistan, and generally reporting negative trends elsewhere (south Pakistan,
Western Himalayas, and India in general). Indices including both regions tend to show an absence of change. The frequency
trend is also sensitive to the definition used for the winter season. Not only this, but WD frequency displays significant inter-
decadal variability (Hunt and Zaz, 2021), so the trend is also sensitive to the dataset period used. There are many competing
factors at play, for example the subtropical jet has strengthened over the last seventy years (Pena-Ortiz et al., 2013; Hunt, 2024),
but static stability over the region has decreased, an important precursor to deep convection (Yuval and Kaspi, 2020). Finally,
caution must be exercised when intercomparing these studies. While some directly compute trends in WD frequency, others
use different WD characteristics, such as geopotential variance, which is a function of both frequency and intensity.

### 7.2.2 Changing impacts

With variability in WD frequency and intensity comes variability in their local impacts, many of which – such as precipitation
and coldwaves – are of considerable importance to the communities based in and around the Western Himalaya. We can also
infer recent trends in WD frequency and intensity through changes in their regional impacts, such as precipitation (Gurung
et al., 2017) and coldwaves (Das and Meher, 2019). Although WD frequency is highly correlated with winter preciptiation
(Dimri et al., 2023), such inferences over climate timescales must be considered with additional caution, since they are also



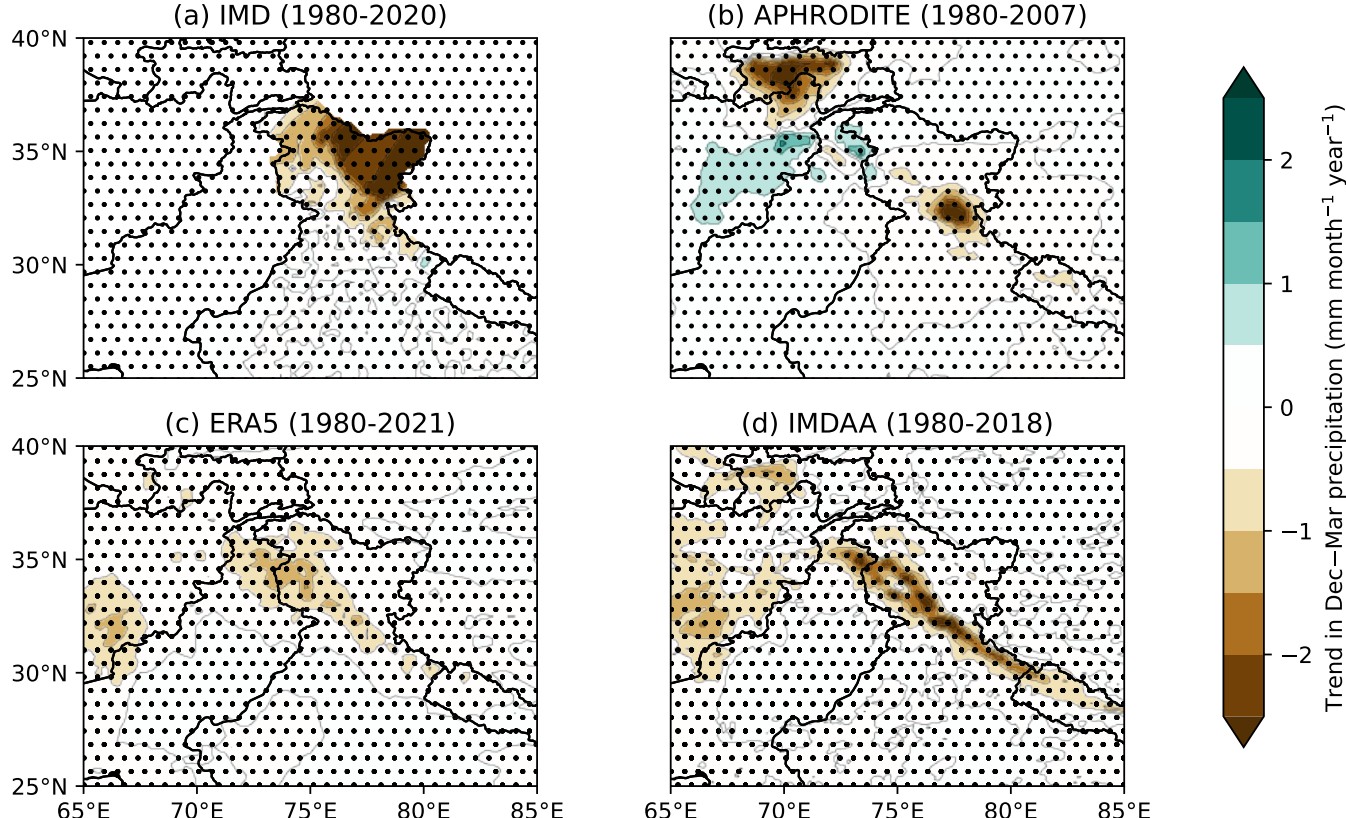

**Figure 22.** Trends in winter (DJFM) precipitation [mm month$^{-1}$ year$^{-1}$ over Pakistan and North India since 1980, as recorded in four datasets, two gridded gauge-based products: (IMD and APHRODITE), and two reanalyses (ERA5 and IMDAA). Stippling indicates where trends are significantly different from zero at the 95% confidence interval.

sensitive to changes in the synoptic environment (e.g., atmospheric moisture content). This highlights one of the most important questions – which has the greater impact on precipitation trends, trends in thermodynamics or atmospheric circulation?

The dominant focus on changing WD impacts has been on precipitation, typically using *in situ* gauge measurements. However, this area is very poorly resolved by the gauge network and there are large biases in measuring snowfall using standard rain gauge equipment (Adam and Lettenmaier, 2003; Strangeways, 2006). This means a proper intercomparison must be conducted carefully, since the heterogeneous behaviour of precipitation means that trends may differ between datasets (Chevuturi et al., 2018) and even between neighbouring basins (Fig. 22). Once datasets or regions with spurious behaviour are removed from the analysis, the key issue is decadal variability – meaning the results are sensitive to the choice of analysis period Baudouin et al. (2020a), who found a regional minimum in winter precipitation between 1995 and 2010. This interdecadal variability has also been found in regional river discharge (Gardelle et al., 2012; Mukhopadhyay and Khan, 2014). These caveats are highlighted by contrasting results in recent studies. In Fig. 22, trends in seasonal winter precipitation from four widely-used datasets are



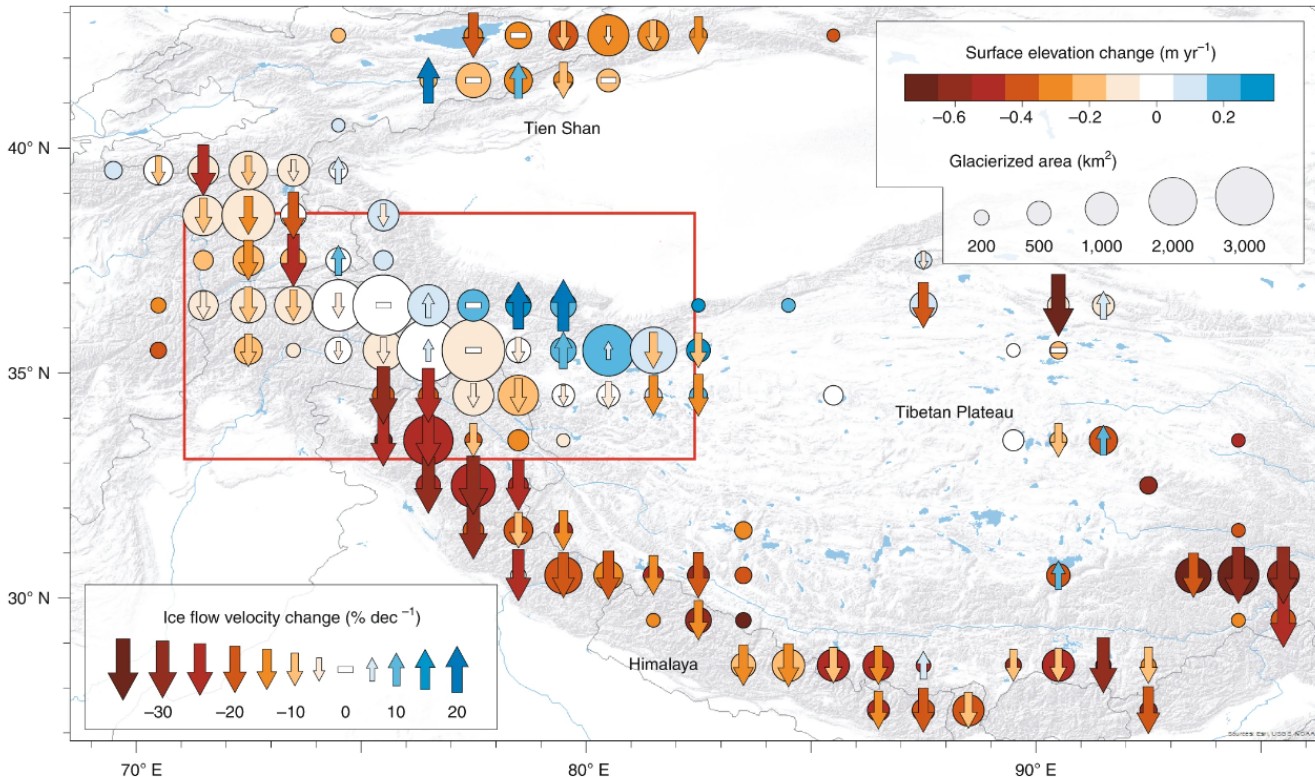

**Figure 23.** From Farinotti et al. (2020). The rate of glacier surface-elevation change (Brun et al., 2017) is shown together with changes in ice flow velocity (Dehecq et al., 2019) for the period 2000–2016. The size of the circles is proportional to glacier area. Data are aggregated on a $1° \times 1°$ grid. The red box marks the Karakoram and surrounding area. Base map from Esri, USGS, NOAA.

shown for the last forty years. The IMD has almost no stations in the only area where the trend is significantly negative, but the other three sources largely agree that there has been significant drying in the Himalayan foothills and over Afghanistan.

Most studies report either a negative or insignificant trend in winter precipitation across the Western Himalaya and surrounding region. In a comprehensive assessment of precipitation trends in different datasets over Leh (Ladakh), stratified by season, Chevuturi et al. (2018) found that that significant interdecadal variability was present across all long-term datasets, making assessment of underlying trends challenging, although the trend in gauge-measured precipitation there was significantly negative. Waqas and Athar (2019) reached a similar conclusion, finding that a weak negative trend in winter rainy days recorded at

stations across northern Pakistan and the Karakoram, and a weak positive trend at stations across the Western Himalaya, were statistically insignificant in the context of large decadal-scale variability. Other studies have reported similar results for the Central Himalaya and Nepal Shrestha et al. (2019), states of north India (Rajasthan, Gujarat, Punjab Narayanan et al., 2016), Jammu (Khan et al., 2023) and Kashmir (Dar, 2023).



Chauhan et al. (2022) found a significant decline in winter precipitation over much of north India between 1901 and 2020, and attributed this to declining WD frequency. However, as in Fig. 22, other studies reporting a significant decline in winter precipitation across regions of north India and Pakistan have measured their trends over comparatively short periods (∼40 years) (Shekhar et al., 2010; Zaz et al., 2019; Ullah et al., 2022; Abbas et al., 2023; Safdar et al., 2023) and so may instead be detecting a mode of interdecadal variability. Such studies typically invoke declining WD frequency and shifts in subtropical jet position as the cause of these trends, as did Gunturu and Kumar (2021), who argued that a recent decline in WDs has been responsible for reduced cloud cover and increased fog over the recent decades.

Several studies have even reported significantly positive trends in winter precipitation, including one long-term study by Nageswararao et al. (2016), who analysed the last century of winter precipitation over north India using the IMD gridded gauge dataset (Pai et al., 2013), in direct contradiction to Chauhan et al. (2022) above. This difference may be resolved by considering that rainfall gauge measurements in the first half of the twentieth century were extremely sparse across the hills and mountains of north India. Long term studies of aridity during the rabi season (i.e. the winter months) have indicated a trend towards wetter conditions over northern Pakistan in the regions typically affected by WDs (Ahmed et al., 2018, 2019), although this too appears to be subject to significant interdecadal variability (Ullah et al., 2022).

Studies agree that climate change has led to earlier snowmelt across the Western Himalaya and Hindu Kush, and that this has a negative impact on rabi crop production (Biemans et al., 2019) and ecosystems (Isaak and Luce, 2023). In a global assessment of extreme wet conditions, Sreeparvathy and Srinivas (2022) found an increasing trend in the variability of short-duration heavy rainfall events over northwest India, which they partially attributed to WDs. Similarly, using gauge, satellite, and reanalysis data, Nischal et al. (2023) reported a significant increase in extreme winter precipitation event frequency across large regions of the western Himalaya and Karakoram since 1980, which they attributed to WDs through observed increases in upper-level baroclinicity and vertical wind shear and a decrease in static stability over the region. Bhat et al. (2024) reported a significant and very large increase in reported pre-monsoon hailstorms in Kashmir between 2007 and 2022.

Some studies have investigated change in coldwave frequency and their relationship to changes in WD characteristics. Both Gupta et al. (2018) – for the Indo-Gangetic Plain – and Dash and Mamgain (2011) – for Jammu and Kashmir – found a significant decrease in coldwaves since the 1980s, albeit again with substantial decadal variability. Both studies attributed their respective trends to decreasing WD frequency or intensity, but such interpretations must be treated with caution given the positive trend in background climatological temperatures. Athira et al. (2024) also identified a significant decrease in coldwave frequency, duration, and intensity over north India.

Interestingly, two studies have discussed the changing impact of WDs on summer precipitation across north India. Prathipati et al. (2019) and Kumar et al. (2020) showed an increase in summer precipitation since 1991 and since 1953 respectively. Because summer monsoon precipitation is reduced elsewhere, both studies suggest that the trend is explained by an increase in WD activity in summer (as hypothesised by Valdiya, 2020).

**elevation-dependent warming.** While the general decline in snowfall is attributed to a warming climate, the spatial variability is thought to be linked to elevation-dependent warming (Dimri et al., 2022), where trends in near-surface warming increase as a function of surface elevation. There is a large body of evidence in support of elevation-dependent warming across



the Tibetan Plateau, Himalayas, Karakoram, and Hindu Kush, though understanding is limited by the sparse observational net-work (Pepin et al., 2015; You et al., 2020; Doblas-Reyes et al., 2021), all the more so for precipitation (Palazzi et al., 2015a). There are thought to be a number of important drivers, depending on season and location, with changes in albedo (Ghatak et al., 2014), snow depth, cloud cover (Duan and Wu, 2006), near-surface humidity (Rangwala et al., 2009), and radiative forcing (Palazzi et al., 2017) chief among them. These point not only to local feedbacks, but the importance of synoptic-scale meteorological drivers (Guo et al., 2021), such as WDs, highlighted by the fact that the elevation-dependent warming signal is strongest in the winter months (Dimri et al., 2022; Hu and Hsu, 2023). Regardless of the sign of change of WD frequency and intensity, the strength of the elevation-dependent warming signal is likely to continue to drive reduced winter snowfall over the Western Himalaya in at least the short term. Still, elevation-dependent warming appears to saturate, or even reverse, at altitudes above about 5 km (Gao et al., 2018; Guo et al., 2019), and appears to have significant interdecadal and spatial variability (Li et al., 2020; Doblas-Reyes et al., 2021). Even so, it is well simulated in CMIP5 (Palazzi et al., 2017) and CMIP6 models (Zhu and Fan, 2022), as well as at a wide range of resolutions (Palazzi et al., 2019).

**The Karakoram Anomaly.** Such results on elevation-dependent warming beg the question as to whether WD activity is playing a role in the Karakoram anomaly, in which glaciers in that region are either growing or stable increasing, counter to the expected impacts of global warming (Hewitt, 2005), and against the trend of glaciers in neighbouring ranges such as the Western Himalaya. Although a handful of articles dispute whether it is a real phenomenon (e.g. Negi et al., 2021), it is generally widely accepted (Gardelle et al., 2013; Miles et al., 2021; Li et al., 2023). The relative growth and shrinking of glaciers across the Himalayan and Tibetan regions over the last two decades is summarised in Fig. 23, from Farinotti et al. (2020). Glacial mass balance in the Himalaya and Karakoram is very sensitive to seasonal snowfall variability (Kumar et al., 2019c), and hence probably also to changes in WD frequency; indeed, WDs contribute approximately 50% of the annual precipitation over Karakoram (Fowler and Archer, 2006; Barros et al., 2006).

The Karakoram anomaly is typically explained through increased cloud cover and snowfall and their associated radiative feedbacks (Bashir et al., 2017; Dimri, 2021; Li et al., 2023) over the region. These are largely explained by changes in local and large-scale circulation, with studies noticing the atmosphere has become more cyclonic over the Karakoram (sometimes referred to as the Karakoram vortex; Forsythe et al., 2017; Norris et al., 2019). This increased cyclonicity is almost certainly related to changes in WD behaviour; however, as we have discussed earlier in this section, studies do not really agree on the sign of change of WD frequency over the Karakoram, although it is probably increasing (Hunt, 2024). Instead, Javed et al. (2022) argued that while there was no significant trend in WD frequency over the Karakoram, WDs have become significantly more intense over the last few decades, and thus have led to a significant increase in winter precipitation over the Karakoram. This mechanism has also led to increased high-altitude snow cover across the upper Indus basin (Bilal et al., 2019).

de Kok et al. (2018) argued that glaciers in the Karakoram have likely been growing due to increased regional irrigation during the pre-monsoon and summer monsoon, which has accelerated significantly in the last few decades. They hypothesised that the anomaly will disappear in the unlikely scenario that this irrigation stabilises or reduces but that this reduction could be amplified, offset, or even reversed by any trends in WD-related snowfall over the Karakoram. Disentangling these two effects may be challenging, as irrigation also provides a local source of moisture for precipitating WDs. Potential sensitivity





to WDs has been highlighted in the Western Himalaya too: using measurements of glaciers there starting in 1901, Mehta
et al. (2021) showed that trends in glacial ablation are most closely associated with increasing temperature, but that trends in
glacial accumulation are mostly closely associated with increased winter precipitation, particularly as a result of changing WD
activity. Over a short study period (2016–2019), they also found a strong sensitivity to altitude: showing that glaciers below
∼5200 m lost mass, but gained mass above that altitude, perhaps due to increased precipitation in a warming environment.
Overall, however, glaciers in the Western Himalaya lost mass at a rate of 0.36 m.w.e. (metres of water equivalent) per year
between 2016 and 2019. The relationship between WD-related precipitation and surface elevation was also described over
Pakistan by Kattel et al. (2019), who showed higher elevations experienced significantly more winter precipitation and cloud
cover than lower elevations.

Despite these advances, it is clear that a great deal more research is needed on how climate change across the Himalayas,
Karakoram and Hindu Kush will have downstream impacts on wetlands, agriculture, and ecosystems in general (Chettri et al.,
2023).

In summary, recent work shows that trends in winter precipitation over the Western Himalaya are a complex function of
period, location, and dataset, though most measurements indicate a strong decadal-scale mode of variability. Inference of WD
behaviour from these trends is thus also very challenging, additionally being sensitive to changes in seasonality and subtropical
jet location, with different methods yielding a wide range of answers. In their review of the topic, Krishnan et al. (2019)
stated that the implications of warming on the hydrological cycle of the Hindu Kush are "not yet clear". Following Forsythe
et al. (2017), they did note that local increases in WD activity were linked to increases in glacial mass in the Karakoram
and Western Himalaya, but carefully stated the uncertainty in their summary for policymakers: "The HKH is experiencing
increasing variability in western disturbances and a higher probability of snowfall in the Karakoram and Western Himalaya,
changes that will likely contribute to increases in glacier mass in those areas. This finding runs counter to many expectations in
the scientific community, and more research is needed to understand the reasons for this and its potential future implications."
The IPCC AR6 working group 1 report (Doblas-Reyes et al., 2021) summarised the region thus: "Over most of the Hindu
Kush Himalayan region, snow cover has reduced since the early 21st century, and glaciers have retreated and lost mass since
the 1970s. The Karakoram glaciers have remained either in a balanced state or slightly gained mass. During the 21st century,
snow covered areas and snow volumes will decrease in most of the Hindu Kush Himalayan, and snowline elevations will rise
and glacier volumes will decline (high confidence). A general wetting across the whole Tibetan Plateau and the Himalaya is
projected, with increases in heavy precipitation in the 21st century."

## 7.3 Future projections

In the final section of the main body of this review, we turn our attention to the projected behaviour of WDs in future climate
scenarios. As we saw in the previous section, quantifying trends in historic WD frequency is a challenging endeavour, and one
which is compounded as we look towards the future. Here, weather reports and reanalyses are replaced with climate models
that come with their own, much larger, uncertainties. Therefore, because direct counting from bulletins and reports cannot be



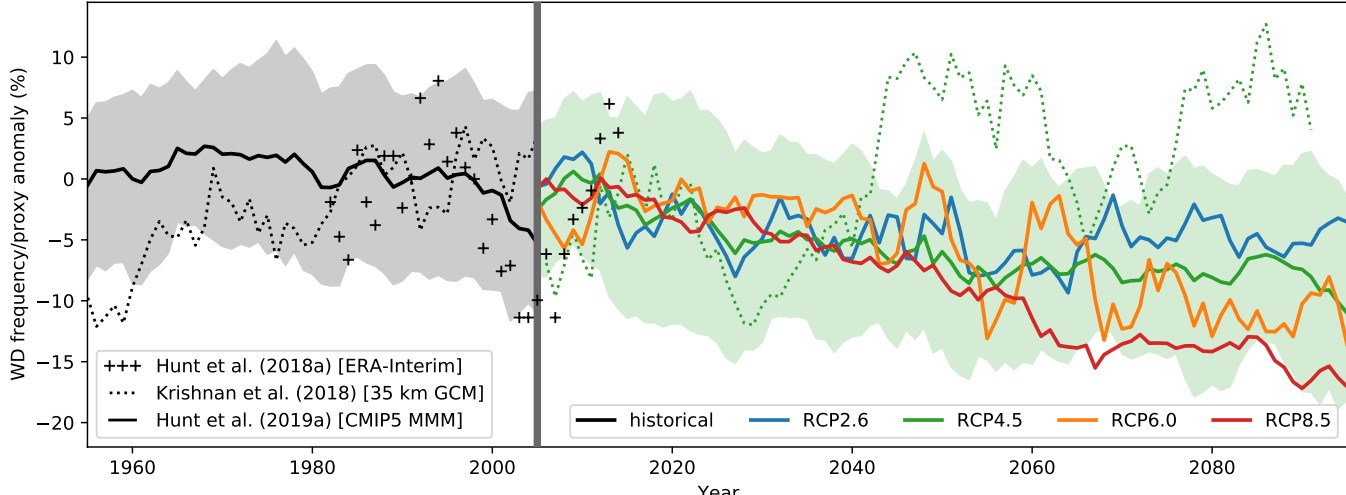

**Figure 24.** Projected future trends of WD activity under historical (black), RCP2.6 (blue), RCP4.5 (green), RCP6.0 (yellow) and RCP8.5 (red) emissions scenarios. Three studies are incorporated: Hunt et al. (2019b), who tracked WDs in 22 CMIP5 models, the multi-model means for which are denoted by solid black lines; Hunt et al. (2018b) who used the same tracking algorithm on the ERA-Interim reanalysis, denoted with crosses; and Krishnan et al. (2018), who used bandpassed geopotential variance as a proxy for WD activity applied to a single 35-km GCM, denoted by dotted lines. All values are computed as five-year running means and are given as anomalies to their respective 1980–2000 means. Colour-shaded regions indicate the inter-model interquartile ranges for the historical and RCP4.5 track frequencies in Hunt et al. (2019b).

used, and because one must either derive and apply a proxy to, or use a tracking algorithm on, climate model output, very few studies have attempted to estimate WD activity in the coming century.

Studies of global extratropical cyclone behaviour in CMIP5 future climate scenarios have suggested a poleward movement of storm tracks (Catto et al., 2011; Chang et al., 2012). Such a trend would result in a decline in WD frequency as an increasing fraction would be deflected north of the Tibetan Plateau. However, Ridley et al. (2013) disagreed with this, in a study using cluster analysis to quantify changes in WD frequency, they found WD-like regimes increased significantly in the RCP4.5 future scenario.

Earlier studies have also reported large uncertainties in projected changes to winter precipitation over the Western Himalaya,
due in part to the typically coarse resolution of CMIP5 models not resolving crucial convective and orographic processes (see Sec. 5). These uncertainties involve disagreement between individual models (Palazzi et al., 2015b) as well as high spatial inhomogeneity (Su et al., 2016). A focused study by Panday et al. (2015) found a small but significant increase in winter precipitation over the Western Himalaya in the RCP8.5 future scenario and projected that extremes would become less frequent but more intense. Either way, it is likely that snowpack depth will continue to decline due to the strong signal in increasing
temperature (Terzago et al., 2014).





### 7.3.1 Counting WDs

Prior to the review of Dimri et al. (2015), this was explored only in Ridley et al. (2013). They applied a clustering technique to output from a 25-km regional climate model forced at the boundaries by output from coarser RCP4.5 experiments conducted with the ECHAM5 and HadCM3 models. They identified four weather regimes associated with anomalous snowfall over the Karakoram and Himalaya, linking one – associated with low surface pressure and anomalous southerlies – to western disturbances. In both experiments, this cluster increased significantly in frequency over the coming century, and was associated with an increase in both seasonal snowfall and intensity of snowfall events.

Krishnan et al. (2018) employed a different proxy technique, following the earlier work of Madhura et al. (2015) in using bandpassed variance of upper-level geopotential height. They found (Fig. 24) that this metric of activity increased in a single-model RCP4.5-forced regional projection, though with a similarly strong decadal variability to that found during the previous century. They attributed this to an enhanced upper-level quasi-stationary trough over central Eurasia which occurs as a result of differential heating over the Tibetan Plateau. Also deploying a proxy method, Midhuna et al. (2023) developed a regression model for predicting precipitation over the Western Himalaya. Using a single CORDEX-SA model, they used days with precipitation exceeding 3.5 mm as a proxy for WD frequency, and seasonal precipitation as a proxy for WD intensity. They found both had insignificant declining trends across the 21st century.

To date, only one study has applied a tracking algorithm to climate model output in an attempt to quantify the future activity of WDs. Hunt et al. (2019b) applied the feature-tracking algorithm developed in Hunt et al. (2018b) to the output of 24 CMIP5 models. Compared to a 1980–2005 baseline, they found that the 2080–2100 WD multi-model mean frequency decreased by 11% and 17% in RCP4.5 and RCP8.5 experiments respectively (Fig. 24), both statistically significant and in contrast to the trend found in recent WDs in Hunt (2024). The decline in frequency was accompanied by significant declines in the WD multi-model mean intensity (12% in RCP8.5) and seasonal precipitation (15% in RCP8.5 over north India). They attributed these changes in frequency and intensity to a widening and weakening of the subtropical jet, which is subsequently associated with a decrease in meridional wind shear and baroclinic vorticity tendency, resulting in less efficient WD development. Aside from the obvious implications for water security, reduced WD frequency would likely result in worsening pollution over north India and Pakistan (Paulot et al., 2022; Xie et al., 2023).

### 7.3.2 Changing impacts

Understanding the role of WDs for both mean and extreme precipitation over the Western Himalaya and surrounding region in the face of climate change is clearly of paramount importance, given the sensitivity of the local and downstream population to changes in the many aspects of water security (e.g. Mishra et al., 2020). CMIP5- and CMIP6-class models alone are not suitable tools for this task: their horizontal resolutions – usually of order 100–200 km – are incapable of resolving either the complex orography or even the most basic of convective processes (Sec: 5.2.2). Well-simulated dynamics, a good land surface model, and a good parameterisation scheme for convection mean that the projected changes to seasonal mean precipitation may be



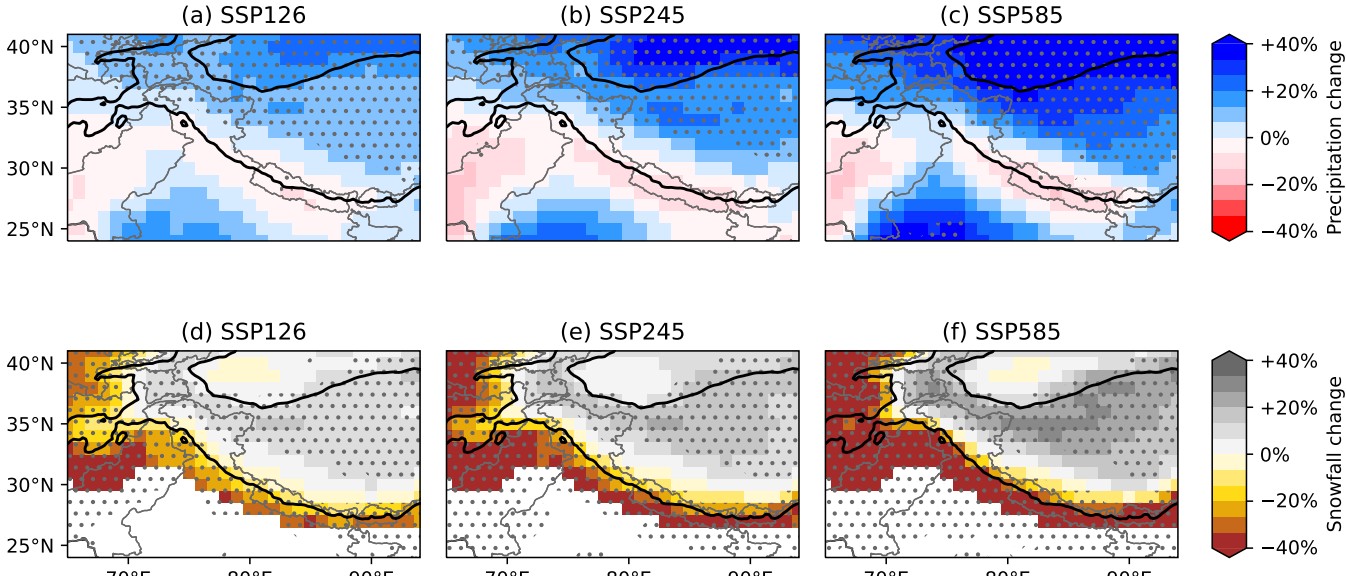

**Figure 25.** Projected change in winter (DJFM) precipitation (a–c) and snowfall (d–f) from the current climate (1960–2000) to the end of the century (2060–2100) as a function of emissions scenario. Computed using 37 CMIP6 models, each regridded to 1°×1°. Stippling indicates where at least two thirds of the models agree on the sign of change. The 2500-m surface elevation contour is denoted with a thick black line.

reasonably accurate, but this does not extend to extreme precipitation – often brought about by sub-gridscale convective storms – for which more advanced tools are needed.

Studies based on CMIP5 and CMIP6 model output have agreed that higher elevations are very likely to receive significantly more precipitation in a warmer climate, with insignificant drying in the foothills, and increased rainfall over the northern Indian plains (Almazroui et al., 2020; Banerjee and Singh, 2022). Banerjee and Singh (2022) also found a shift in precipitation seasonality, with the peak snowfall month moving from February to March by the end of the 21st century. They attributed this to changes in WD frequency. Looking at the Hindu Kush, Rahman et al. (2022) reported significantly increasing trends

for projected winter precipitation, strongest in the lower- and mid-elevations, in contrast to the recent drying they found in station-based observations for the region. Deng and Ji (2023) also found significant winter wetting of the western Tibetan Plateau in dynamically-downscaled CMIP6 experiments. This general wetting trend appears to be at odds with the projected decline in WD frequency and intensity (Hunt et al., 2019b), but these studies agree that the region that receives the heaviest WD precipitation – the foothills – is projected to undergo insignificant drying. The changes described in these studies are

summarised in Figure 25, where we use a slightly longer baseline of forty years to smooth out decadal variability.

The Coordinated Regional Climate Downscaling Experiment (CORDEX) offers an opportunity to move beyond these coarse resolutions, with the regional experiments over South Asia (CORDEX-SA) reaching horizontal resolutions of between 25 and 50 km. While this is still inadequate to resolve convection, the considerable improvement in orographic representation at these



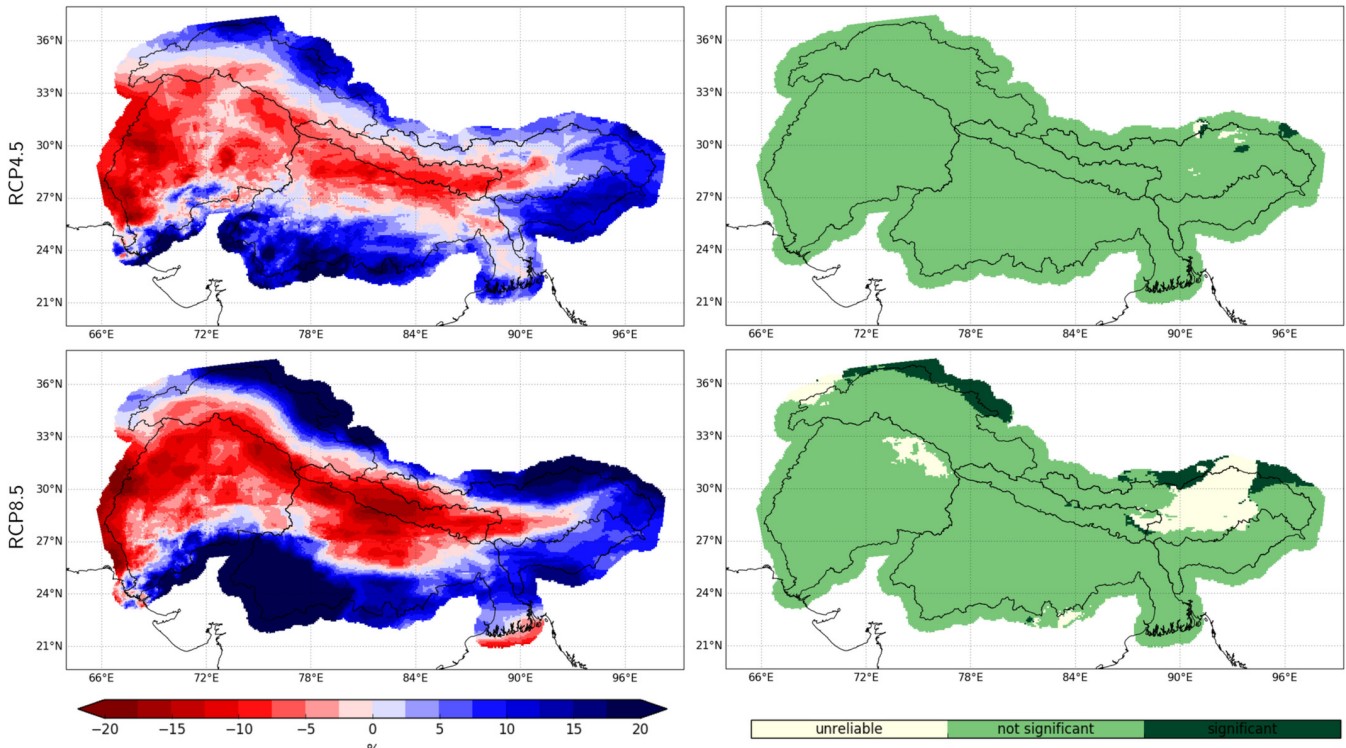

**Figure 26.** From Jury et al. (2020). Median climate change of precipitation for Dec–Apr (left column) under the RCP4.5 (top) and the RCP8.5 (bottom) scenarios at the end of the 21st century $((X_{2071--2100}/X_{1981--2010} - 1) \times 100)$ and the respective levels of agreement between the models (right column; "unreliable": $\leq 50\%$ of models show significant changes but agree with $<80\%$ on the sign of change; "not significant": $<50\%$ of models show significant changes (but $\geq 80\%$ agree on the sign); "significant": $\geq 50\%$ of models show significant changes and agree with $\geq 80\%$ on the sign of change; significance has been derived using the Wilcoxon-Mann–Whitney test). There are no areas where $<80\%$ of models agree on the sign but $\geq 50\%$ of models show significant changes.This is a multi-model ensemble comprising output from CORDEX-SA and dynamically-downscaled regional models inheriting from CMIP5-class GCMs.

resolutions has made CORDEX-SA the tool of choice for recent studies investigating the fate of winter precipitation over the Himalaya.

Jury et al. (2020) constructed an ensemble from 36 CMIP5 models and 13 CORDEX-SA regional climate models, based on their ability to reproduce circulation patterns – including WDs – and elevation-dependent warming signals (e.g. Palazzi et al., 2017, 2019). They found considerable spatial heterogeneity in projected changes to extended winter (Dec–Apr) precipitation, but noted a tendency to decrease along the foothills but increase at both higher and lower elevations (Fig. 26), a pattern similar to that recovered from simple CMIP6 analysis (Fig. 25), and a value consistent with the expected decline in WD frequency of 11–17% reported by Hunt et al. (2019b). As the heaviest precipitation falls along the foothills, it is this projected decrease that potentially has the greatest impact. However, these trends in total precipitation were not significant at any elevation, although the authors reported a strong and significant decline in snowfall across the region by the end of the century. Such results reflect



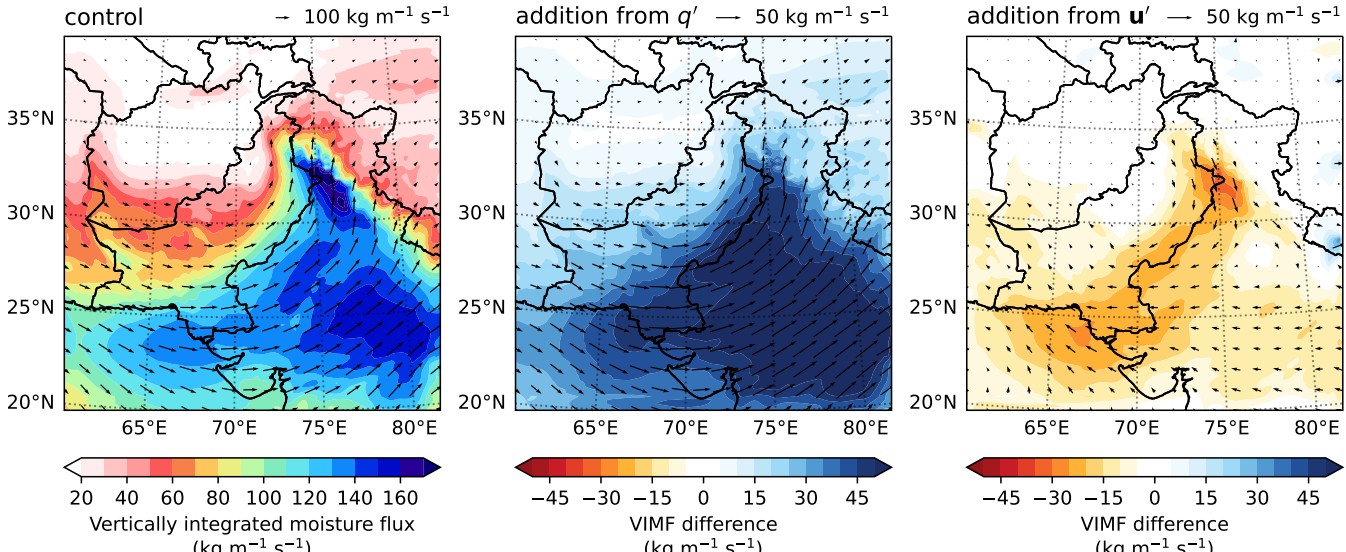

**Figure 27.** Adapted from Fig. 11 of Hunt et al. (2020), where 40 historical WDs were simulated in WRF at a resolution of 20 km. The left panel shows their composite vertically integrated moisture flux. The next two panels show the change in composite vertically integrated moisture flux from changes to tropospheric moisture content (centre) and dynamics (right) when the same WDs are simulated in an RCP8.5 PGW framework – i.e., by perturbing the boundary conditions to reflect projected changes in atmospheric and surface variables.

the fact that regardless of the overall trend, snowfall will tend to decline at the expense of increasing rainfall across the foothills and Himalayas, but will tend to increase over the Karakoram and Tibetan Plateau (Fig. 25; Rajbhandari et al., 2015; Hunt et al., 2020). Glaciated regions with a projected increase in snowfall in Fig. 25, namely much of the Karakoram, are the same regions experiencing anomalous glacier growth in the present climate (Fig. 23).

Other studies using CORDEX-SA or statistical downscaling have come to very similar conclusions (Midhuna and Dimri, 2020; Ballav et al., 2021; Midhuna and Dimri, 2022; Dimri et al., 2022; Meher and Das, 2022), typically attributing the overall increase in precipitation in the region to a strengthening of the subtropical jet and increased southerly moisture flux over the Arabian Sea. These studies only really differ in which regions they find statistically significant trends. Among these, Midhuna and Dimri (2022) and Meher and Das (2022) also reported changing seasonality, finding a relatively larger increase in mean precipitation during the late winter (February) and spring (March and April) respectively, and thus providing further evidence for a lengthening of the active WD season due to climate change. These results show that trends in mean precipitation are not particularly sensitive to model resolution and thus appear to be adequately captured in coarser CMIP5 and CMIP6 models. However, given the importance of convective-scale processes in this region, convection-permitting climate models may yet overturn this conclusion.

Although the relationship between climate change and WD frequency remains unclear, Hunt et al. (2020) used the recently-developed technique of pseudo-global warming (PGW; Kimura and Kitoh, 2007; Prein et al., 2017) to quantify the relationship between climate change and WD intensity, and the subsequent impact on WD-related precipitation. The authors isolated 40



case studies of WDs from earlier literature, simulating each in a a high-resolution (4.4 km) set up of WRF to ensure that both convection and orography were well represented. In addition to this control experiment, they also ran additional RCP4.5 and RCP8.5 experiments. In each case, the boundary conditions of the experiment were perturbed using CMIP5 multi-model means to reflect the different large-scale environments that such WDs would encounter if they developed within those respective

climates. They found significant increases in net WD-related precipitation in the future climate scenarios, despite a weakening of WD intensity. For example, in their RCP8.5 composite (Fig. 27), they found that WD-induced moisture flux incident on the Western Himalayas was reduced 20% through weakened WD circulation, but increased 28% through increased moistening of the tropical atmosphere. They also found significant decreases in the return periods (i.e. significant increases in the frequencies) of extreme WD-related precipitation events in both RCP4.5 and RCP8.5 future scenarios.

These studies all lean towards a wet-dry-wet trend with increasing elevation in the Western Himalaya, as shown in Fig. 25. The wetting trend to the south is mostly over the southern IGP and northern Deccan Plateau, where WDs provide little precipitation. Over the foothills, where the mean winter precipitation is greatest, there is a projected drying trend. This drying is perhaps related to the projected decline in WD frequency and intensity reported in Hunt et al. (2019b), but most studies find that it is statistically insignificant. At higher elevations, across much of the western and central Himalaya, there is a pro-

jected increase in precipitation, likely driven by a combination of elevation-dependent warming and the Clausius-Clapeyron relationship[5]. There is also a nonlinear component (Baudouin et al., 2021) whereby WDs are intensified through enhanced moist baroclinic instability associated with the orography, i.e., changes in specific humidity lead to local circulation changes. This pattern underscores the complex nature of precipitation trends over the winter Himalaya, where it is likely that large scale circulation is weakening the transport of moisture (Fig. 27c) and this is counterbalanced by increasing specific humidity. The

uncertainty that remains lies in determining which of the two processes will win, and where.

## 8    Future research questions and challenges

Despite the large volume of research on WDs, a number of important questions remain about their behaviour and impacts – either because they have not yet been investigated or because studies have arrived at divergent results. In this section we summarise these important unknowns, based on the discussions throughout this paper. For each entry, the relevant section or

subsection is given in square brackets.

1. What is the sensitivity of WD tracks and their statistics to the choice of reanalysis? This could be explored between reanalyses or by using multi-member reanalyses (e.g. the 10-member ERA5 ensemble). [Tracking (Sec. 2.2)]

2. Do WDs occur in clusters, arriving in rapid serial succession with longer gaps between groups? If so, why? This behaviour has been found in extratropical cyclones, e.g. over the North Atlantic (Priestley et al., 2017; Dacre and Pinto,

2020). Such work would need to overcome the limitations of tracking algorithms, where WDs in rapid succession may not be easily separable, degrading the quality of detection. [Tracking (Sec. 2.2)]

---

[5]Since a warmer tropical atmosphere can hold more moisture (about 7% $K^{-1}$ at saturation), in the absence of a large-scale circulation change, we would expect a robust increase in winter precipitation.





3. Can WDs merge or split? It is certainly possible given the behaviour of similar PV streamers and cutoffs (Kew et al., 2010; Portmann et al., 2021) as well as the discussion in Cannon et al. (2016). How would such events affect associated precipitation? As in the previous question, this work would need to overcome the limitations of current tracking algorithms. [Tracking (Sec. 2.2)]

4. What is the relationship between the different regions of WD cyclogenesis and their downstream behaviour and impacts? Do those forming away from the subtropical jet have inherently different characteristics? Are there different WD cyclogenesis mechanisms beyond simply dynamical (baroclinic) instability? [Cyclogenesis (Sec. 2.3.1)]

5. Do conditional instability of the second kind (CISK; heating rate proportional to relative vorticity) or wave-CISK (heating rate proportional to vertical velocity) instabilities play a role in the PV-convection relationship as WDs intensify near the Himalayan orography? This could be complementary to the baroclinic instability mechanism proposed by Baudouin et al. (2021). [Intensification (Sec. 2.3.2)]

6. Why do some WDs contain frontal regions while others do not? This is one of the major characteristics separating WDs from extratropical cyclones, which almost invariably contain frontal structures (Browning, 1999; Catto, 2016; Schemm et al., 2018). Such analysis would benefit from front-centered composites, e.g., using gradients in equivalent potential temperature (e.g. Dacre et al., 2012), which has not yet been carried out for WDs. [Structure (Sec. 2.3.4)]

7. Related to the previous question, some case studies have identified that bands of very high vorticity can develop near the surface during the passage of WDs, stretching from the Sulaiman Range in central Pakistan to the Himalayas. What causes these bands, and what are their impacts? [Structure (Sec. 2.3.4)]

8. A rigorous quantification of each of the moisture sources of WD precipitation has not yet been performed. Even though it is known the majority comes from the neighbouring Arabian Sea, what fractions are carried over longer distances from the Mediterranean, Black, or Caspian Seas? Do different ratios affect the resulting precipitation? How sensitive are these ratios to WD characteristics such as latitude? Such research could build on the comprehensive case studies by Boschi and Lucarini (2019). [Moisture sources (Sec. 2.4)]

9. WDs vary considerably in their intensity and impacts. What is the best way of classifying or categorising them? Should we base a classification on WD impacts (e.g., precipitation or latitude), or WD dynamics (e.g., upper-level vorticity or wind speed)? [Variability (Sec. 2.6)]

10. Are there differences between WDs that arise as PV cutoff lows and those that arise as dynamical instabilities in the subtropical jet? [Variability (Sec. 2.6)]

11. What are the primary causes of intraseasonal variability in WD frequency and other characteristics? [Variability (Sec. 2.6)]

12. Neal et al. (2020) showed that weather regimes associated with WDs also lead to increased winter precipitation over northeast India. It is not known whether this is caused by WDs propagating into the region, instability caused by the position of the jet, or some other weather phenomenon. [Precipitation (Sec. 3)]



13. How can better knowledge of WDs translate to decisions around water management? This is a multidisciplinary topic which also requires better observations, operational prediction, forecast skill assessment and decision-support tools. [Water security (Sec. 3.3) and present climate (Sec. 7.2)]

14. What are the socioeconomic losses associated with historic WD activity? How are these best quantified? What kind of WD characteristics are associated with them? [Hazards (Sec. 3.4 and water security (Sec. 3.3)]

15. Which WD characteristics, such as location, intensity and timing, is associated natural hazard occurrence most sensitive to? It is likely that this will be different for different hazards. [Hazards (Sec. 3.4)]

16. Coldwaves have thus far only been anecdotally linked to WDs. Is there a climatological link between the two? If so, which is more important: northerly advection of cold air in the rear flank or sustained heavy cloud cover? [Coldwaves (Sec. 3.4.2)]

17. What role do WDs play in driving nor'westers (storms that impact northeast India and Bangladesh), if any? Understanding this link could be crucial to reducing lightning-related mortalities in this region. [Lightning (Sec. 3.4.3)]

18. Do WDs occurring during the summer monsoon have different characteristics than typical winter WDs? 'Monsoonal' WDs have access to an abundant moisture supply that is not present in the winter months, but does this significantly affect their structure or evolution? There is also evidence that WDs occurring during the summer arise from high-latitude PV cutoff lows, rather than arising as instabilities within the subtropical jet. [Summer monsoon (Sec. 3.5)]

19. Is the seasonality of WDs changing, as Valdiya (2020) suggests? [Interaction with summer monsoon (Sec. 3.5) and present climate (Sec. 7.2)]

20. More work is needed to establish the link (if any) between WD characteristics and slowly-varying teleconnections. Many of these have been investigated in detail in previous studies, but the connection with the MJO, the SAM, PDO, IOD, and MJO are still not well understood. These would in turn improve seasonal forecasts. [Large-scale forcing and teleconnections (Sec. 4)]

21. The Pamir bifurcation occurs when the subtropical jet passes near the Pamir mountains. These mountains act like a wedge, deflecting WDs south into India or north into central Asia with only small variation in jet latitude. How can the impact of the Pamir bifurcation be reduced in forecasts? Is there a critical subtropical jet latitude at which forecast uncertainty increases significantly? [Forecasting (Sec. 6)]

22. It would be beneficial to conduct quantitative analysis on the quality of IMD warning bulletins so that stakeholders know at what lead times and in which locations they are most reliable. [Short-range forecasting (Sec. 6.2)]

23. More quantitative research is also needed on extended range and seasonal WD forecasting. This includes, but is not limited to, quantification of track, frequency, and intensity errors. [Extended-range forecasting (Sec. 6.4)]





24. Aside from better understanding of teleconnections, how might subseasonal and seasonal prediction of WD activity be improved? [Extended-range forecasting (Sec. 6.4)]

25. Studies of western Himalayan paleoclimate over the last ten millennia often disagree considerably on periods of surplus or deficient winter precipitation (and hence WD activity). To what extent is this due to contamination from the summer monsoon signal, and to what extent can it be explained by the large spatial variability in climatic trends of Himalayan precipitation? [Paleoclimate (Sec. 7.1)]

26. What is the relationship between observed elevation-dependent warming and changes in WD behaviour? [Present climate (Sec. 7.2)]

27. What role do aerosols play in the moist thermodynamics of WDs? Are regions or months of high aerosol optical depth associated with increased WD precipitation? How have observed trends in different aerosol species affected WD activity? [Present climate (Sec. 7.2)]

28. What is the impact of climate change on WD intensity, and why? [Present climate and future climate (Secs. 7.2 and 7.3)]

29. What role have WDs played in supporting or destroying the Karakoram anomaly in the last century? How is this likely to change over the coming century? [Present climate and future climate (Secs. 7.2 and 7.3)]

30. Only two studies have investigated in detail how WD behaviour will change in future climate scenarios, and they are in disagreement. More refined studies, looking at a broad range of WD characteristics, are needed to resolve this. [Future climate (Sec. 7.3)]

31. Which has the greater impact on trends in winter precipitation over the western Himalaya and surround region – changes to thermodynamics, or changes in atmospheric circulation? [Future climate (Sec. 7.3)]

32. There is also only a weak consensus on the projected future decrease of winter precipitation in the western Himalaya. Studies leveraging higher-resolution models that are capable of resolving orographic feedbacks are needed to make more robust estimates of these changes. [Future climate (Sec. 7.3)]

For more open research questions relating to the hydroclimate of the Indus region, the reader is encouraged to read Orr et al. (2022).

# 9 Summary

Western disturbances (WDs) are upper-tropospheric lows that travel eastward along the subtropical westerly jet in the northern hemisphere. WDs have a positive vorticity anomaly that extends through much of the troposphere, with a maximum at about 350 hPa, and are often associated with surface lows. They are most common during the Boreal winter months (December to March), during which they bring heavy precipitation to the mountain ranges of Pakistan, north India, northern Afghanistan and





Tajikistan: the Western Himalayas, the Karakoram, the Hindu Kush, and the Pamirs. As a result, they are a crucial element of the water security of both the Indus and Ganges river basins. This paper reviews the large volume of recent research on WDs. That research has leveraged a wealth of novel computational, modelling, and experimental techniques that have significantly improved our understanding of WD behaviour and impacts, making this updated review necessary.

The recent development of a tracking algorithm for WDs, and its subsequent application to reanalyses and climate model output, has allowed the production of WD track catalogues. These catalogues have been a key step in building robust WD statistics, such as their climatology and variability – with the total number of WDs per year varying between about 45 and 65, and the number of strong (also referred to as 'active') WDs per year varying between 15 and 25.

Similarly, robust links to moisture supply, large-scale forcings, and natural hazards have been established through the use of WD catalogues. Moisture trajectory and isotopic analyses have revealed the dominant role of the Arabian Sea as a moisture source for typical WDs; although extreme WD rainfall is typically produced through interaction with elements of the summer monsoon.

Forcing of WD activity by teleconnections remains relatively unexplored, with the exceptions of the NAO and ENSO. Studies have demonstrated the NAO can vary monthly WD frequency by about 20% through upstream modulation of the subtropical jet. Studies have not been able to agree on the role of ENSO, although it has been proposed that different flavours of ENSO, such as El Niño Modoki, may be important. The reason for the ENSO teleconnection, and the variability of its strength, need further investigation

Climatological studies have now linked WDs to increased risk of landslides, fog, and extreme precipitation events. They have not yet been statistically linked to an increased risk of coldwaves or avalanches, although a significant body of case studies exists to support these claims.

High resolution models can now explicitly resolve certain convective and local orographic processes. Recent studies have shown that accurate representation of these processes and their interaction with each other is vital for the accurate simulation of heavy precipitation events associated with WDs. Operational forecasts are not typically run at these fine scales, thus accurate predictions of hazardous WD precipitation still present a challenge.

A comparison of 16 studies revealed that trends in WD frequency over the last fifty years depend strongly on the choice of study region and season. These studies used a range of valid techniques to estimate WD frequency, such as counting them in weather bulletins, tracking them in reanalyses, and using proxy variables such as bandpassed geopotential variance. Trends in WD frequency tend to be positive if measured over the Hindu Kush, Karakoram and north Pakistan, and tend to be negative elsewhere, although some studies disagree. Crucially, there is a strong consensus that WD frequency has increased significantly in the spring (March to June).

There is, as yet, insufficient evidence to determine whether future climate change will increase or reduce WD frequency as the two key studies investigating this disagree with each other. However, the study projecting a decline noted that the decline was consistent and significant across a range of CMIP5 models, with the multi-model mean projecting declines in WD frequency and intensity of 15% and 12% respectively by the end of the 21st century.





Future projections of winter precipitation only weakly agree on a decline over the foothills of the Western Himalayas, but agree on a significant increase in precipitation at higher elevations. The proposed causes of these changes include, but are not limited to, poleward movement of the subtropical jet, bringing WDs to higher latitudes, and an increase in moisture flux associated with WDs brought about by the greater moisture-carrying capacity of a warmer troposphere.

So despite significant progress having been made in the research and understanding of WDs over the last decade, many important challenges remain. Of these, two are particularly important. Firstly, accurate forecasts of their tracks and associated precipitation are hindered by the complex interactions with orography and convective processes. Secondly, there is still no consensus on the historical or future response of WD activity to climate change, partially due in the former to inconsistent definitions of WDs, and partially due in the latter to insufficient model resolution. The likely pathway to resolving both of these challenges is high-resolution convection-permitting models, both in operational forecasting and climate science.

### 9.1 A definition of western disturbances

The formal definition of WDs, published by the IMD, is "Weather disturbances noticed as cyclonic circulation/trough in the mid and lower tropospheric levels or as a low pressure area on the surface, which occur in middle latitude westerlies and originate over the Mediterranean Sea, Caspian Sea and Black Sea and move eastwards across north India." Much of the research presented in this review shows that this definition[6] is now outdated, and has led to authors using their own criteria. To aid consistency and reduce ambiguity in future research, we synthesise the research discussed in Secs. 2 and 3 to create a new, up-to-date, unified definition of WDs.

Western disturbances are eastward-travelling upper-level troughs passing over South Asia that arise as instabilities on the jet, or as extratropical PV cutoff lows. By drawing moisture primarily from the Arabian Sea, they bring precipitation, and sometimes other impacts, primarily to the western Himalaya, Karakoram, Hindu Kush, but also to the surrounding regions of Pakistan and north India. They can occur at any time of year, but are strongest and most frequent between December and May.

*Data availability.* The WD track data used for summary figures throughout are available from https://zenodo.org/records/8208019.

*Author contributions.* KMR Hunt and AP Dimri conceptualised the paper. KMR Hunt conducted the literature review, wrote the initial manuscript draft, and prepared most of the figures. JP Baudouin and AG Turner provided extensive detailed feedback throughout, leading to substantial improvements in structure, fine detail, and figures. JP Baudouin also helped with the literature review. G Jeelani helped to write the section on moisture sources. Pooja and E Palazzi reviewed the draft manuscript. R Chattopadhyay provided the outline for the subseasonal forecasting section and the data used for Fig. 18, as well as helpful feedback on parts of the manuscript. F Cannon provided useful early feedback on the structure and water security sections. Arulalan T provided the data for Fig. 17 and helped with the development

---

[6]The IMD also define a "western depression", as a strong WD that has two or more closed isobars in surface pressure. That term has not entered common use and has become conflated with "active WD", which is the term we recommend for systems associated with heavy precipitation.





of this figure. MS Shekhar provided useful feedback on parts of the manuscript relating to precipitation impacts. Sabin TP provided some of
data needed for Fig. 24.

*Competing interests.*    The authors declare no competing interests.

*Acknowledgements.*    KMRH and AGT were funded in part through the Weather and Climate Science for Service Partnership (WCSSP) India,
a collaborative initiative between the Met Office, supported by the UK Government's Newton Fund, and the Indian Ministry of Earth Sciences
(MoES). KMRH is now supported by a NERC Independent Research Fellowship (MITRE; NE/W007924/1). AGT is additionally supported
by the National Centre for Atmospheric Science through the NERC National Capability International Programme Award (NE/X006263/1).
JPB would like to acknowledge financial supports from the German Federal Ministry of Education and Research (BMBF) which funded
the PalMod project (subproject no. 01LP1926C) KMRH wishes to thank Dr Ashis Kumar (Aryabhatta Research Institute of Observational
Sciences) for providing the data for Fig. 6.



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
