# Peer review of "Western disturbances and climate variability: a review of recent developments"

_EGUsphere, 2024_

## Referee Comment (RC4)

**Review on "Western disturbances and climate variability: a review of recent developments" by Kieran M. R. Hunt et al. 2024**

Thank you for giving me the opportunity to assess this comprehensive and well-written review article about western disturbances. I focused on the aspects related to moisture transport and isotope studies, which I found very well-summarized and inspiringly written. I have no problem with length since the well-thought-through structure helps the reader to orient efficiently. I have only few minor comments, mainly small questions on some phrasings.

**Abstract**:

1) L. 7: "Recent studies…" -> what is the time period covered in this review?
2) L. 8: "novel analysis techniques: mention automated tracking capabilities explicitly? Since this is mentioned in Section 2 as one of the key changes since the last review on the topic in 2015.

**Introduction:**

3) Fig. 1: add a black contour for topography? it could help to link to other Figs (such as Fig. 2).
4) L. 87: "a westward moving synoptic-scale trough" -> eastward-moving?
5) L. 136: "inexpensive" sounds a bit inaccurate to me, the simulations are just getting comparably less expensive but high-resolution model simulations are still very expensive in terms of computational costs.
6) L. 150: "rather **than**"
7) L. 159: "observed responses **in** the instrumental record". If it's really "observed responses to the instrumental record" that the authors mean, I don't understand the sentence.

**Section 2:**

8) In my opinion Section 9.1 would be better placed in this section or after Section 3.
9) L. 179: put references in parentheses
10) L. 179-182: so combining early reanalyses with WD track data has been done already before 2015, right? Maybe put this sentence before the important remark of the turning point around 2015 with the start of automated tracking algorithms to keep the story chronological.
11) L. 195: rephrase the first sentence: yes, of course, detection depends on detection but can you say more? Detection depends on the characteristics of interest and may therefore vary among algorithms?
12) L. 198: which characteristics are meant here? I would be careful when using characteristics because you seem to differentiate between characteristics i.e. properties of WDs in terms of circulation vs. impacts, i.e. surface weather-related hazards.
13) L. 252: what is a weather distinct weather regime?
14) Section 2.2.: I like the bottom-up vs. top-down approach and my reading would benefit from a short introduction of these two approaches and what is meant by

it at the beginning of the section. Is it event-based case studies (bottom up) vs. climatological composite analysis using tracking algorithms (top-down)?

15) Are WDs included in existing global climatologies of extratropical cyclones and cyclone-related features (i.e. WCBs)? E.g. in Wernli and Schwierz 2006 or Madonna et al. 2014.

16) L. 267: IMD has not been introduced as an abbreviation yet

17) L. 275: that seems also by design of most detection schemes since WDs are identified as eastward travelling, resp. the (probably very rare) westward travelling WDs are ignored?

18) L. 276: "also because any system propagating eastward…" -> you mean westward here, right?

19) L. 425: why "above"?

20) L. 427: by compositing you mean an Eulerian analysis of the typical circulation associated with WDs and their environment? I think this ought to be clarified. Moisture sources based on trajectory-based diagnostics can also be composited using on a series of precipitation events or WD events.

21) L. 430: cite Dansgaard 1964 when defining the deuterium excess

22) L. 432: the delta values are deviations of the mentioned ratio from a commonly agreed-upon standards representing ocean water. How about writing "where d2H is derived from the ratio of deuteriated water to the most abundant H216"? Also delta18O should be **\delta**.

23) L. 432: both expressed as a deviation of the isotope ratio from a standard reference representing the isotope ratio of the mean ocean water.

24) L. 425: missing space before reference

25) L. 435: this is not entirely correct: see Thurnherr et al. 2020 for a study of ship-based measurements of dexcess in oceanic regions with different SSTs. I think for the regional setting in this paper, one important point that can be made is that the deuterium excess shows different signatures for water vapour that has undergone continental recycling vs. originates from oceanic source regions. And, furthermore, isotope signals can help partitioning land-derived sources into soil evaporation and plant transpiration (see e.g. Aemisegger et al. 2014). Rather than a reliable measure of the moisture source location, isotopes are a tracer of moisture source conditions (i.e. processes that characterise the source).

26) L. 455: majority ->major?

27) L. 462: Here maybe a short statement on trajectory-based moisture source detection algorithms could be made. I.e. different techniques exist including Eulerian and Lagrangian approaches with each having their own specific limitations.

28) L. 490: calling for Eulerian moisture tracking methods (with numerical tracers for different sources) to be used in future studies?

**Section 7**

29) L. 1451: **of** the subtropical jet

30) L. 1453: and corroborates

31) L. 1485: misspelling of precipitation

32) L. 1502: remove one that

33) L. 1540: **E**levation-dependent warming

**Section 8:**

34) L. 1750: "… each of the moisture sources…": why "each"? Does it imply that they are a priori clear? I would remove "each".
35) L. 1751: "… it is known **that**…"
36) L. 1752: "Do different ratios… -> source ratios?
37) L. 1756: is latitude really a WD impact? Or a WD property?
38) L. 1814: higher resolution than what? -> high resolution

**Summary:**

39) I think the concluding section should be more than just a summary. It should put the review into the current context. And with this in mind, it reads a bit strange to come back to the summary after the future research questions and challenges. One could imagine having one big Conclusion and summary section, which includes Section 8 about "Future research questions and challenges".
40) Section 9.1 is out of place in my opinion. This is a sort of glossary remark or definition that should probably be placed much earlier in the paper. After the summary it comes very much as a surprise to me.
41) I must emphasise that I really like Section 8, it is very inspiring, and I would have liked to finish reading the paper in this opening-view.
42) L. 1876: repetition of primarily

---

## Author Comment (AC1)

Review on "Western disturbances and climate variability: a review of recent developments" by Kieran M. R. Hunt et al. 2024

Thank you for giving me the opportunity to assess this comprehensive and well-written review article about western disturbances. I focused on the aspects related to moisture transport and isotope studies, which I found very well-summarized and inspiringly written. I have no problem with length since the well-thought-through structure helps the reader to orient efficiently. I have only few minor comments, mainly small questions on some phrasings.

We would like to thank the reviewer for their positive assessment of our manuscript, and for their detailed comments, which we respond to point-by-point in red below. Planned revisions to our manuscript will be highlighted in blue.

**Abstract**:

1) L. 7: "Recent studies..." -> what is the time period covered in this review?
Approximately ten years (i.e. since Dimri et al, 2015). We will add this in the revised abstract.

2) L. 8: "novel analysis techniques: mention automated tracking capabilities explicitly? Since this is mentioned in Section 2 as one of the key changes since the last review on the topic in 2015.
Thanks, this a good idea and we will include this in the revised abstract.

**Introduction:**

3) Fig. 1: add a black contour for topography? it could help to link to other Figs (such as Fig. 2).
Thank you for the suggestion. Given that this is already a very complicated figure, orography is already displayed using filled coloured contours, and there are already two sets of lines (rivers and national borders), we don't think this would improve its clarity. However, following a suggestion from reviewer 2, we will add state borders to Fig 2.

4) L. 87: "a westward moving synoptic-scale trough" -> eastward-moving?
Correct – thanks for spotting this typo. We will correct this to eastward-moving in our revised manuscript.

5) L. 136: "inexpensive" sounds a bit inaccurate to me, the simulations are just getting comparably less expensive but high-resolution model simulations are still very expensive in terms of computational costs.
We agree and will rephrase this accordingly: "Firstly, recent studies are making increased use of high-resolution models, which are becoming cheaper to run,

both for regional climate modelling and numerical weather prediction."

6)     L. 150: "rather **than**"
Agreed – we will fix this.

7)     L. 159: "observed responses **in** the instrumental record". If it's really "observed responses to the instrumental record" that the authors mean, I don't understand the sentence.
The reviewer's suggestion is correct. We will fix this typo.

**Section 2:**

8)     In my opinion Section 9.1 would be better placed in this section or after Section 3.
We agree, and has been requested by other reviewers. In our revised manuscript, we will move Sec 9.1 to a new Sec 3.6.

9)     L. 179: put references in parentheses
Thanks for spotting this, we will fix these.

10)     L. 179-182: so combining early reanalyses with WD track data has been done already before 2015, right? Maybe put this sentence before the important remark of the turning point around 2015 with the start of automated tracking algorithms to keep the story chronological.
Our phrasing was a bit misleading here. Mohanty et al (1998, 1999) used an Eulerian approach and showed simply that a passing WD was well represented. The first automated WD tracks didn't appear until Cannon et al (2015). We will clarify this in the revision.

11)     L. 195: rephrase the first sentence: yes, of course, detection depends on detection but can you say more? Detection depends on the characteristics of interest and may therefore vary among algorithms?
Yes, we will add this suggested clarification.

12)     L. 198: which characteristics are meant here? I would be careful when using characteristics because you seem to differentiate between characteristics i.e. properties of WDs in terms of circulation vs. impacts, i.e. surface weather-related hazards.
Thank you for this suggestion. The original sentence was: "Before 2015, Dimri (2013) used bandpass-filtered precipitation and outgoing longwave radiation (OLR) to build a composite analysis that first defined the characteristics and atmospheric circulation of a mean WD that lead to precipitation." We will revise this to: "Before 2015, Dimri (2013) used bandpass-filtered precipitation and outgoing longwave radiation (OLR) to build a composite analysis that first showed WDs leading to heavy precipitation were associated with strong southwesterly

moisture flux and deep convection."

13) L. 252: what is a weather distinct weather regime?
This was a typo – we will correct this to read "distinct weather regimes".

14) Section 2.2.: I like the bottom-up vs. top-down approach and my reading would benefit from a short introduction of these two approaches and what is meant by it at the beginning of the section. Is it event-based case studies (bottom up) vs. climatological composite analysis using tracking algorithms (top-down)?
Thank you. We will add the following text at the beginning of the section: "These can be broadly categorised into top-down and bottom-up approaches. Bottom-up approaches include case studies and Eulerian composites, often starting with the impacts of WDs and then working `upwards' to quantify the characteristics that drive these impacts. Top-down approaches use a prescribed WD characteristic (e.g., vorticity), often combined with Lagrangian compositing, then working `downwards' to quantify surface impacts."

15) Are WDs included in existing global climatologies of extratropical cyclones and cyclone-related features (i.e. WCBs)? E.g. in Wernli and Schwierz 2006 or Madonna et al. 2014.
Thank you for this interesting question. WDs do appear in these catalogues, though to a lesser extent in the W&S climatology, as they use SLP as their detection metric, and many WDs have very weak surface pressure signatures. We will include this near the beginning of our revised section 2.2: "Tracking has been done successfully for, e.g. extratropical cyclones (Dacre et al., 2012) and monsoon depressions (Hurley and Boos, 2015). In fact, depending on the detection criteria used, WDs appear in global climatologies of extratropical cyclone tracks (Wernli and Schwierz, 2006) and their features (e.g., warm conveyor belts Madonna et al., 2014). However, only recently have authors started to track WDs in reanalysis data."

16) L. 267: IMD has not been introduced as an abbreviation yet
Thank you – we will add that here.

17) L. 275: that seems also by design of most detection schemes since WDs are identified as eastward travelling, resp. the (probably very rare) westward travelling WDs are ignored?
Yes, this is a good point. While it is true that eastward movement is often specified as a tracking criterion, we have found that the final output is very insensitive to this choice. This is because flow configurations that advect upper- and mid-tropospheric vortices southward or westward in this region are extremely rare. We will clarify this in our revision: "All WDs originate from regions to the west of the Western Himalaya, or occasionally spin up *in situ*. This occurs largely because most WDs are advected along the subtropical jet, but also

because the flow configurations to advect systems westward or southward into the region virtually never occur."

18) L. 276: "also because any system propagating eastward…" -> you mean westward here, right?
Yes indeed – thank you for spotting this.

19) L. 425: why "above"?
This meant "in the text above", but we appreciate it is confusing and will remove it.

20) L. 427: by compositing you mean an Eulerian analysis of the typical circulation associated with WDs and their environment? I think this ought to be clarified. Moisture sources based on trajectory-based diagnostics can also be composited using on a series of precipitation events or WD events.
Yes, this is a good point – we indeed mean Eulerian and will clarify this.

21) L. 430: cite Dansgaard 1964 when defining the deuterium excess
Thank you for the suggestion, we will add this reference.

22) L. 432: the delta values are deviations of the mentioned ratio from a commonly agreed-upon standards representing ocean water. How about writing "where d2H is derived from the ratio of deuteriated water to the most abundant H216"? Also delta18O should be **\delta**.
Thank you, we will make these changes.

23) L. 432: both expressed as a deviation of the isotope ratio from a standard reference representing the isotope ratio of the mean ocean water.
Thank you, we will make this correction.

24) L. 425: missing space before reference
Thanks, will fix.

25) L. 435: this is not entirely correct: see Thurnherr et al. 2020 for a study of shipbased measurements of dexcess in oceanic regions with different SSTs. I think for the regional setting in this paper, one important point that can be made is that the deuterium excess shows different signatures for water vapour that has undergone continental recycling vs. originates from oceanic source regions. And, furthermore, isotope signals can help partitioning land-derived sources into soil evaporation and plant transpiration (see e.g. Aemisegger et al. 2014). Rather than a reliable measure of the moisture source location, isotopes are a tracer of moisture source conditions (i.e. processes that characterise the source).
Thank you for this information. We will revise this statement, hopefully capturing the reviewer's advice as intended: "D-excess shows different signatures for

different moisture sources – ocean evaporation, soil evaporation, and plant transpiration (Aemisegger et al., 2014). D-excess also tends to be higher in atmospheric water vapour that has evaporated from surfaces in less humid climates, and thus precipitation arising from moisture originating from different basins can have different D-excess values (Pfahl and Sodemann, 2014), but it can be hard to disentangle this signal from other drivers (Thurnherr et al., 2020). For example, the Mediterranean has D-excess values of around 22‰, much higher than the global average of 10‰ (Gat and Carmi, 1970; Natali et al., 2022) and higher than the Arabian Sea (Jeelani et al., 2017; Jeelani and Deshpande, 2017)."

26) L. 455: majority ->major?

Thank you for the suggestion. We do mean "majority" here, as it can be used as an adjective in this way.

27) L. 462: Here maybe a short statement on trajectory-based moisture source detection algorithms could be made. I.e. different techniques exist including Eulerian and Lagrangian approaches with each having their own specific limitations.

Thank you for the suggestion. Note that the trajectory-based approaches discussed here are Lagrangian – it is the moisture flux composites that are Eulerian. We will clarify this in the revision.

Slightly later in the section, we discuss the relative advantages and disadvantages of each technique: "Both isotope and trajectory methods are useful, but each have shortcomings that mean it is better to draw on results from both methods where possible. Trajectory methods give more precise results for moisture sources, and along-trajectory statistics like parcel humidity can be computed. However, large uncertainties can arise from the representation of orographic and boundary layer processes, both of which are crucial ingredients for WD precipitation. Indeed, the evaporative processes that increase parcel humidity are subgrid processes that are not necessarily well simulated in reanalyses. Further to this, trajectory calculations can be computationally expensive as large ensembles are required to reduce uncertainty. Isotope methods can therefore provide more accurate estimates of moisture partitioning, since they do not depend on small-scale processes being adequately represented by a reanalysis model. However, long time series are required to ensure a representative contribution from all moisture sources. Results from studies that depend on a single year of data, e.g. Lone et al. (2020) and Dar et al. (2021), must therefore be taken cautiously. Further work is needed with isotopic methods to better distinguish between Mediterranean and local recycling sources, both of which are associated with high D-excess (>20‰). Composite moisture flux analysis is more robust to orographic and subgrid processes, but by definition does not describe the entire distribution of possible sources, as back-trajectories can."

28) L. 490: calling for Eulerian moisture tracking methods (with numerical tracers for different sources) to be used in future studies?
Yes, we will add this in our revision: "Indeed, the evaporative processes that increase parcel humidity are subgrid processes that are not necessarily well simulated in reanalyses, and perhaps call for Eulerian moisture tracking methods with numerical tracers for different sources to be used in future studies."

**Section 7**

29) L. 1451: **of** the subtropical jet
Thank you, will fix.

30) L. 1453: and corroborates
Thank you, will fix.

31) L. 1485: misspelling of precipitation
Thank you, we will fix this and the other three(!) instances.

32) L. 1502: remove one that
Thank you, will fix.

33) L. 1540: **E**levation-dependent warming
Thank you for spotting this. Following comments from an earlier reviewer we are going to remove these paragraph headings.

**Section 8:**

34) L. 1750: "… each of the moisture sources…": why "each"? Does it imply that they are a priori clear? I would remove "each".
OK, we will remove this.

35) L. 1751: "… it is known **that**…"
Thanks, will fix.

36) L. 1752: "Do different ratios… -> source ratios?
Yes, we will add this.

37) L. 1756: is latitude really a WD impact? Or a WD property?
This is a good point, we will amend this in the revision: "Should we base a classification on WD impacts (e.g., precipitation), characteristics (e.g., latitude), or dynamics (e.g., upper-level vorticity or wind speed)?".

38) L. 1814: higher resolution than what? -> high resolution
Thanks, good point – we will fix this.

**Summary:**

39) I think the concluding section should be more than just a summary. It should put the review into the current context.
Thank you for the suggestion. As suggested by you and other reviewers, we will reframe the conclusions – including a table (below) that lists key statements and the confidence espoused in them by the literature.

| Statement | Confidence | Section |
|---|---|---|
| Tracking algorithms are a useful tool for understanding WDs. | high | 2.2 |
| WD cyclogenesis mostly occurs over ocean or downstream from mountain ranges. | medium | 2.3.1 |
| WDs intensify through baroclinic instability, sometimes with moist or orographic coupling. | very high | 2.3.2 |
| WDs primarily affect the Western Himalaya and surrounding mountain ranges. | very high | 2.3.3 |
| WDs have mid- to upper-tropospheric vorticity maxima with ascent ahead of their centre. | very high | 2.3.4 |
| The Arabian Sea is the primary moisture source for WD precipitation. | high | 2.4 |
| WDs are most frequent between December and March but can occur at any time of year. | high | 2.5 |
| There is large variance in most WD characteristics, such as lifetime, intensity, and latitude. | high | 2.6 |
| WDs provide the majority of winter precipitation to the Western Himalaya and surrounding area. | high | 3.2 |
| By recharging glaciers and the snowpack, WDs are vital for regional water security. | very high | 3.3 |
| Rabi crops rely on WD rainfall. | medium | 3.4.1 |
| Heavy hail or snow from WDs can damage crops. | high | 3.4.1 |
| WDs provide conditions conducive to widespread fog. | very high | 3.4.2 |
| WDs reduce pollution levels through increased rainfall and near-surface winds. | medium | 3.4.2 |
| WDs can cause coldwaves over north India. | high | 3.4.3 |
| WDs are the primary cause of pre-monsoon lightning over north India. | high | 3.4.4 |
| Landslides in the Western Himalaya are often triggered by WDs. | medium | 3.4.5 |
| WDs can trigger avalanches in the Western Himalaya. | very low | 3.4.5 |
| The interaction between WDs and the summer monsoon often leads to very heavy rainfall. | very high | 3.5 |
| A positive phase of the NAO leads to increased WD frequency and intensity. | very high | 4.2 |
| A positive phase of the AO leads to increased WD frequency. | high | 4.2 |
| El Niño leads to increased WD frequency. | low | 4.3 |
| El Niño leads to increased seasonal precipitation over the Western Himalaya. | very high | 4.3 |
| A positive phase of the IOD leads to increased WD frequency. | very low | 4.4 |
| Simulations of WDs are mostly insensitive to the choice of parameterisation schemes. | high | 5 |
| Simulations of WDs are sensitive to the choice of land surface dataset and parameterisation. | high | 5 |
| Increasing model resolution considerably improves simulations of WDs and their impacts. | very high | 5 |
| WD tracks can be skilfully forecast in operational models. | very low | 6 |
| WD frequency was higher during most of the Late Pleistocene (60–12 ka). | medium | 7.1.1 |
| WD frequency was much lower during the Early Holocene (12–8 ka). | medium | 7.1.2 |
| WD frequency was lower during the Mid Holocene (8–4 ka). | very high | 7.1.3 |
| WD frequency was lower during the Roman (2.5–1.9 ka) and Medieval (1.5–0.7 ka) Warm Periods. | high | 7.1.4 |
| WD frequency was higher during the Little Ice Age (0.7–0.2 ka). | very high | 7.1.4 |
| There is no clear trend in WD frequency during the instrumental period. | medium | 7.2.1 |
| Winter precipitation over the Western Himalaya has declined in recent decades | high | 7.2.2 |
| Climate change will cause WD frequency to decline. | very low | 7.3.1 |
| Climate change will cause winter precipitation to increase over the Western Himalaya | high | 7.3.2 |
| Climate change will cause winter precipitation to decrease along the Himalayan foothills | high | 7.3.2 |
| Climate change will cause the ratio of snowfall to rainfall to decrease | very high | 7.3.2 |

**Table 2.** Summary of the key statements that have emerged from WD literature in the last decade, along with the confidence in those statements (following the IPCC definitions of confidence) and section in which the relevant studies can be found.

And with this in mind, it reads a bit strange to come back to the summary after the future research questions and challenges. One could imagine having one big Conclusion and summary section, which includes Section 8 about "Future research questions and challenges".

We will adopt this suggestion, moving the summary to Sec 8.1 and the future research questions to Sec 8.2 (part of an overarching Sec 8, "Conclusions").

40) Section 9.1 is out of place in my opinion. This is a sort of glossary remark or definition that should probably be placed much earlier in the paper. After the summary it comes very much as a surprise to me.

Agreed, and this has been suggested by other reviewers too. In the revised version, we will move this to the end of section 3.

41) I must emphasise that I really like Section 8, it is very inspiring, and I would have liked to finish reading the paper in this opening-view.

Thank you!

42) L. 1876: repetition of primarily

Thanks, although both were intended this does stick out. We will replace the former with "mostly".

---

## Author Comment (AC2)

I have reviewed this article with my limitations, I hope authors may find it helpful for improving the readability and scientific credentials.

Title: Western disturbances and climate variability: a review of recent developments

Author(s): Kieran M. R. Hunt et al.

MS No.: egusphere-2024-820

MS type: Review article

Recommendation: Accept with revision

This is a comprehensive review of WDs, authors have put meticulous efforts in this review research work by including all the available relevant research studies. This is surely useful for researchers interested in this field. However, I strongly feel simplified description will be more beneficial for new researchers to get crucial interest in the subject. It is clear that this review is more focussed on boreal winter time WDs, having baroclinic structure, basically 'frontal synoptic scale' in nature? Here the dynamical processes are dominant over thermodynamical. Though it is more confined to Himalayan regions, it also extends over central and western Indian regions. In my opinion this review of the past and present studies can be better structured (like IPCC report) where any scientific argument is categorised with low, medium and high confidence level. This may help in simplifying the description, otherwise it very confusing at each stage. The simplified description will enhance the readability as well as its scientific credentials. In deed this article should be accepted in this journal but with revision.  Kindly find the line by line comments below.

We would like to thank the reviewer for their positive assessment of our manuscript, and for their detailed comments, which we respond to point-by-point in red below. Planned revisions to our manuscript will be highlighted in blue. This includes an IPCC-like confidence statement at the end of each section, which will then be summarised in our conclusions section.

Line 90: While describing western disturbances (in addition to it's interaction with summer monsoon systems) it would be more appropriate to distinguish it from typical summer monsoon synoptic systems in which the complex thermodynamics as well as dynamics plays a crucial role.
OK – we will add a sentence here drawing a contrast between WDs and the summer monsoon: "WDs differ considerably from monsoon low-pressure systems, the other

synoptic-scale vortex that regularly affects the subcontinent, whose development and propagation is driven by moist thermodynamics coupled to the mean monsoon flow."

Line 100: Along with Chevuturi and Dimri, 2016, you may like to refer Vellore et al. 2015/16
Thank you for drawing our attention to this reference. We will include it in our revision. (Reviewer is not clear exactly what paper they are referring to here, but we think it is "Monsoon-extratropical circulation interactions in Himalayan extreme rainfall" in Climate Dynamics)

Caption of Figure 3: It is Cold and dry 'air' advection?
Yes, though "air" is often conventionally omitted from such phrases. We will include it here for clarity.

Line no. 136: Firstly, recent studies …… increasingly ?????? high-resolution models, ….
The original sentence is "Firstly, recent studies have made use of increasingly inexpensive high-resolution models, both for regional climate modelling and numerical weather prediction." We will revise this to read: "Firstly, recent studies are making increased use of high-resolution models, which are becoming cheaper to run, both for…"

Line No. 140 and 145: Please Consider simplifying these statements.
The existing sentence on L140 is: "The large number of high-resolution experiments also serves as a primitive large ensemble -- as these models are able to capture processes more faithfully, experiments can more easily establish which physics schemes, forcings, and configurations are most important, collectively driving down the model uncertainty from which earlier studies suffered." We will revise this to: "The large number of high-resolution experiments also act as a primitive large ensemble. As these models better capture small-scale processes, experiments can more easily establish which physics schemes, forcings, and configurations are most important, reducing the model uncertainty from which earlier studies suffered."

The existing sentence on L145 is: "These developments have helped to link the physical processes of individual storms to the large-scale weather in which they are embedded, and to understand directly the influence of climate change on the statistical behaviour of WDs." We will revise this to: "These developments have helped to link the physical processes of individual storms to the larger weather systems they occur in, clarifying the direct influence of climate change on the statistical behaviour of WDs."

Line No. 160: Kindly include Vellore et al. 2015/16

Please refer to our response to an earlier comment regarding this reference. We will include it in the relevant section (3.5).

Line No. 191 and 192: These studies are 'more recent analyses'???? Sentence may be corrected.

Yes, 1999 and 2011 are more recent than 1947, 1956, and 1969. We will keep this sentence as it is.

Line No. 271: How WDs are different from Frontal system?

We agree that the difference between WDs and (we think the reviewer means) extratropical cyclones should be made clear. L271 is not the correct place for this, but we will include the following: "However, many features present in extratropical cyclones, such as frontal fractures, sting jets, and warm seclusions, have not yet been observed in WDs" at the beginning of Section 2.1. Also note the difference is already raised as an open question in Section 9 (q6).

Line 282,283: Sentence is not clear.

The sentence in question is: "There is a preference for cyclogenesis in regions of dynamical instability; typically downstream from mountain ranges, but also within the North Atlantic jet stream." We will rephrase this to: "Cyclogenesis tends to occur in areas of dynamic instability, often found downstream from mountain ranges or within the North Atlantic jet stream."

Line No. 344: Simplify the sentence for better readability describe how a negative correlation with ......?

The full sentence is "This is supported by Chand and Singh (2015), who found, when using satellite data to analyse a group of 10 WDs, that WD propagation speeds varied between 280 and 670 km day$^{-1}$ and had a negative correlation with cloud-top height downstream, implying that WDs associated with stronger convection tended to propagate more quickly." We agree that this is quite long (and also contained a mistake) and will replace it with: "This is supported by Chand and Singh (2015), who used satellite data to find a negative correlation between WD propagation speed and downstream cloud-top height, implying that WDs associated with stronger convection tended to propagate more slowly. They also showed that WD propagation speeds vary substantially, from 280 and 670 km day$^{-1}$".

Line No. 349 – 351 : Do you mean baroclinicity?

No – but we appreciate this sentence could be more clearly worded: "In summary, the deep ascent ahead of WDs primarily occurs due to downstream upper-tropospheric divergence. This is supported by quasigeostrophic differential vorticity advection and mechanical uplift of induced lower-level southerlies as they interact with the orography."

Line No. 378 : Please correct the sentence for better readability.
The sentence is "The second WD spun up over northern Europe on Jan 22, before migrating southward and then propagated rapidly towards and then over the Western Himalaya, where it resulted in heavy precipitation." We will replace this with "The second WD spun up over northern Europe on Jan 22. It then migrated southward, before moving rapidly towards and then over the Western Himalaya, where it subsequently resulted in heavy precipitation."

Line No. 436-439: Sentence not clear.
The sentence here is: "However, this is complicated by fractionation – wherein rain preferentially forms from low D-excess water – further increasing the D-excess in moisture in air parcels that have been transported a long distance, orographically lifted, or even locally recycled (Kong et al., 2013)."
We will revise this to: "However, this is complicated by fractionation. Rain forms preferentially from low D-excess water, and so D-excess increases in moisture in air parcels that have been transported a long distance, orographically lifted, or even locally recycled (Kong et al., 2013)."

Line No. 440: Provide suitable references.
The sentence is "Ideally, therefore, the results of isotope analysis over the western Himalaya should be disambiguated with a complementary moisture trajectory or moisture flux analysis." This follows on from the previous sentence which discusses the uncertainties arising from fractionation. A reference is therefore not needed here.

Line No. 445: flawed????
It is unclear whether the reviewer is uncertain of the definition of "flawed" or its application to the list of references. The latter is clearly explained in the sentence itself: "…relying on only short sample periods or applying trajectory analysis either only to case studies or for whole seasons." A simple definition of "flawed" is thus provided here: *having a fundamental weakness or imperfection*.

Line No. 455: How significant is Mediterranean moisture?? here when it is not a majority moisture source?
Here we were quoting the conclusions sections of both papers. Jeelani et al (2017) does not explicitly quantify the Mediterranean contribution. Dar et al (2021) does in their Table 5, where they give the probabilities of each basin being the majority contributor for certain types of event. For the Mediterranean, they give values in the range of 20–30%, which we will include in our revision.

Line 475-485: In fact Section 2.4 is too confusing, you may kindly retain very relevant references?
The reviewer is here referring to the paragraph at the end of Section 2.4 which discusses how (Eulerian) moisture flux analysis can also be a useful tool in deducing moisture sources, alongside isotope-based or Lagrangian methods. We briefly

mention the recent results of Baudouin et al (2021), who examined these moisture pathways on seasonal timescales, before linking those results to earlier work on atmospheric rivers. We will rephrase, shorten, and try to improve clarity as follows: "Beyond isotope and trajectory methods, recent work by Baudouin et al. (2021) highlighted the potential use of composite moisture flux analyses in investigating precipitation moisture sources, with the caveat that such analysis only works on seasonal timescales or longer. They identified a mean moisture pathway between the Red Sea and the North Arabian Sea and showed that WDs transiently steer this pathway towards the western Himalaya and surrounding region. Results obtained using this method are very similar to those obtained from large-sample back-trajectory studies (e.g., Fig. 9). These pathways are analogous to the atmospheric rivers that are responsible for winter precipitation and flooding to the west, in Iran (Dezfuli, 2020; Dezfuli et al., 2021; Esfandiari and Lashkari, 2021). Atmospheric rivers have also been explicitly linked to the majority of winter precipitation variability and extremes over the western and central Himalayas (Rao et al., 2016; Thapa et al., 2018; Lyngwa et al., 2023), where composite analysis shows circulation that strongly resembles that of a WD. The altitude of these moisture pathways also appears to be important, with the largest moisture transport occurring between 850 and 700 hPa, a higher altitude than usual in the tropics (Baudouin et al., 2020b)."

Line No. 510: Figure 10: Caption- Is the percentile calculation based on entire time-seires or has been calculated on monthly basis.
As already stated in the caption, this is *overall* intensity percentile based on the full time series rather than monthly. If it were monthly, the deciles would all have the same size for a given month. We will clarify this in our revised caption.

Line No. 531: '….. associated with all winter WDs' What about other seasons?
The reviewer is here asking about our definition of "active" WDs. For this, we look at the daily precipitation over the Western Himalaya and surrounding region for all winter WDs, and take the top quartile of systems. There are several reasons we restrict this definition to winter. Firstly, it is consistent with earlier literature cited in this section (e.g., Datta and Gupta, 1967; Rao and Srinivasan, 1969; Chattopadhyay, 1970; Subbaramayya and Raju, 1982; etc), and this is, after all, a review paper. The vast majority of WDs occur in the winter months and the majority of their impacts are felt in this season. Secondly, we want to highlight the links between heavy precipitation in WDs and other WD characteristics. If we included monsoonal WDs in this, they would almost all by definition be active, since they can draw in monsoonal air masses and thus tend to precipitate much more heavily. Our section on variability would not then contrast strong and weak WDs, rather winter and summer WDs – which we already do in Section 2.3.3 and 3.5. Finally, a significant fraction of monsoonal WDs may arise as polar PV cutoff lows (see Sec 2.3.3, or Thomas et al., 2023) and may have different structure, characteristics, and behaviour. As these differences are not yet known (see Sec 8, Q10), we do not want to contaminate this overview with a small sample of potentially very different systems.

Line 535: why 350 hPa is being considered in analysis? please provide the supporting argument
This is the pressure level at which the average WD has its maximum vorticity (see Figure 8). We will clarify this with a footnote in the revision thus: "The choice of 350 hPa arises from Fig. 8, which shows composite WDs have their maximum vorticity at this pressure level."

Line no. 550: The difference between two studies is not understood here.
These references support the prior statement, which is that WD latitude can have a significant impact on WD characteristics and impacts. Both studies discuss this, in slightly different ways: Baudouin et al (2020b) show how WDs at different latitudes manipulate the mean moisture pathway (and hence precipitation) to different extents; Baudouin et al (2021) show how WDs at different latitudes encounter different orographic configurations, and hence varied thermodynamic environments. For the sake of brevity, we do not include these specific details in our manuscript.

Line no. 570: dynamical characteristics and categories are two separate issues?
Yes, categories typically discretise and label certain characteristics. Consider tropical cyclones in the North Atlantic – the characteristic is wind speed, but this is often discretised into five category bins (the Saffir-Simpson scale) which helps with public, operational, and even academic communication. Our point here is that no such system yet exists for WDs, and that developing one requires careful consideration given the complex relationship between WD characteristics and their impacts. We will slightly adjust the last sentence here for clarity, replacing "categorise" with "categorise or classify".

Line no. 642-643: This could be part of data and methodology?
This sentence discusses a shortcoming of one study that uses gauge data in NW India to assess the reliability of various publicly available gridded precipitation datasets. The flaw in this study is that they did not realise their gauge dataset was not independent from gridded gauge datasets that they rated highly. As this section is on evaluating precipitation datasets in the region, we believe it is appropriately placed. Note that as this is a review paper, we do not have a data and methodology section.

Line no. 665: is it supported by back trajectories etc?
Yes, although not in Jeelani and Deshpande (2017). We discuss this in much greater detail in Sec. 2.4, which we will reference here in the revision.

Line no. 670: any reference?
Thanks for the suggestion. We will add Kulkarni et al (2021; https://doi.org/10.1016/j.wasec.2021.100101) and Mukherji et al (2019; https://doi.org/10.1007/s10113-019-01484-w) here.

Line no. 702:  This can be shifted to next section?

The reviewer is here referring to section 3.3.3 "crops and flora". We believe they mean into the next subsection (3.4, "natural hazards and other impacts") rather than the next section (4, "large-scale forcing and teleconnections"). We are happy to make the suggested change, so that "crops and flora" will be section 3.4.1 in the revision.

Line no. 753:  'radiation fog' – any reference?

Yes, the relevant reference, Patil et al (2020) is at the beginning of the previous sentence. The next sentence then clearly runs on from that "they show that... WDs... provided perfect conditions for radiation fog."

Line no. 757: You mean blocking high?

It is not clear what the reviewer is attributing to a blocking high here. The sentences in question are: "Hingmire et al. (2019) also found a significant increase in foggy days from 1980 to 2013 using data from four major cities over the IGP (Delhi, Lucknow, Hissar, Amritsar). While this increasing trend in fog events may be explained by changes in WD activity, increasing the relative tendency of a solid substance to absorb moisture from its surrounding environment levels of pollution over the region and the increased moisture flux associated with WDs in a warming world may also play a role (Verma et al., 2022, see also Sec. 7.3)."

Line no. 795: what about sub continental blocking?

Yes, both Ratnam et al (2016) and Athira et al (2024) mention that coldwaves not associated with WDs appear to arise from blocking patterns. We already state this at the end of this paragraph: "Subsequent composite analysis linked normal coldwaves to WDs, but the intense coldwaves were found to be more commonly associated with omega blocking over Siberia (Athira et al., 2024)."

Line no. 869-874: How these past and recent studies are connected?

The inclusion of the Pisharoty and Desai reference here is in error. We will fix this and correct the sentence accordingly in the revision.

Line no. 880: Please be clear what you want impress upon.

It is not clear what the reviewer is requesting here. The sentences in question are: "This shift in seasonality was later confirmed by Hunt (2024), as we will discuss in Sec. 7.2. In fact, WDs can occur at any time of the year (hence their occasional interaction with the summer monsoon), but are usually most active between November and February (Fig. 10)." To improve clarity, we will replace "active" with "frequent" in our revised manuscript.

Line no. 992-994: I get lost between Agricultural applications and features over Indo-Gangetic plains

We're not sure what the reviewer is asking for here. Firstly, there is no mention of the IGP or agriculture on L992-994. The nearest mention of either is the IGP on line

943, but that is to do with fog variability rather than agriculture. No other mention of the IGP in our manuscript references agriculture.

Line no. 944: What is fir tree? In this sentence
We believe the reviewer means L948. We will clarify this by revising the sentence thus: "This signature also appears in paleoclimate studies, with a positive NAO linked to increased precipitation over the Indus Basin in both fir tree -- a type of confiner – cellulose..."

Line no. 946-949: How this connected with WDs?
The paragraph in question refers to winter precipitation rather than WDs specifically – noting that studies have found a strong covariance with the NAO. As we state at the end of the introduction: "In some parts of this review, we have included additional papers that cover winter precipitation over the relevant region, as this can be a useful proxy for WD frequency and such papers can add useful evidence to the discussion."

Line no. 956: I am again lost here to connect with WDs.
Please see response to previous comment.

Line no. 991: Sudden jump to stratosphere? when ENSO relation itself is not clear?
In this paragraph, we are discussing possible reasons *why* the ENSO relationship is unclear. These studies fall into two groups. Firstly, we discuss those that investigate different flavours of ENSO (e.g., Central Pacific vs Eastern Pacific). Secondly, we discuss those that investigate the role of the QBO in modulating the effects of ENSO. This is not, therefore, a sudden jump to the stratosphere; rather a discussion of all the possible confounding factors in the ENSO-WH precipitation relationship.

Line no. 994: What is SSW? In this sentence?
SSW stands for sudden stratospheric warming. This is mentioned in the previous sentence but we appreciate we did not add the abbreviation in parentheses there and so it is easily missed. This will be corrected in the revised manuscript.

Line no. 1005: What is IWM in this sentence?
This stands for Indian winter monsoon. However, the inclusion here is in error and it will be removed in our revised manuscript.

Line no. 1019: Needs more attention.
We agree, this is why it is included as one of our future research questions (Sec 8, Q20).

Line no. 1034: Is it region specific? As it is not seen in case of summer monsoon convection over Western Ghats?
This appears to be true wherever convection and orography interact, since better representation of both intuitively leads to a better representation of their

interaction (e.g. Hohenegger et al, 2008 doi: 10.1127/0941-2948/2008/0303; Fosser et al, 2015 doi:10.1007/s00382-014-2242-1). This is also true for other parts of the Indian subcontinent (Willetts et al, 2016 doi:10.1002/qj.2991).

Line no. 1045: This may be true when dynamics is dominant in the weather system?
The sentence in question is: "How important is the choice of convection scheme in simulating WDs?" We're not sure what the reviewer is asking here, but since dynamics are important in all WDs (since they are upper-tropospheric lows that pass along the subtropical jet) it is not clear what contrast they want us to draw.

Line no. 1064-1065: This statement is irrelevant here.
We are happy to follow the reviewer's discretion here and remove it.

Line no.  1067-1069: repeated statement.
This is true, we refer to Sarkar et al (2019) in the previous paragraph as well. The methodology of this paper is repeated and we will remove it in the revision.

Line no. 1070-1074: This sentence is not clear.
The original sentence reads: "This is because they are still capable of capturing much of the necessary local thermodynamics – Patil and Kumar (2017) demonstrated realistic CAPE and OLR behaviour in two WRF case studies -- as well as the synoptic-scale dynamics – Mannan et al (2017) demonstrated realistic precipitation even for the unusual situation of WDs passing over Bangladesh, where they draw on moisture flux from the Bay of Bengal." We will revise this to: "This is because they are still capable of capturing much of the necessary local thermodynamics as well as the synoptic-scale dynamics (Mannan et al., 2017; Patil and Kumar, 2017)".

Line no. 1076: Infact the local dynamics seems to play important role.
This is indeed true, as we discuss in Secs. 2.3 and 2.4. As the dynamics are invariably coupled to both convection and the orography, representation of these smaller-scale processes in models is crucial for accurate forecasting of WD impacts.

Line no. 1084-1089: very confusing statements, needs reformation.
The original passage reads: "Moving away from WRF, Laskar et al. (2015) comprehensively examined two cases of intense WDs that occurred during March 2015. Using output from the IMD operational model, the GFS, and local Doppler weather radars, they found that extreme precipitation associated with the WDs, linked to anomalous southerly moisture flux from the Arabian Sea, was undersimulated by the models due to their poor representation of deep convection. Dutta et al. (2022) showed that this negative wind bias in forecast WDs could be overcome by assimilating winds from Doppler radars in north India."

We will revise this to: "Apart from WRF studies, Laskar et al. (2015) conducted an in-depth analysis of two intense WDs occurring in March 2015 using data from the IMD

operational model, the GFS, and local Doppler weather radars. They found that the models underestimated the extreme precipitation associated with these WDs due to a poor representation of deep convection, despite correctly modelling the strong southerly moisture flux from the Arabian Sea. Dutta et al. (2022) further showed that the negative wind bias in these forecasts could be reduced by assimilating wind data from Doppler radars in northern India."

Line no. 1100: How it is connected to WDs.
This was about the representation of WH precipitation in CMIP6 models. We will rewrite this sentence to clarify the link to WDs: "These results were extended for CMIP6 models by Meher and Das (2024), who argued that almost all CMIP models have different strengths and weaknesses in representing the range of mechanisms required to drive precipitation, including from WDs, over the Western Himalaya. They identified the representation of mid-latitude winds, choice of land-surface dataset, and choice of physical parameterisation schemes as important drivers of model skill."

Line no. 1115: Is it connected to WDs?
Yes – this is about the simulation of winter precipitation over the Hindu Kush and Karakoram in high resolution climate models. Most of that precipitation is provided by WDs. We will clarify that in the revision: "Indeed, higher resolution climate models do perform better: Iqbal et al. (2017) found that models of the CORDEX-SA experiment simulated winter precipitation across the Hindu-Kush and Karakoram – most of which is provided by WDs – well."

Line no. 1150: In fact, these early studies explored the qualitative analysis.
This is a good point, we will add this into the revision: "Before this, forecast verification was largely confined to qualitative case studies…"

Line no. 1164: This is a serious concern needs to be addressed appropriately.
Thank you – we agree. That is why this issue is mentioned in future research questions 22 and 23.

Line no. 1180: '…..context of WDs is left an important ….. ' This is a serious concern needs to be addressed appropriately.
Thank you – we agree. That is why the issue is mentioned in future research question 22.

Line no. 1190: Which is tract 1 in Figure 17??
Track 1 is labelled in blue in Fig 17 (see the legend directly underneath the map). This WD is particularly interesting as it highlights the large uncertainties that can arise in WD track forecasts from the jet moving either side of the Pamirs.

Line no. 1203: '….. sensitivity had to be reduced …..' Needs to be elaborated here.
Yes, we will do this. The original sentence is: "The modification was required

because the forecast output has daily sampling frequency and so, among other things, the sensitivity had to be reduced to mitigate incorrect linkages." We will revise this to: "The modification was required because the forecast output has daily sampling frequency. This included a reduction in the sensitivity of the detection algorithm which mitigates incorrect linkages by increasing the minimum vorticity threshold at which candidate WDs are detected -- which in turn reduces aliasing, false positives, and hence incorrect linkages."

Line no. 1230: '….. winter precipitation there is brought by WDs.' Sentence is not clear.
Please see our response below.

Line no.1234-1236: This statement is contrary to that of line no. 1230.
We agree this introduction was unclear. Following the advice of several reviewers, we have rewritten this to explain the caveats of interpreting WD activity from paleoclimate studies: "Paleoclimate research has become increasingly popular over the last few decades, especially as more advanced proxy techniques have been developed and refined. For precipitation, these include speleothems, marine and lake sediments, tree rings, and pollen analysis. As we discussed in Sec. 3.2.2, present-day WDs are responsible for the majority of total winter precipitation over the Western Himalaya and surrounding region and likely – through changes in WD frequency and intensity – the majority of its interannual variability as well. For these reasons, precipitation is often used in paleoclimate studies as a proxy for WD activity over the Western Himalaya. However, there are several important sources of uncertainty that arise with this approach. Firstly, the relative contributions of winter precipitation (i.e., WDs) and summer precipitation (i.e., the monsoon) to the annual total may change over time. However, this uncertainty can largely be removed by quantifying the d-excess of the sample studied (see Sec. 2.4). Secondly, as mentioned above, some winter precipitation variability must arise from non-WD sources, the primary source of which is cloudbursts. The fraction is unknown, but probably small, and may also have varied over long time periods. Thirdly, analyses often make do with proxies from winter precipitation dominated areas nearby (e.g., Iran, central Asia), and extrapolate the result to the study area (e.g. Petrie and Weeks, 2018). Thus, while we can be reasonably confident that long-term changes in winter precipitation are related to changes in WD activity, we must bear these caveats in mind when discussing the results of the paleoclimate studies that follow."

Line no. 1269: This is very confusing.
The sentence in question is: "Kar and Quamar (2020) also argued for increased WDs in the early Holocene, although their technique could not readily distinguish between summer and winter precipitation." We will revise this to: "Kar and Quamar (2020) also supported increased WD frequency during the early Holocene, but their methodology was unable to clearly differentiate between summer and winter precipitation."

Line no. 1281:' … Paleoclimate modelling' It would be more appropriate to segregate observational and modelling studies

Thank you for the suggestion. We disagree for two reasons. Firstly, modelling studies make up only a small minority of studies discussed in this section; and secondly, for readability, we want to discuss the literature in chronological order of study period.

Line no. 1315: What is the confidence level here?

This is a good point, we used "probably" when in fact the confidence level is very high. We will remove this in our revision.

Line no. 1413: Section 7.2.1 Counting WDs - Very interesting section can be better presented - it is very complex at the moment

Thank you. Following your comment and one from reviewer 2, we will revise Sec 7.2.1. to be shorter and clearer.

Line no. 1463: it is Krishnan et al. 2019?

Yes, thanks for spotting this. This is different from the other Krishnan et al (2019), and was first published online in 2018 (though in a journal in 2019, which we will change this reference to).

Line no. 1471: No confidence?

Yes, as we discuss, the sign and significance of the trend varies with region, methodology, season, and study period. While we are able to disentangle some of these factors, we still have no confidence in the overall sign of the trend of WD frequency during the historical period. We will clarify that in the revision: "In summary, there is disagreement among recent studies on the sign and significance of the trend in WD frequency over the past 70 years. There is thus no confidence in the overall sign of the trend of WD frequency over the western Himalaya in the instrumental record.".

Line no. 1480: Here - The impact of climate forcing over the trend would be very interesting?  Though may not have confidence level.

We agree, yet no study has attempted to disentangle the respective roles of interdecadal variability and climate forcing on WD trends. We will add this as a future research question: "28. What are the respective roles of interdecadal variability and climate change in recent observations of seasonal and regional trends in WD frequency?"

Line no. 1495: '…interdecadal variability' - There are lots of jumps from long-term trends to decadal scale trends?

No, the focus is indeed on long-term (climate trends). The difficulty in synthesising these studies arises from the fact there is a lot of decadal-scale variability. We mention this in the original manuscript on L1494: "Once datasets or regions with spurious behaviour are removed from the analysis, the key issue is decadal

variability -- meaning the results are sensitive to the choice of analysis period" and then explain in subsequent sentences. Essentially, any discussion of trends in WD behaviour must explain why those trends vary in sign and strength depending on the study, and the answer here is that many such studies are picking up decadal-scale trends from natural variability instead.

Line no. 1506-1508: Very difficult to understand this content.
The sentence is "Other studies have reported similar results for the Central Himalaya and Nepal Shrestha et al. (2019), states of north India (Rajasthan, Gujarat, Punjab Narayanan et al., 2016), Jammu (Khan et al., 2023) and Kashmir (Dar, 2023)." The only part we imagine the reviewer must not understand is the "similar results" part, which refers to the previous sentence. We will replace "similar results" with the more explicit "similar results – i.e. a weak trend dominated by interdecadal variability –".

Line no. 1528: '....which attributed to WDs' Is it the frequency of WDs?
Yes, we will clarify this, replacing "which they attributed to WDs" to "which they attributed to increased WD frequency".

Line no. 1530 : is it related to increased WD frequency?
It most likely is, since WDs are a major cause of convective storms in the region. However the authors did not explicitly make this link, and so neither did we. We will update this sentence in the revision thus: "Bhat et al. (2024) reported a significant and very large increase in reported pre-monsoon hailstorms in Kashmir between 2007 and 2022. This is likely due to WDs, as the predominant source of non-monsoonal convective activity in the region."

Line no. 1544: '..... surface levation.' What about lapse rate?
This sentence is the definition of elevation-dependent warming: "While the general decline in snowfall is attributed to a warming climate, the spatial variability is thought to be linked to elevation-dependent warming, where trends in near-surface warming increase as a function of surface elevation." Including discussion on lapse rate would not thus be relevant here, but we will add a clause later in the paragraph: "There are thought to be a number of important drivers, depending on season and location, with changes in albedo (Ghatak et al., 2014), snow depth, cloud cover (Duan and Wu, 2006), near-surface humidity (Rangwala et al., 2009), lapse rate (Qin et al., 2024), and radiative forcing (Palazzi et al., 2017) chief among them."

Line no. 1554: is it also supported by in-situ observations?
Yes, Li et al (2020), cited in this sentence, is based on surface meteorological stations with long records. We will clarify this in the revised manuscript.

Line no. 1576: '.... Anomaly' - you mean positive anomaly? If so mention it for better readability.
Thanks – we will add this.

Line no. 1580-1581: most closely and mostly closely? Correct the sentence.
Thank you, the "mostly closely" should read "more closely". We will correct this.

Line no. 1581-1582: '..... particularly as a result of changing WD activity.' Please explain how?
This follows from the line in the study cited in this sentence, Mehta et al (2021): "The glaciers in the study area (Suru River valley) are mostly nourished by the Western Disturbances (during the December, January, and February) with maximum solid precipitation, and melt during the ablation period (May–October)." It also follows from earlier arguments that interannual variance in WH/Karakoram winter precipitation is predominantly driven by WDs. We will rephrase this sentence accordingly: "Mehta et al (2021) showed that trends in glacial ablation are most closely associated with increasing temperature, but trends in glacial accumulation are more closely associated with increased winter precipitation, particularly due to WD activity, which they state is the primary source of glacier recharge in this region."

Line no. 1588-1590: Please restructure the sentence for better clarity.
The original sentence was: "Despite these advances, it is clear that a great deal more research is needed on how climate change across the Himalayas, Karakoram and Hindu Kush will have downstream impacts on wetlands, agriculture, and ecosystems in general (Chettri et al, 2023)." We will rephrase this in the revision: "Despite these advances, further research is urgently needed to understand how climate change in the Himalayas, Karakoram, and Hindu Kush regions will affect downstream wetlands, agriculture, and ecosystems more broadly (Chettri et al, 2023)."

Line no. 1625: It would be more appropriate to summarise the contents here before proceeding further.
This section comprises two short paragraphs, and so we will add only a very brief summary: "There was thus no consensus on whether climate change would cause WD frequency to increase or decrease, and only low confidence that winter precipitation would increase."

Line no. 1720: Though it is a comprehensive description of future projections, it would be more appropriate to classify this in near-future, mid-future and far-future. The uncertainty of near future projection say 2030 or 2040 could be very useful for various sectors.
Thank you for this suggestion, but this would require an advanced synthesis as many authors do not make these data available in their studies. As such, it is out of scope for this review, but we will include it in our revised future research questions: "33. There is also only a weak consensus on the projected future decrease of winter precipitation in the western Himalaya. Studies leveraging high-resolution models that are capable of resolving orographic feedbacks are needed to make more robust estimates of these changes, both in the near future and far future.".

Line no. 1721: Section 8 Future research questions and challenges: This section is very well written.
Thank you very much.

Line no. 1819: In view of the above comments Section 9 Summary needs to be considerably improved for quantitative description and better readability.
Following this comment and your summary at the beginning, we will revise Section 9 (now Section 8.1) to include a table of all the key points synthesised in the review and the confidence level associated with them (see below). We will also make improvements to the clarity of the text.

| Statement | Confidence | Section |
|---|---|---|
| Tracking algorithms are a useful tool for understanding WDs. | high | 2.2 |
| WD cyclogenesis mostly occurs over ocean or downstream from mountain ranges. | medium | 2.3.1 |
| WDs intensify through baroclinic instability, sometimes with moist or orographic coupling. | very high | 2.3.2 |
| WDs primarily affect the Western Himalaya and surrounding mountain ranges. | very high | 2.3.3 |
| WDs have mid- to upper-tropospheric vorticity maxima with ascent ahead of their centre. | very high | 2.3.4 |
| The Arabian Sea is the primary moisture source for WD precipitation. | high | 2.4 |
| WDs are most frequent between December and March but can occur at any time of year. | high | 2.5 |
| There is large variance in most WD characteristics, such as lifetime, intensity, and latitude. | high | 2.6 |
| WDs provide the majority of winter precipitation to the Western Himalaya and surrounding area. | high | 3.2 |
| By recharging glaciers and the snowpack, WDs are vital for regional water security. | very high | 3.3 |
| Rabi crops rely on WD rainfall. | medium | 3.4.1 |
| Heavy hail or snow from WDs can damage crops. | high | 3.4.1 |
| WDs provide conditions conducive to widespread fog. | very high | 3.4.2 |
| WDs reduce pollution levels through increased rainfall and near-surface winds. | medium | 3.4.2 |
| WDs can cause coldwaves over north India. | high | 3.4.3 |
| WDs are the primary cause of pre-monsoon lightning over north India. | high | 3.4.4 |
| Landslides in the Western Himalaya are often triggered by WDs. | medium | 3.4.5 |
| WDs can trigger avalanches in the Western Himalaya. | very low | 3.4.5 |
| The interaction between WDs and the summer monsoon often leads to very heavy rainfall. | very high | 3.5 |
| A positive phase of the NAO leads to increased WD frequency and intensity. | very high | 4.2 |
| A positive phase of the AO leads to increased WD frequency. | high | 4.2 |
| El Niño leads to increased WD frequency. | low | 4.3 |
| El Niño leads to increased seasonal precipitation over the Western Himalaya. | very high | 4.3 |
| A positive phase of the IOD leads to increased WD frequency. | very low | 4.4 |
| Simulations of WDs are mostly insensitive to the choice of parameterisation schemes. | high | 5 |
| Simulations of WDs are sensitive to the choice of land surface dataset and parameterisation. | high | 5 |
| Increasing model resolution considerably improves simulations of WDs and their impacts. | very high | 5 |
| WD tracks can be skilfully forecast in operational models. | very low | 6 |
| WD frequency was higher during most of the Late Pleistocene (60–12 ka). | medium | 7.1.1 |
| WD frequency was much lower during the Early Holocene (12–8 ka). | medium | 7.1.2 |
| WD frequency was lower during the Mid Holocene (8–4 ka). | very high | 7.1.3 |
| WD frequency was lower during the Roman (2.5–1.9 ka) and Medieval (1.5–0.7 ka) Warm Periods. | high | 7.1.4 |
| WD frequency was higher during the Little Ice Age (0.7–0.2 ka). | very high | 7.1.4 |
| There is no clear trend in WD frequency during the instrumental period. | medium | 7.2.1 |
| Winter precipitation over the Western Himalaya has declined in recent decades | high | 7.2.2 |
| Climate change will cause WD frequency to decline. | very low | 7.3.1 |
| Climate change will cause winter precipitation to increase over the Western Himalaya | high | 7.3.2 |
| Climate change will cause winter precipitation to decrease along the Himalayan foothills | high | 7.3.2 |
| Climate change will cause the ratio of snowfall to rainfall to decrease | very high | 7.3.2 |

**Table 2.** Summary of the key statements that have emerged from WD literature in the last decade, along with the confidence in those statements (following the IPCC definitions of confidence) and section in which the relevant studies can be found.

Line no. 1822: Again to remind that WD over the region of interest is Importantly a synoptic frontal type of system having baroclinic structure and dominance of dynamics.

Thank you for the suggestion. We will certainly include that WDs are baroclinic here. As we discuss in Sec. 2, and then again in future research question #6, only a few WDs have traditional frontal characteristics, so we will not include that here. It is not clear what is meant by "dominance of dynamics" here.

Line no. 1839: Indeed, Quantitative description may be more beneficial for readers.
As we mention, studies have not been able to agree on the relationship between ENSO and WDs, and thus we are not able to provide a sensible quantitative estimate here.

Line no. 1866: Yes the future scope of this study is well defined in this manuscript.
Thank you.

---

## Author Comment (AC3)

This manuscript presents an excellent and thorough review of the current state of understanding of western disturbances. It is well structured and quite easy to follow while including substantial insights. A similar review was carried out in 2015 and much research has been published in the intervening years. The manuscript describes this more recent research in detail and also briefly discusses what was understood in 2015 for context. The manuscript makes a substantial contribution mainly by bringing together existing knowledge to increase overall understanding of the subject and formulating coherent plans for future research; there is also new material and new presentation of previously published material. The manuscript is very long: my personal recommendation is that its length should not be reduced (subject to any formal length limits imposed by the journal) as all of the material is of interest and relevance and any repitition serves to improve its clarity; it is also fairly well packaged into sections for those who are interested primarily in one aspect of western disturbances research. I note that it is longer than all of the 5 existing published review articles in WCD but of a similar length to the longest one.

I recommend publication subject to satisfactorally addressing the following issues/questions. Numbers without other context refer to line numbers in the manuscript.
We would like to thank the reviewer for their positive assessment of our manuscript, and for their detailed comments, which we respond to point-by-point in red below. Planned revisions to our manuscript will be highlighted in blue. We would especially like to thank this reviewer for taking time to check our interpretation of the many references within this review!

87: This is at first confusing as on line 94 they are described as moving eastwards. Do you mean westwards relative to the jet? In that case I suggest changing what is inside the parenthetical dashes to something like "a synoptic-scale trough moving westward relative to the subtropical westerly jet , in which it is embedded"
Thanks for spotting this typo. This should read eastward and will be corrected in our revision.

A general question that I thought might come up for a reader relatively new to western disturbances (more on reading Section 2 but perhaps could be addressed in Section 1) is whether WDs occur by definition in this particular part of South Asia (and perhaps similar phenomena occur elsewhere but have different names) or if they can only occur in this region (e.g., due to the unique orography of the Himalaya and what is to the west of it).
This is a very good question. Dynamically, they sit somewhere between extratropical cyclones and (mid-latitude) upper-tropospheric troughs – but it is through the combination of both proximity to mountains and maritime moisture supply that allows them to make such an impact. We are not aware that systems of this nature exist in other regions, but are happy to be corrected if any of the reviewers or editor is aware of any. We will add the following text to section 1: "Thus, the dynamics of

WDs sit somewhere between extratropical cyclones and mid-latitude upper-tropospheric troughs. Their unique proximity to both mountains and a maritime moisture source, however, leads to a unique array of impacts that means such storms are not found in other regions."

173-174: Can you explain how this is shown by Dimri (2004)?
Good catch – this was the wrong reference. We will update this with the correct reference (Dimri, 2008) in the revision.

I felt it might be worth clarifying that Figure 6 refers to all times rather than just during WDs.
Agreed, we will make this change: "Computed for all days (not just WDs) in winter months (December to March) using CloudSAT data."

344-345: Naively I'd associate a higher cloud top height with more convection -- can you explain how a negative correlation of speed with cloud top height implies that WDs associated with stronger convection tend to propagate more quickly?
This was a mistake – we will correct this to say that deeper convection is associated with more slowly propagating WDs.

Figure 8: is the mean for the anomaly 10 days either side of the WD?
Yes – that's correct. We will clarify this in the revised caption: "The anomalies are computed against a 20-day mean centred on the WD event."

401: It is not clear from Figure 8 how there is a northwestward tilt with height.
We agree that this is quite subtle in the composite. You can see it best in the PV field (upper-level PV maximum is several degrees to the west of the centre) and meridional winds, where the zero isotach starts at about +4° at the surface and finishes at about -4° at the tropopause. We will note this evidence in the revised manuscript.

435: Where does Pfahl & Sodemann (2014) say that colder climates lead to higher d values? I could only find a positive relationship between SST and d.
Agreed – and thanks for spotting this. We will make the appropriate correction here (removing the reference to air temperature).

Section 2.4: One question I had when reading this, was how much the methods rely on modelling, and whether they do extensive sampling of the isotopic ratios of collected precipitation. Obviously one can read the references but a sentence clarifying this in general might be interesting.
Good suggestion. In general, studies use isotope analysis to make a first guess and then support that with trajectory analysis. We will mention this in the end-of-section summary: "Many of the studies discussed in this section use both: typically making a first guess of moisture source using isotope analysis, and then supporting their hypothesis with trajectory analysis."

568-569: Is it more correct to say that Javed et al. (2023) thresholds on vorticity? (I realise this is directly dependent on wind speed though!)
No, they stratify on 300-400 hPa wind speed (see Table 1 of their paper).

580-581: I don't know if you are trying to say that these proportions of active WDs are surprisingly low, but if so the manuscript itself earlier defines active as only the top quarter!
Good point! Our choice of 25% was actually based on these studies, which we will clarify in the revision: "This leaves 25% of our WD population defined as active WDs, a relatively small number, but consistent with earlier definitions (Mohanty et al., 1998; Dimri, 2006)." So yes, perhaps a surprisingly small fraction associated with heavy precipitation, but it serves as a benchmark threshold nonetheless.

Section 3.4.1: Is it fair to say that this is particularly uncertain (compared with other topics discussed in the manuscript)? So the overall message is there is strong evidence that WDs affect fog, but in what way is as yet quite unclear?
We think this is a reasonable summary, and will add the following to the summary at the end of the section: "In summary, there is strong evidence that WDs affect the frequency of fog events over north India, but the exact nature of the relationship and its driving mechanism are as yet unclear."

596: Can you point out where Bamzai & Shukla (1999) and Liu & Yanai (2002) say this?
Good spot. We have revisited the interpretation given in Dimri's earlier review. We will use a more appropriate reference in the revised version (Biemans et al., 2019) and remove the part on delayed summer monsoon onsets.

774-775: Does the reference to Patnaik et al. (2024) relate to their mention of increased PDNC during WDs (so less pollution during WDs?)?
I think PDNC is the percentage error of their lidar compared to satellite observations but this is a good catch regardless. Their results are unclear (you could argue there's a small reduction in $N_2O$, but it's very small) and we will remove this reference in our revised manuscript.

809-810: Why does their box 1 not cover the high-strike-intensity area to the west?
Good question. Perhaps as this region is in Pakistan rather than India, which is the focus of their study, but such speculation is out of the scope of this review!

Figure 15: Please define the acronyms and what the signs mean in the caption.
Yes, we will do this.

901,904: Roy (2006) reports negative correlations with PDO and ENSO over India as a whole so presumably Western Himalaya is an exception to this? Is this based on their Figure 4?
Yes, that is correct. They report positive loading (and hence, in their case, negative

correlation) over most of peninsular and central India, but the sign is reversed in both the northwest and northeast, hence a positive correlation.

958: Is this based on the positive correlation between NAO and temperature during cold periods (their Figure 7b) and cold periods being linked with increased winter precipitation?
Yes, that is correct. The authors use this interpretation themselves in their own Fig 6c, where they associate cold reconstructed temperatures with strong winter precipitation and vice versa.

965: Are you suggesting that paleoclimate studies are less reliable because the historical climate is more difficult to observe?
I think the reviewer means that the paleoclimate is more difficult to observe? If so, then yes – in part. But it may also be that the relationship between the NAO and winter precipitation has changed, as we have seen for the ENSO-summer monsoon teleconnection over the last century. We can clarify this if needed, but I'm not sure it's necessary here.

1087-1088: Is the bias weak (as in small) or negative (as in winds being too weak)?
The latter. We will rewrite this more clearly in our revision: "negative wind bias".

1131: Seems odd that heavy precipitation is not mentioned in the list of hazards, although I accept you want to emphasise the less commonly considered ones.
Agreed. We will add it.

Figure 16: Should these be accessible from the website given in the caption? I was not able to find them easily.
We agree, unfortunately the IMD does not have a standard archive of their historical weather warnings to reference. Perhaps the easiest option is their official X/Twitter account, which we will also reference in revised figure caption.

1219: I would argue that the issue of how to forecast them (i.e., shorter range) is of similar importance (see also lines 1862-1866).
Agreed, we will rephrase this in the revision to "One of the most important questions on WDs is how they respond to climate change"

Figure 19: Does the horizontal axis increase into the future or into the past?
This is a paleoclimate figure, so larger numbers indicate deeper into the past, as is typical in that discipline. We will add this to the caption for clarity.

1329: Don't these studies look at somewhat later periods than 3500 to 1500 years ago?
Our original statement was not well phrased, "this period" refers to the late Holocene, not specifically 3500-1500 years ago. We will correct this. However, the

cited studies here do span the late Holocene: Kotlia et al (2012) covers the last 400 years; Sanwal et al (2013) the last 1800; and Kotlia et al (2015) the last 4000.

1383: Wasn't the link to global warming in Munz et al. (2017) with the weakening IWM?
Yes, that is one of the key results of that paper. However, in the final paragraph of their results section, they discuss the weakened teleconnection and hypothesise that arises due to a GHG-driven increase in the strength of local circulation.

1396-1397: What makes these two different from the other studies in blue in Figure 20?
They were published prior to (and thus included in) the last WD review, so we do not cover them in as much detail as the other studies. However, for completeness, they are included in Fig 20.

1416: "Earlier studies have suggested a decline in WD frequency": is this based on the two black minus signs (and no black plus signs) amongst the blue studies from Figure 20?
Not quite, this is based on the trends from (blue) studies from 2015 or earlier (two grey and one black minus; one grey plus). We will clarify this in the revision.

Figure 20: Presumably the different shades of blue/red/green are just to differentiate the studies and don't have any other meaning?
Correct. We will update the caption to clarify this: "Different shades of blue, red, and green are used only to differentiate between studies."

1494-1495: Looks like something has gone wrong with the text here so that the meaning is not clear.
Thank you – looks like some text got deleted here. We will revise this to: "Once datasets or regions with spurious behaviour are removed from the analysis, the key issue is decadal variability -- meaning the results are sensitive to the choice of analysis period. This was highlighted by Baudouin et al (2020), who found a regional minimum in winter precipitation between 1995 and 2010."

Figure 22: What do the grey contours (that are not very clearly visible) represent?
These have the values as the filled contours, plus an extra set for zero. We agree this is quite messy and will remove them in our revision.

And can one tell from these panels where the gauges are for each dataset?
Unfortunately not. Authors of the IMD dataset do not make the gauge data location available (except through figures in their paper). The APHRODITE authors similarly only make gauge location available through their paper figures, although they do release a gridded gauge density product. Unfortunately, density is not easily included in this figure.

1519-1520: Does this mean that we know which of Pai et al. (2013) and Chauhan et al. (2022) provide the correct interpretation?
The disagreement is between Nageswararao et al (2016) and Chauhan et al (2022). Pai et al (2013) is the dataset both used. Having re-read both studies, we realise they largely agree and that the trends vary by region. We will therefore rewrite this section to improve clarity and better bring out the role of interdecadal variability:

"The role of decadal-scale variability is most clearly highlighted by opposing results of long- and short-term studies. Using gridded gauge data from 1901 until present, both Nageswararao et al. (2016) and Chauhan et al. (2022) found generally positive trends in winter precipitation over north India. However, as in our Fig. 22, those studies that have measured their trends over comparatively short periods (~40 years) (Shekhar et al., 2010; Zaz et al., 2019; Ullah et al., 2022; Abbas et al., 2023; Safdar et al., 2023) instead report a significant decline in winter precipitation across north India and Pakistan. These studies are therefore likely to be detecting a mode of interdecadal variability. Long term studies of aridity during the rabi season (i.e. the winter months) have also indicated a trend towards wetter conditions over northern Pakistan in the regions typically affected by WDs (Ahmed et al., 2018, 2019), although this too appears to be subject to significant interdecadal variability (Ullah et al., 2022).

"Those studies reporting declining trends in winter precipitation typically invoke declining WD frequency and shifts in subtropical jet position as the cause, as did Gunturu and Kumar (2021), who argued that a recent decline in WDs has been responsible for reduced cloud cover and increased fog over the recent decades."

1541,1556: The formatting implied by the bold text is not clear here.
Yes, these were supposed to be new subheadings but EGU journals cap to three levels. We will remove these two bold subheadings.

1692-1693: Meher & Das (2022): is this based on their Figure 5 (standard deviation)? Are you arguing that an increased standard deviation implies an increased mean?
Yes, although we believe that this is a reasonable assumption as precipitation tends to follow a gamma distribution, we will clarify this in the revised text thus: "Among these, Midhuna and Dimri (2022) and Meher and Das (2022) also reported changing seasonality, finding a relatively larger increase in mean precipitation during the late winter (February) and spring (March and April) respectively, and thus providing further evidence for a lengthening of the active WD season due to climate change."

Figure 23: It might be helpful to move this forward a bit, nearer to where it is first referred to in the text.
We agree, but at this stage the location of figures is mostly set by LaTeX. We will encourage it forward, but ultimately this can be fixed by typesetters during production.

Figure 24: Which study are the solid coloured lines from?

There is a mistake in the caption – thanks for pointing it out. We will fix this (Hunt 2019b – coloured lines for CMIP5 RCPs and black line for CMIP5 historical).

1833: Does this imply that extreme WD precipitation rarely occurs in the core WD season? Or could this interaction occur over a timescale of a few months?

No, such events are comparatively rare (and interactions with the monsoon occur more-or-less simultaneously). Note the original text reads "rainfall", not "precipitation".

I would also like to make the following typographical recommendations/comments.

103: I would suggest changing "avalanches" --> "and avalanches" to improve comprehension of the sentence here.

Agreed, we will make this change.

150: "rather" --> "rather than"

Agreed, we will make this change.

251: The word "weather" appears twice: I think this is a mistake?

Agreed, we will make this change.

336: Could change "greater" --> "higher" just to make absolutely clear that you don't mean higher pressures (and thus lower altitudes).

Agreed, we will make this change.

432: One of the "delta"s has not been rendered correctly.

Both seem to be ok in the PDF version of the manuscript given to reviewers. Perhaps the issue is that we write 8$\delta$, which we now change to 8·$\delta$ for clarity.

483: "occur" --> "occurring".

Thank you, we will fix this.

1005: Please define IWM here.

This shouldn't be here – we will remove this.

1333: "differential": do you mean "different"?

Our intention with "differential" here was to say that the studies in question compared speleothems in different locations. But we agree this is confusing and will remove it.

1345: Please define LIA here

This is the "Little Ice Age", which we will explain in the revision.

1451: "if" --> "of"
Thanks, will fix this.

1783-1784: MJO is listed twice.
Thanks, will fix this.

---

## Author Comment (AC4)

Review of EGUsphere 2024-820 "Western disturbances and climate variability: a review of recent developments" by Kieran M. R. Hunt et al.

Synopsis:

This review paper by Hunt et al. aims at providing an overview of recent research on the topic of western disturbances (WDs) over the Indian subcontinent. The review addresses many aspects such as the structure and dynamics of WDs, natural hazards associated with them, their predictability and their response to climate change. Overall, the paper is well written and it covers many aspects. However, the paper appears to be overly detailed in places and it is difficult for the reader to identify the essential points. Accordingly, I suggest some points for revision before the paper can be published.

We would like to thank the reviewer for their positive assessment of our manuscript, and for their detailed comments, which we respond to point-by-point in red below. Planned revisions to our manuscript will be highlighted in blue.

Comments:

1) My major comment is that several subsections have a 'list' structure summarizing one result after another (e.g., 4.3, 4.2, 3.4.1, 7.1.3).

We will completely rewrite Sec 3.4.1 (which will be 3.4.2 in the revision) to provide a shorter and clearer narrative and thus to better synthesise the lists. We will also do this for 4.2, 4.3, and 7.1.3 as requested, as well as several other sections requested by other reviewers.

Thus, the storyline of the review does not necessarily become clear and it may be helpful to decide on a consistent conceptual framework for each subsection early in the paper. For example, a brief statement at the beginning of each subsection could describe our current understanding of a certain aspect of WDs and indicate the confidence that the research community has. The statement could then be followed by a more detailed summary. Still, this summary should not simply list all studies but for example comment on the confidence that we have or explain why several aspects are still uncertain.

We thank the reviewer for this comment. We will go through the paper and add short summary statements with IPCC-style measures of confidence, using their calibrated uncertainty language, throughout.

We will explain this in the introduction: "Each section starts with a summary of older literature, to orientate the reader and to provide context for the newer research. Each section then concludes with a short summary statement, including a measure of confidence in the main points, following the IPCC calibrated uncertainty language. These statements are summarised in Sec. 8.1."

[revised manuscript text omitted]

Overall, I would hope the paper to become more concise once the subsections with a 'list' structure have been revised.

Thank you, please see our response to your first comment, above.

2) Even for state-of-the-art reanalysis data sets, there are uncertainties on the fraction of precipitation that can be attributed to WDs. Accordingly, it seems to be even harder to quantify the fraction of precipitation associated with WDs for past or future climate states. Still, quite often the authors refer to studies establishing a link between precipitation and WDs. For such statements, it would be important to

explain at least briefly how the authors come to the conclusion that a clear link between WDs and precipitation exists even if the WDs have not been identified objectively in some cases (e.g., l. 1568, l. 1581, l. 1664).

We agree that this is not sufficiently well discussed in the text. Following this suggestion, and that of reviewer 1, we will add the following to the beginning of our paleoclimate section: "Paleoclimate research has become increasingly popular over the last few decades, especially as more advanced proxy techniques have been developed and refined. For precipitation, these include speleothems, marine and lake sediments, tree rings, and pollen analysis. As we discussed in Sec. 3.2.2, present-day WDs are responsible for the majority of total winter precipitation over the Western Himalaya and surrounding region and likely – through changes in WD frequency and intensity – the majority of its interannual variability as well. For these reasons, precipitation is often used in paleoclimate studies as a proxy for WD activity over the Western Himalaya. However, there are several important sources of uncertainty that arise with this approach. Firstly, the relative contributions of winter precipitation (i.e., WDs) and summer precipitation (i.e., the monsoon) to the annual total may change over time. However, this uncertainty can largely be removed by quantifying the d-excess of the sample studied (see Sec. 2.4). Secondly, as mentioned above, some winter precipitation variability must arise from non-WD sources, the primary source of which is cloudbursts. The fraction is unknown, but probably small, and may also have varied over long time periods. Thirdly, analyses often make do with proxies from winter precipitation dominated areas nearby (e.g., Iran, central Asia), and extrapolate the result to the study area (e.g. Petrie and Weeks, 2018). Thus, while we can be reasonably confident that long-term changes in winter precipitation are related to changes in WD activity, we must bear these caveats in mind when discussing the results of the paleoclimate studies that follow."

l. 92: Can you comment here or later on what processes lead to the WD cyclogenesis?

Yes, we will add the following sentence here in the revision: "They then propagate as troughs embedded within the subtropical westerly jet until they reach South Asia (Singh et al., 1971)."

l. 136-142: In principle, I agree. But what about the fact that WDs travel along distance before they reach India/that they are embedded in the large-scale weather? Wouldn't this require global models?

Yes – some studies do require global models. However, many WD modelling studies tend to focus on their impacts (as we mention later in Sec. 5) and thus tend to be high-resolution limited-area models forced at the boundaries by reanalyses. We will revise the wording here to make this distinction clearer: "Research on WDs, especially of their impacts, benefits from these developments, not only because the models are now often convection-resolving…".

l. 169: Please specify that it is cyclonic potential vorticity anomalies.
Yes, will do.

l. 175: What about diabatic processes? I assume these also play a crucial role in the development of mid-tropospheric PV anomalies of WDs.
Yes, this is certainly correct (as we discuss later in Sec 2.3.2) although not much covered in earlier research. We will add that, including a new reference, here: "As deep troughs, WDs are associated with high vorticity in the mid-troposphere which can be further enhanced through orographic interaction and diabatic heating (Rao, 2003; Hara et al., 2004; Dimri and Chevuturi, 2014b)."

l. 229 and elsewhere: Please double-check whether all acronyms have been introduced before their first use.
Thank you for this comment – we have identified all unique abbreviations in the text using regular expressions and will ensure the first instance of each is explained in the revision. We will introduce meteorological terms (e.g. NAO, CAPE) in the text itself and use footnotes for product names (e.g. IMDAA, APHRODITE), as these can be lengthy and reduce readability.

l. 230: Are you referring to the layer mean relative vorticity between 450 to 300 hPa.
Yes, this is correct. We will include this in the revision.

l. 225 & l. 232: The study regions of WDs differ. Would it be an important future step to agree on one region across the the WD science community? Likewise, the minimum lifetime seems to be quite variable. Also concerning this aspect, a critical discussion of the different criteria would be well suited in a review paper (see also comment on Fig. 5).
We agree such a discussion would be valuable and will add the following: "The reader will have noticed that each of these studies uses different criteria -- both capture regions and minimum track lengths or durations -- to filter their WDs. This is because no standard yet exists, and so authors typically choose their capture regions to reflect the impacts they want to investigate. This makes intercomparison between studies challenging, as even basic statistics such as frequency are sensitive to these choices. Therefore, based on the discussion in this section, we propose basic criteria that could be adopted in future WD tracking studies in order to standardise the results.

Firstly, rather than a capture region, where the choice of longitudinal extent can have a significant impact on the characteristics of the WDs in the final catalogue, we propose simply that WDs must cross 70°E (to the east of almost all genesis areas, see Sec. 2.3.1; and to the west of the regions of greatest impact), and do so between 20°N (to filter out tropical systems) and 50°N (to filter out polar systems, but retain northward-tracking WDs that can still have an impact over the Karakoram or Pamirs). We also propose a minimum track duration of 48 hours to filter out

transient systems; but no minimum track length, as WD genesis can occasionally be very close to the Western Himalaya."

l. 239-252: Though I appreciate the authors attempt to list existing techniques to approximate WD frequency and related statistics, it may be more important to the reader to understand what the implications of the different techniques are. For example, what fraction of uncertainties in WD statistics (number per year, speed etc.) can be attributed to different tracking techniques.
Thank you. We agree that such a discussion would be very valuable. However, the uncertainties in WD statistics are sensitive not only to the choice of these techniques but also to thresholds used (e.g., the spectral power/variance must exceed some value for it to be considered a WD day). Therefore, to quantify these uncertainties would require replicating each of these studies, which we deem to be beyond the scope of this review. We will add the following sentences in the revised version of the manuscript explaining this: "This wide range of indirect techniques leads to a wide range of estimates of WD statistics, such as frequency. Quantifying these uncertainties is made significantly more challenging by the sensitivity of each method to the cutoff thresholds used to define WD activity (e.g., variance), and so they are difficult to compare directly."

l. 277: Is it really the case that disturbances are blocked by the Tibetean Plateau? For example, taking relative vorticity at 400-300 hPa, I'd be surprised if the systems were blocked by the Tibetean Plateau. Is it not rather the case that flow configurations advecting disturbances southward do hardly occur? Or are you referring here to disturbances near the surface?
WDs can be blocked by the Tibetan Plateau (although this phrasing is slightly misleading – rather the jet is only stable on either side of the TP, not on top of it). However, you are indeed correct in that this is actually an issue of flow configuration, and we will correct this accordingly in our revision.

l. 292: Diabatic heating can also occur over a prolonged period and with considerable latent heat release in stratiform precipitation (e.g., in WCBs). So, is it a necessary condition for the intensification that WDs are associated with convective precipitation?
This is a good question and one that does not appear to have been addressed in previous literature. There is no reason that stratiform precipitation couldn't also play the same role, and as we see in Fig 6, stratiform clouds are not uncommon along the foothills in the winter. We will add the following sentence: "Stratiform clouds are also common along the foothills (Fig. 6) and therefore may also play a role in WD intensification, but this has not yet been investigated."

l. 304: Through which process does latent heat release increase the strength of the upper-level (cyclonic) PV anomaly? If the latent heat release occurs in the mid-troposphere, it would rather lead to an anticyclonic PV anomaly in the upper

troposphere, i.e., reduce the strength of the upper-level (cyclonic) PV anomaly. Perhaps you can also refer to lower and upper troposphere, instead of lower and upper level.

This is a good point and arises from an earlier misreading of Para et al (2020). The original sentence reads: "However, Para et al (2020) used a case study of extreme precipitation over Jammu and Kashmir in 2017 to show that the broad quasigeostrophic ascent can be coupled with convection, as latent heat release increases the strength of the upper-level PV anomaly." We will revise this to:

"However, Para et al (2020) used a case study of extreme precipitation over Jammu and Kashmir in 2017 to show that WD circulation can be coupled with convection, as latent heat release increases the strength of the lower-tropospheric PV anomaly."

l. 342: Are you referring to divergence in the upper- or lower troposphere? Also, I found it very difficult to follow the line of arguments here. If there is a negative correlation between propagation speed and cloud-top height, would this not mean that systems with low cloud-top height propagate faster, and those with high cloud-top heights propagate slower? This at least would be dynamically understandable: Systems with high cloud-top heights are associated with stronger upper-tropospheric divergence and a corresponding divergent outflow which would be directed against the eastward propagation of the WD. If it is the case that WDs with stronger convection tend to propagate more quickly: Is this because of diabatically generated lower-tropospheric cyclonic PV similar to diabatic Rossby waves/vortices?

This was an error in the original manuscript. We meant to say that the correlation is positive and will fix this in the revision. We will also include the dynamic reasoning provided.

l. 359: The quasi-geostrophic ascent could also occur before the moisture reaching the orography. Is this also observed?

Yes – as is the convection. We will rephrase this sentence to make this clear:

"Orography is not required for WDs to precipitate, as neither the convection nor QG forcing depend on it."

l. 370: It reads as if there is "orographic instability" which is certaintly not meant here. Please clarify.

We will remove the parenthetical here.

l. 408: Please clarify trough which process the transport of ozone is happening since a PV anomaly per-se does not necessarily lead to stratosphere-troposhere exchange.

Good question. According to the original paper (Satheesh Chandran et al., 2022), "The equatorward intrusion of high PV associated with the western disturbance (Figure 8d) facilitates the transport of mid-latitude lower stratospheric air deep into the tropical latitudes. Associated with this intrusion, a strong subsidence is observed in the eastward side and upwelling in the westward side (Figure 8c)." We will clarify

this in our revision: "As in extratropical cyclones, the deep PV anomaly associated with WDs -- which extends to the tropopause -- can result in substantial transport of ozone through advection of mid-latitude stratospheric air into the tropical upper-troposphere (Satheesh Chandran et al., 2022)."

l. 439: How exactly could a moisture flux analysis supplement the isotope analysis? Moisture flux analysis provides only a Eulerian viewpoint and does therefore not provide information on the actual moisture source.
This is certainly true for individual events, but the Eulerian approach does work for seasonal composites. We will remove the usage here, and then clarify its inclusion at L475.

l. 475: See previous comment.
We will clarify that moisture flux is valid in this context only on longer timescales: "Beyond isotope and trajectory methods, recent work by Baudouin et al (2021) highlighted the utility of composite moisture flux analyses in investigating precipitation moisture sources, with the caveat that such analysis only works on seasonal timescales or longer."

l. 620: Is this a general statement or specific to the region affected by WDs?
It probably does hold generally, but was intended for the study region. We will clarify accordingly in the revision.

l. 655: That the percentage of precipitation attributed to WDs varies substantially between studies calls for a consistent approach when matching WDs and rainfall. Have there been approaches where the distance at which rainfall is still attributed to a WD is based on objective criteria such as the Rossby radius of deformation? A critical discussion would be worthwhile here.
This is a good question. Unfortunately, the studies discussed here are the only one that have attempted to quantify the attribution fraction. The methods used in these studies (either taking a fixed radius or all precipitation on the day – i.e. an infinite radius) follow earlier studies attempting to do the same for monsoon depressions. My own (unpublished) experience is that the answer is relatively insensitive to the choice of radius, if sufficiently large, since almost all precipitation that falls on WD days is due to the environment created by the WD itself. The other main source of winter precipitation in this region is cloudbursts, which are either triggered by WDs or occur independently on other days.

The reason, we believe, that Midhuna et al (2020) only managed to explain 20% of the monthly precipitation *variance* is because they *only* considered whether precipitation fell on a WD day, when in fact there is also a strong correlation between WD intensity and precipitation, as we discuss in Sec 2.6.

What is clear, then, is that WDs cause a majority of total winter precipitation in the Western Himalaya and surrounding regions, with a conservative lower bound of

55% and an upper bound of over 90%. What remains unknown, though, is the fraction of precipitation variance for which WDs are responsible across different timescales.

We will rephrase and extend the final paragraph to reflect these points: "These results suggest that the true value of the seasonal winter precipitation contributed by WDs is likely to be somewhere between the values stated by Hunt et al. (2019a) and Midhuna et al. (2020), but it is certainly a majority. The uncertainty likely arises not from the choice of attribution radius – to which attribution fraction has been found to be relatively insensitive for other types of system (e.g. Hunt and Fletcher, 2019) – but from the method used to detect WDs. This is because the other main source of winter precipitation in this region is cloudbursts -- very intense but short-lived thunderstorms that drive highly localised extremely heavy precipitation. Cloudbursts are either triggered by WDs or occur independently on other days (Singh and Thapliyal, 2022; Dimri et al., 2023), and so almost all precipitation on WD days arises from the environment created by the WD. What remains unknown, however, is the fraction of precipitation variance across different timescales for which WDs are responsible. There is evidence that this is probably a large majority on intraseasonal timescales: firstly, because WDs are responsible for most of the seasonal total and their frequency and intensity varies substantially between different years (Sec. 2.6); and secondly, because WDs are responsible for 90% of extreme winter precipitation events over the Western Himalaya and northern Pakistan (falling to only 20% in the summer). Further work is needed to quantify this explicitly, and in so doing validate results from climate and paleoclimate studies that infer changes in WD activity directly from precipitation".

l. 730: I am wondering whether the heading of section 3.4 would be a better one for Section 3. Precipitation can lead to natural hazards so the separation between precipitation and other natural hazards seems a bit arbitrary.
We agree and will make this change.

l. 739: What is the reason for the cool ground?
Winter. The overnight lows in Delhi and Chandigarh both average about 7°C, both with record lows around freezing. We will clarify this by adding "wintry" in the revision.

l. 750: Are there already insights on why the boundary layer turbulence is suppressed in the rear sector of WDs? Is it due to descending air masses causing an inversion layer that prevents the downward mixing of momentum.
This is likely one of the reasons, although it is not stated in Patil et al (2020) (they attribute it to surface cooling) and we couldn't find it in earlier literature either. We will add a short statement in the revision to reflect this: "Patil et al (2020) attributed this reduced turbulence to surface cooling induced by the WD, but it may also be

due to descending air masses causing an inversion layer that prevents the downward mixing of momentum."

l. 758: Though I am not an expert in this field, I would expect that the increasing pollution is the most important factor.
Please see our response to the next comment.

l. 770: This somewhat confirms my previous statement.
Agreed. We will include this in the revised version of the manuscript (bearing in mind that this section will also be revised considerably): "Fog has increased significantly over north India in recent decades (Hingmire et al., 2019), as much as doubling since 1970 (Srivastava et al., 2016). Most studies agree that this is primarily due to increased aerosol loading and urban expansion (Smith et al., 2022; Hingmire et al., 2022; Verma et al., 2022). However, WDs may also play a role – a recent reduction in WD frequency, and hence weaker near-surface winds, increased radiative cooling, and reduced precipitation, has also been linked to increased pollution over north India, both in models (Paulot et al., 2022) and observations (Gunturu and Kumar, 2021; Xie et al., 2023)."

l. 735-786: This section needs to be revised substantially. It currently reads as a collection of literature, but due to the partly opposing research results it is difficult to develop a conceptual picture. A different approach would be to rather summarize the findings about which we are certain and then to mention the uncertainties which still need to be quantified.
Thank you for this suggestion. Following this and your first major comment, we have revised and shortened this section substantially.

l. 805: Can you explain what is meant by "nor'westers"? It reads like a phenomenon associated with strong winds, but this would not fit to the section dealing with lightning and hailstorms.
These are squally northwesterly winds that impact Bengal and the surrounding region. We will add a sentence in the revision that states they are associated with both hail and lightning: "Pre-monsoon storms can also occur near the Himalayan foothills in northeast India and Bangladesh, where they are associated with nor'westers, known locally as *kalbaisakhi* (Roy and Chatterji, 1929; Das et al., 2014). These kalbaisakhi often bring heavy hail and lightning (Midhya et al., 2021)".

l. 890: A further shortcoming might be that the role of lower-level PV maxima and upper-level PV maxima has not been quantified yet. The concept of piecewise potential vorticity inversion would be one diagnostic to assess the role of different PV anomalies.
Agreed, we will add this in the revised manuscript: "Another shortcoming was that the roles of lower- and upper-tropospheric PV maxima were not quantified. Further work might use piecewise PV inversion to assess the role of different PV anomalies."

l. 898: Though I agree that SAM, NAO rectify onto SSTs with similar patterns to those shown here, it is still questionable whether a physical link exists. For example, through which physical process would SAM be connected to the WD occurrence frequency? This aspect definitely needs some explanation.

This is a good question. We already discuss in the text potential causal mechanisms linking the NAO to WD frequency, e.g., "A composite analysis by Hunt and Zaz (2021) found that winters with a strongly positive NAO resulted in a stronger subtropical jet, which in turn forced more frequent and more intense WDs, driving increased winter precipitation over the Western Himalaya and Indus Basin" and "[Attada et al, (2019)] showed that warmer winters there are associated with negative phases of the NAO and AO, which decrease the upper-level meridional temperature gradient, weakening the subtropical jet and decreasing WD frequency."

An extended discussion of the SAM is out of scope for this review, as no studies have explicitly linked it to winter weather in our region of interest. However, there is literature linking the SAM to variability in the summer monsoon, and we will add a very brief discussion of that to the section summary in our revision: "Despite its distance from the subcontinent, there does appear to be a link between the SAM and the both the East Asian and South Asian summer monsoons (Pal et al., 2017; Fogt and Marshall, 2020), and so a connection with the winter weather of these regions is also plausible."

l. 1012-1019: These lines should not be part of section 4.4 since they summarize Section 4 in total.

Agreed. As there is no way to leave a subsection without creating a new one, we will use a blank line to separate this summary in the revision.

l. 1122: Could you explicitly state which land surface datasets were found to yield superior results? Or do you only want to state that the representation of WDs is more sensitive to the land surface dataset than to model parametrisations.

Although a discussion of which land surface dataset is "best" would be interesting, our purpose here is the latter. Generally, newer datasets tend to perform better, but as the cited studies only test different pairs, this would probably require further research.

l. 1165: Are these short-range forecasts deterministic? If so, can you comment on the added value of ensemble forecasts and this is one way forward?

Yes, the short-range forecasts produced by the IMD (from which they draw these warnings) are deterministic, either from a 3-km WRF model or a T1534 (12 km) version of the GFS. We will add a short comment on this in the revision (at the end of this subsection): "These warnings are derived from deterministic short-range operational forecasts, and so one approach to improve skill may be to use an ensemble forecast – which are often better at capturing extremes – instead

(Boucher et al., 2011). However, a full analysis of warning bulletins and nowcasts in the context of WDs is left as an important topic for future work."

l. 1200: Extended range and subseasonal forecasts often use hybrid approaches combining statistical and dynamical models. Have there been insights on whether statistical models for the occurrence of WDs are useful? For example, though their connections to ENSO, NAO etc. there could be valuable predictors on this longer time scale.

This is a very good point. These kinds of forecasts exist for the summer monsoon, but not, to the best of our knowledge, for winter precipitation or WD activity. We will note this both here (Sec 6.4) and in Sec 8 as an important avenue for future research: "One such avenue might be the use of hybrid statistical-dynamical forecasts, such as those already used for the summer monsoon (e.g. Rajeevan et al., 2007), which could leverage additional predictability offered by teleconnections to ENSO, NAO, and other large-scale modes of variability."

Section 7: Overall, Section 7 needs to be shortened. On several occasions the link to WDs is not clear and it is difficult to synthesize all the given information to form a consistent picture.

Thank you for the suggestion. In the revision, we will rewrite sections 7.1 and 7.2.1 to shorten them, make the links to WDs clear, and synthesise the key themes of the relevant literature.

Further, it is not clearly explained how WDs are linked to precipitation in paleo-climate studies. Such explanation is necessary given the difficulty of identifying WDs even in state-of-the-art reanalysis data sets.

Thank you for raising this point. Please see our response to your major comment #2.

l. 1220: Could you include an initial hypothesis on why a response of WDs to climate change is expected?

Yes, we will add the following to the beginning of our revised Sec 7: "As the climate warms, we expect changes to WD dynamics -- as inhomogeneous upper-tropospheric warming modifies the subtropical jet. We also expect thermodynamic changes to WDs, as warmer near-surface conditions modify static stability and increase atmospheric moisture content."

l. 1541: The "elevation-dependent warming" can presumably be removed.

Thank you, this was errant subsection labelling that we will remove in the revision.

Fig. 2: Please include state borders.

We use orographic contours here primarily as that is the strongest control on mean precipitation patterns. However, we appreciate state borders are more familiar to most readers and will include them in our revision (see figure below).

[Figure]

Fig. 3: WDs are tilted northwestward with height. Is there a reason for not showing this tilt?

Yes – this is a good point. We decided not to show the tilt because it is relatively slight (see Fig 8 and related discussion) and the schematic figure is already quite complicated.

Fig. 5: To my understanding a "commonly used WD track capture region" does not really exist in literature and the definition of regions varies from paper to paper. Please reconsider the formulation.

This is correct. We will rewrite the caption so that it reads "The black box indicates the WD capture region used in some tracking studies (60—80°E, 20--36.5°N). The dotted box shows the extension to 42.5°N which has also been used in some more recent studies."

Fig. 10: Given the stronger jet during winter, it is probably not too surprising that the WDs are stronger in winter. If the intensity was normalized with the seasonal mean vorticity, would this show intense WDs also during summer?

Yes, by the definition of normalisation, this would indeed be the case if we normalised summer WDs by the summer mean WD vorticity.

---

## Referee Report (RR1)

I would like to thank the authors for their comprehensive response to my comments and their thorough review of the manuscript based on the comments of all the reviewers. There were two minor points to make from the authors' responses:

Line 88: this is now "an eastward".

Lines 123—124: "Key characteristics of the WD, along with its environment, are summarised in Fig. 3" – would it read better if this sentence was kept to being the last one of the subsection (i.e., so that the two new sentences are inserted before it)?

I feel that the paper is almost ready for progressing to typesetting from my point of view, but I would first like to follow up some more of the citations. Ideally, when reviewing a regular article, a reviewer will have a good general overview of the relevant literature themselves and will only need to follow up a relatively small number of citations. However, I think for such a long and wide-ranging review article such as this one it is not feasible to find reviewers who would not need to follow up a huge number of citations, making the exercise impossible within the normal constraints of performing the role in the margins of a regular scientific job. Therefore, in my initial review I took a statistical approach to following up the citations, by checking in detail roughly 8% of them (chosen as far as possible at random); this is of course in addition to reading through the manuscript and having in mind my general knowledge of the literature (which of course varies in its extent for the different parts of the paper!).

My thinking was that, if all reviewers take the same approach, and no errors/discrepancies are found (or only a very small number are found and corrected), then one can be reasonably confident that the total number of discrepancies is very small. Five such discrepancies were found and corrected just from my check of 8%; it is of course true that they were all fairly minor and do not change the overall conclusions of the paper so I don't think this is a major concern. However, since the review article is intended to act as a reference, synthesising the available literature, it is important to ensure that it provides as accurate a representation of previous work as possible. I would therefore suggest that the authors go through the citations carefully and ensure that there are no discrepancies as far as they can; additionally, I have repeated my "statistical" exercise for this second review and followed up 10% of the citations (spread evenly throughout the paper). Most of these I can see are correct, and for the remaining ones it would be useful if the authors could address my concerns listed here: it may well be that in most cases I have misinterpreted or not found the relevant statement/plot in the paper.

Line 231: Is the interpretation that some of the extratropical cyclones in Wernli & Schwierz (2006) fulfil the same criteria used to define WDs but do not reach as far south/east? Is this something demonstrated in the article or something you have followed up with the data set?

Line 485: I could not find where D-excess values explicitly in the Arabian Sea were referred to by Jeelani & Deshpande (2017). Is this based on interpreting D-excess values from some of the regions they mention as having a source in the Arabian Sea?

Line 592: Where do Riley et al. (2021) mention feeble/weak WDs in particular?

Line 660: Don't Thayyen et al. (2013) investigate an August flooding case (rather than pre-monsoon)?

Line 834: Hingmire et al. (2022) look at future change; where do they say that the recent increase in fog is due to increased aerosol loading and urban expansion?

Lines 1127-9: Patil & Kumar (2016) only show maps for the best model don't they, so how can you describe the biases in this detail for all experiments? And I would say that their Experiment 5 led to some fairly substantial improvements in the precipitation RMSEs (their Table 4), but overall "They found only a low sensitivity to the choice of microphysics scheme." is probably a fair statement to make!

Lines 1220-1: Das et al. (2003) do recommend using 10 km spacing to improve forecasts but I couldn't find where they explicitly demonstrate that the biases can be significantly reduced by dynamical downscaling.

Lines 1363-6: It seems strange that the same authors would make contradicting statements in different works (I wasn't able to access the book reference).

Line 1426: Where does Lone et al. (2022b) state that it was warm and dry for these particular years (maybe I got mixed up converting from their "years since the present" value to CE/BCE!)?

Lines 1472-3: Where to Singh et al. (2015) mention the elevation dependence?

---

## Author Response (AR2)

We reiterate our sincere thanks to all reviewers, especially the two that have followed up below. Our responses are again given point-by-point in red, with material changes to the manuscript denoted in blue.

**Response to reviewer 2 (report 3)**

I would like to congratulate the authors on their careful and comprehensive revisions. My previous concerns were addressed in a suitable manner and I am convinced that the review article will be of great interest to the WCD readership. Before publication I still have some minor remarks that should be addressed:

l. 130: As a midlatitude dynamicist, I find it difficult to separate between dynamics of extratropical cyclones and upper-tropospheric troughs. In my view troughs and extratropical cyclones are intricately linked. So, does the statement mean that the dynamics of WDs share the same distribution of processes as those associated with extratropical cyclones?
Yes, this is correct – while WDs do share some features with extratropical cyclones, they are often missing others (e.g. clear frontal zones). As such, we argue that they sit on the continuum between upper-level troughs (e.g. Rossby waves) and ETCs. Either way, what makes WDs unique is their environment, as we state in L124-5: "Their unique proximity to both mountains and a maritime moisture source, however, leads to a unique array of impacts that are not found in other regions."

l. 1041: It would be beneficial to include a suitable reference to where the concept of piecewise PV inversion is explained.
We agree and have added a reference to Davis (1992) and Bracegirdle and Grey (2009) in L964, where we first discuss PV inversion.

l. 1415: Please clarify that the Keller et al. 2019 paper deals with tropical cyclones undergoing extratropical transition but not with extratropical cyclones per-se.
We assume the reviewer means the Keller et al. (2019) reference on L1277 (since it doesn't appear elsewhere). We have made the suggested change.

**Response to reviewer 3 (report 4)**

I would like to thank the authors for their comprehensive response to my comments and their thorough review of the manuscript based on the comments of all the reviewers. There were two minor points to make from the authors' responses:

Line 88: this is now "an eastward".
Thank you, we have made this correction.

Lines 123—124: "Key characteristics of the WD, along with its environment, are summarised in Fig. 3" – would it read better if this sentence was kept to being the last one of the subsection (i.e., so that the two new sentences are inserted before it)?
We agree and have made this change.

I feel that the paper is almost ready for progressing to typesetting from my point of view, but I would first like to follow up some more of the citations. Ideally, when reviewing a regular article, a reviewer will have a good general overview of the relevant literature themselves and will only need to follow up a relatively small number of citations. However, I think for such a long and wide-ranging review article such as this one it is not feasible to find reviewers who would not need to follow up a huge number of citations, making the exercise impossible within the normal constraints of performing the role in the margins of a regular scientific job. Therefore, in my initial review I took a statistical approach to following up the citations, by checking in detail roughly 8% of them (chosen as far as possible at random); this is of course in addition to reading through the manuscript and having in mind my general knowledge of the literature (which of course varies in its extent for the different parts of the paper!). My thinking was that, if all reviewers take the same approach, and no errors/discrepancies are found (or only a very small number are found and corrected), then one can be reasonably confident that the total number of discrepancies is very small. Five such discrepancies were found and corrected just from my check of 8%; it is of course true that they were all fairly minor and do not change the overall conclusions of the paper so I don't think this is a major concern. However, since the review article is intended to act as a reference, synthesising the available literature, it is important to ensure that it provides as accurate a representation of previous work as possible. I would therefore suggest that the authors go through the citations carefully and ensure that there are no discrepancies as far as they can; additionally, I have repeated my "statistical" exercise for this second review and followed up 10% of the citations (spread evenly throughout the paper). Most of these I can see are correct, and for the remaining ones it would be useful if the authors could address my concerns listed here: it may well be that in most cases I have misinterpreted or not found the relevant statement/plot in the paper.

We thank the reviewer again for their attention to detail, which has certainly made the manuscript more robust. As we hopefully demonstrate below, the discrepancies identified in this round only uncover one material error and two overconfident syntheses.

Line 231: Is the interpretation that some of the extratropical cyclones in Wernli & Schwierz (2006) fulfil the same criteria used to define WDs but do not reach as far south/east? Is this something demonstrated in the article or something you have followed up with the data set?

Wernli and Schwierz (2006) used closed contours of MSLP to track extratropical cyclones. This does capture some WDs, but only those with a sufficiently pronounced surface low (which is not all systems, as we state in L121: "WDs [are] usually associated with a weaker surface low"). That WDs, as a distinct phenomenon, are present in the W&S dataset is visible in their Figure 4.

Line 485: I could not find where D-excess values explicitly in the Arabian Sea were referred to by Jeelani & Deshpande (2017). Is this based on interpreting D-excess values from some of the regions they mention as having a source in the Arabian Sea?

Yes, that is correct – the authors directly attribute high D-excess in moisture sources to their Arabian Sea origin.

Line 592: Where do Riley et al. (2021) mention feeble/weak WDs in particular?

Our sentence "Although feeble WDs are only associated with light precipitation, they are sufficiently frequent that they comprise a large fraction of the total seasonal precipitation (Riley et al, 2021)" is deduced from their Fig 5, where they show that only about 40% of all WD precipitation arises from 90% percentile precipitation events.

Line 660: Don't Thayyen et al. (2013) investigate an August flooding case (rather than pre-monsoon)?

This is indeed an error. This should be Thayyen et al. (2010), which we have now corrected.

Line 834: Hingmire et al. (2022) look at future change; where do they say that the recent increase in fog is due to increased aerosol loading and urban expansion?

We made this statement based on the following sentences in Hingmire et al (2022): "Figures 2 and 3 show the time series of daily fog fractions... for the study period 1981–2018 using the station observations.... One can clearly see fog fractions are predominantly below 0.5 during the pre-1997 period compared to post-1997 period, which is referred by Syed et al. (2012) and Kutty et al. (2020) as a regime shift in fog variability in the IGP region (Fig. 2). The causes for the regime shift in fog variability are still not clearly understood, however these studies rest their conjectures on the increasing anthropogenic aerosols as well as other regional and global changes in environmental parameters due to global warming." We have modified our statement slightly to reflect this uncertainty and the new Kutty et al (2020) reference: "Most studies agree that this is probably primarily due to increased aerosol loading and urban expansion (Kutty et al., 2020; Smith et al., 2022; Hingmire et al., 2022; Verma et al., 2022)."

Lines 1127-9: Patil & Kumar (2016) only show maps for the best model don't they, so how can you describe the biases in this detail for all experiments? And I would say that their Experiment 5 led to some fairly substantial improvements in the precipitation RMSEs (their Table 4), but overall "They found only a low sensitivity to the choice of microphysics scheme." is probably a fair statement to make!

Our original statement was made on the grounds that their best-performing experiment demonstrated these biases, and the RMSEs of precipitation and circulation were so similar with the other experiments that they most probably suffered from the same bias. However, we accept that this was overly confident and so we've revised our text slightly to read: "Even their best-performing experiment showed a southwesterly lower-tropospheric wind bias over the Western Himalaya, leading to significant biases in the location of heavy precipitation."

Lines 1220-1: Das et al. (2003) do recommend using 10 km spacing to improve forecasts but I couldn't find where they explicitly demonstrate that the biases can be significantly reduced by dynamical downscaling.

Again, we agree our statement was overly confident – it was based on a combination of them running a downscaling experiment (10-km mesoscale model forced at the boundaries by an 80-km global model) and then subsequently recommending use of the 10-km model. We have adjusted our phrasing to reflect the reviewer's comment: "Das et al. (2003) showed that such forecasts could be improved by dynamical downscaling – in their case using NCMRWF operational forecasts to drive a 10 km nested model."

Lines 1363-6: It seems strange that the same authors would make contradicting statements in different works (I wasn't able to access the book reference).

We agree this is surprising, but it is indeed the case.

Phartiyal et al. (2022): "The westerly [i.e., WDs] dominates in the beginning of Holocene".

Phartiyal and Nag (2022): "Recurrence of ISM dominance is seen from ~12 ka to Early Holocene." and "Early Holocene Khalling Glacial Stage suggests strong ISM [over Ladakh] driven by global insolation maxima"

Line 1426: Where does Lone et al. (2022b) state that it was warm and dry for these particular years (maybe I got mixed up converting from their "years since the present" value to CE/BCE!)?

We derive these claims from their Figure 3, which shows substantial decreases in ice accumulation (rightmost column) during these periods (~2.5kya and ~1.5-0.5 kya) as well as a marked reduction in WD activity, especially in the latter.

Lines 1472-3: Where do Singh et al. (2015) mention the elevation dependence?
They do not, explicitly, but this inference can be made by comparing their Table 2 (containing station locations and elevations) and Table 5 (containing trends in daily snowfall, precipitation days, etc). There is a negative correlation between daily snowfall trend and elevation.